# Learning Soft Sparse Shapes for Efficient Time-Series Classification

Zhen Liu [1 2]   Yicheng Luo [1]   Boyuan Li [1]   Emadeldeen Eldele [2]   Min Wu [† 2]   Qianli Ma [† 1]

## Abstract

Shapelets are discriminative subsequences (or shapes) with high interpretability in time series classification. Due to the time-intensive nature of shapelet discovery, existing shapelet-based methods mainly focus on selecting discriminative shapes while discarding others to achieve candidate subsequence sparsification. However, this approach may exclude beneficial shapes and overlook the varying contributions of shapelets to classification performance. To this end, we propose a **Soft** sparse **Shape**s (**SoftShape**) model for efficient time series classification. Our approach mainly introduces soft shape sparsification and soft shape learning blocks. The former transforms shapes into soft representations based on classification contribution scores, merging lower-scored ones into a single shape to retain and differentiate all subsequence information. The latter facilitates intra- and inter-shape temporal pattern learning, improving model efficiency by using sparsified soft shapes as inputs. Specifically, we employ a learnable router to activate a subset of class-specific expert networks for intra-shape pattern learning. Meanwhile, a shared expert network learns inter-shape patterns by converting sparsified shapes into sequences. Extensive experiments show that SoftShape outperforms state-of-the-art methods and produces interpretable results. Our source code is available at https://github.com/qianlima-lab/SoftShape.

## 1. Introduction

Time series classification (TSC) is a critical task with various real-world applications, such as human activity recog-

[1] School of Computer Science and Engineering, South China University of Technology, Guangzhou, China [2]Institute for Infocomm Research, Agency for Science, Technology and Research, Singapore. Correspondence to: Qianli Ma <qianlima@scut.edu.cn>, Min Wu <wumin@i2r.a-star.edu.sg>.

*Proceedings of the 42nd International Conference on Machine Learning*, Vancouver, Canada. PMLR 267, 2025. Copyright 2025 by the author(s).

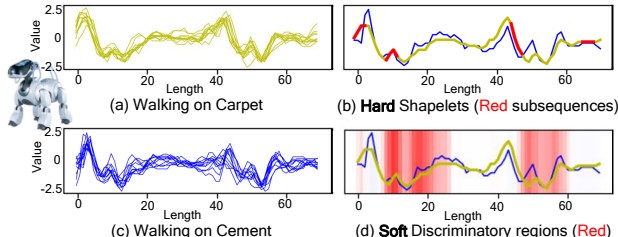

Figure 1: (a) Yellow and (c) Blue lines represent two time series class samples from the *SonyAIBORobotSurface1* UCR dataset. (b) Red subsequences denote shapelets for the "Walking on Carpet" class. (d) Red regions show discriminative parts of the "Walking on Carpet" class, with darker red indicating greater discriminative power.

nition (Lara & Labrador, 2012) and medical diagnostics (Wang et al., 2024). Unlike traditional data types, time series data consists of ordered numerical observations with temporal dependencies, which is often difficult for human intuition to understand. Recent studies (Mohammadi Foumani et al., 2024; Middlehurst et al., 2024; Jin et al., 2024; Ma et al., 2024) indicate that deep neural networks showed remarkable success in TSC, even without specialized knowledge. However, their black-box nature is a barrier to adopting effective models in critical applications such as healthcare, where insights about model decisions are important. Therefore, enhancing the interpretability of these models for TSC is a crucial ongoing issue.

Shapelets, which are discriminative subsequences that characterize target classes, offer a promising approach towards interpretable TSC models (Ye & Keogh, 2009). For instance, Figures 1(a) and 1(c) show time series classes of a robotic dog walking on carpet and cement (Vail & Veloso, 2004). The red subsequences shown in Figure 1(b) illustrate shapelets, as selected by Hou et al. (2016). However, identifying these shapelets is usually computationally intensive (Rakthanmanon & Keogh, 2013), as it requires evaluating subsequences across varying positions and lengths.

Existing methods (Grabocka et al., 2014; Li et al., 2020) address this issue via a shapelet transformation strategy for candidate subsequence sparsification. Specifically, they convert each subsequence into a feature that quantifies the distance between a time series sample and the subsequence (Ma

et al., 2020). Subsequently, this strategy was followed by many methods (Li et al., 2021; Le et al., 2024) to learn shapelet representations, thereby enhancing the model's interpretability. While efficient, this approach discards many subsequences from the entire time series in a *hard* manner, those that could be beneficial for the classification tasks. In addition, this approach fails to account for the varying importance of shapelets to classification. As shown in Figure 1(b), this leads to poor capture of class patterns in the third and fourth subsequences due to minimal class differences, while omitting many informative regions.

Recent advances in time series patch tokenization (Nie et al., 2023; Bian et al., 2024), i.e., converting time series into subsequence-based tokens, and in the mixture of experts (MoE) architectures (Riquelme et al., 2021; Fedus et al., 2022) inspired a new direction. In computer vision, MoE routers dynamically assign input patches to specialized experts, which can improve both efficiency and class-specific feature learning (Chen et al., 2022; Chowdhury et al., 2023). By analogy, time series shapelets, which can exhibit intra-class similarity and inter-class distribution, can act as patch tokens, with MoE enabling adaptive focus on the most discriminative subsequences. Yet, existing shapelet-based methods overlooked such an approach.

In this paper, we propose the **Soft** sparse **Shape**s (**SoftShape**) model for TSC, which replaces hard shapelet sparsification with soft shapelets. Specifically, we introduce a soft shape sparsification mechanism to reduce the number of learning shapes, thereby enhancing the efficiency of model training. To clarify our motivation, Figure 1(d) shows the classification results of InceptionTime (Ismail Fawaz et al., 2020) using multiple-instance learning (Early et al., 2024). Unlike the *hard* way depicted in Figure 1(b), our *soft* shapelet sparsification assigns weights to shapes based on their classification contribution scores, effectively preserving and differentiating all subsequence information from the time series. Furthermore, we enhance the discriminability of the learned soft shapes by combining: (1) Intra-shape patterns via MoE, where experts specialize in class-specific shapelet types; and (2) Inter-shape dependencies via a shared expert that models temporal relationships among sparsified shapes.

In summary, the major contributions are as follows:

- We propose SoftShape, a soft shapelet sparsification approach for TSC. This weighted aggregation of subsequences is based on classification contribution scores, avoiding information loss from hard filtering.

- We introduce a dual-pattern shapelet learning approach based on an MoE-driven intra-shape specialization and sequence-aware inter-shape modeling, improving the discriminative power of learned shape embeddings.

- We conduct extensive experiments on 128 UCR time series datasets, demonstrating the superior performance of our proposed method against state-of-the-art approaches while showcasing interpretable results.

## 2. Related Work

### 2.1. Deep Learning for Time-Series Classification

Recently, deep learning has received significant attention from scholars in time series classification tasks due to its powerful feature extraction capabilities (Ismail Fawaz et al., 2020; Zhang et al., 2024). For example, Ismail Fawaz et al. (2019) and Mohammadi Foumani et al. (2024) have systematically reviewed deep learning approaches for time series classification, focusing on different types of deep neural networks and deep learning paradigms, respectively. In addition, Middlehurst et al. (2024) and Ma et al. (2024) have conducted extensive experiments to evaluate recent time series classification methods. Their findings demonstrate that deep learning models can be effectively applied to time series classification tasks. Despite their strong performance, these deep learning-based time series classification methods often struggle to improve the interpretability of classification results.

### 2.2. Time-Series Classification with Shapelets

Ye & Keogh (2009) first introduced shapelets, providing good interpretability for time series classification results. Early methods (Mueen et al., 2011; Rakthanmanon & Keogh, 2013) selected the most discriminative subsequences as shapelets using an evaluation function, but the sparsification process was computationally intensive due to the numerous candidate subsequences. To tackle this problem, Hills et al. (2014) introduced the shapelet transform, converting all candidate subsequences into distance-based features for shapelet discovery. Recent studies (Qu et al., 2024; Le et al., 2024; Wen et al., 2025) have applied the shapelet transform strategy in deep learning models for time series classification, thereby improving model interoperability. However, these methods often handle candidate subsequences in a hard way, which easily leads to the loss of helpful temporal patterns from the time series sample.

More recently, Nie et al. (2023); Wu et al. (2025) showed that using patches of time series as input data enhances forecasting performance and reduces training time. Building on this, Luo & Wang (2024), Wang et al. (2024), and Wen et al. (2024) validated the effectiveness of patch techniques for time series classification. Eldele et al. (2024) further demonstrated that using time series patches as CNN inputs outperformed Transformer-based methods. However, these methods typically use all patches as inputs, leading to computational inefficiency in TSC for long sequences.

## 2.3. Mixture-of-Experts in Time Series

Mixture of experts (MoE) typically consist of multiple sub-networks (experts) and a learnable router (Shazeer et al., 2017; Zhou et al., 2022). MoE has been utilized in time series forecasting for decades (Zeevi et al., 1996; Ni et al., 2024) and recently applied to improve the efficiency of time series forecasting foundation models (Liu et al., 2024a; Shi et al., 2025). In addition, Huang et al. (2025) extended MoE to graph-based models for time series anomaly detection. Wen et al. (2025) introduced a gated router within MoE to integrate DNNs for learning time series representations, combining them with shapelet transform features to address the performance limitations of using shapelet transform alone in TSC. Unlike the above methods, we utilize the MoE router to activate class-specific experts for learning intra-shape local temporal patterns. Furthermore, we employ a shared expert to capture global temporal patterns across sparsified shapes from the same time series, thereby enhancing the discriminative power of shape embeddings.

## 3. Preliminaries

### 3.1. Problem Definition

In this study, we focus on the use of deep learning models for univariate time series classification. Let the time series dataset be represented as $\mathcal{D} = \{(\mathcal{X}_n, \mathcal{Y}_n)\}_{n=1}^N$, where $N$ denotes the total number of time series samples. Each univariate time series $\mathcal{X}_n = \{x_1, x_2, \ldots, x_T\}$ is an ordered sequence of $T$ real-valued observations. The corresponding label $\mathcal{Y}_n \in \mathbb{R}^C$ is a one-hot encoded vector, where each value $y_c \in \{0, 1\}$ indicates whether $\mathcal{X}_n$ belongs to class $c$, with $C$ representing the total number of classes. Given a deep learning model parameterized by $\theta$, the objective of the TSC task is to optimize the function $f_\theta : \mathbb{R}^T \to \mathbb{R}^C$ such that it can accurately predict the label $\mathcal{Y}_n$ corresponding to any input time series $\mathcal{X}_n$.

### 3.2. Time Series Shapelets

Given a time series $\mathcal{X}_n$, a subsequence of length $m$, where $m < T$, is denoted as $\mathcal{S}_{n,p}^m = \{x_p, x_{p+1}, \ldots, x_{p+m-1}\}$. Here, $p$ denotes the starting index of $\mathcal{S}_{n,p}^m$, where $0 \leq p < T - m$. All subsequences of length $m$ from the time series $\mathcal{X}_n$ can be extracted using a sliding window of size $m$ with a fixed step size $q$ (commonly $q = 1$), traversing from $t = 1$ to $T$. Thus, the set of all subsequences of length $m$ from $\mathcal{X}_n$ can be defined as $\mathcal{S}_n^m = \{\mathcal{S}_{n,p}^m \mid 0 \leq p < T - m\}$.

Due to the length of the subsequences varying within the range $2 \leq m \leq T - 1$, the candidate set $\mathcal{A}$ of all possible subsequences for the time series dataset $\mathcal{D}$ is expressed as:

$$\mathcal{A} = \bigcup_{n=1}^N \bigcup_{m=2}^{T-1} \mathcal{S}_n^m = \{\mathcal{S}_{n,p}^m \mid 0 \leq p < T - m\}. \quad (1)$$

Shapelets are defined as a subset of discriminative subsequences within $\mathcal{A}$ that maximally represent the class of each time series $\mathcal{X}_n$ (Ye & Keogh, 2009). Given that every shape in $\mathcal{A}$ could potentially serve as a shapelet, a brute-force search to evaluate all subsequences of each $\mathcal{X}_n$ using an evaluation function becomes computationally prohibitive as the number of samples $N$ and the sequence length $T$ increase. Thus, the design of appropriate sparsification techniques for the set $\mathcal{A}$ is crucial for enhancing the computational efficiency of shapelet-based TSC methods.

## 4. Methodology

### 4.1. Model Overview

The overall architecture of the SoftShape model is illustrated in Figure 2. SoftShape begins with a shape embedding layer that transforms the input time series into shape embeddings corresponding to multiple equal-length subsequences. These embeddings are normalized with a norm layer and then processed using soft shape sparsification (see Section 4.2). Specifically, an attention head assigns weight scores to each shape embedding based on its classification contribution. High-weight embeddings are scaled by their scores to form soft shapes, while low-weight embeddings are merged into a single shape for sparsification. The normalized sparsified shape embeddings are then input into the soft shape learning block to learn temporal patterns (see Section 4.3). Within this block, a MoE router activates a few class-specific experts to capture intra-shape temporal patterns (see Section 4.3.1). Meanwhile, these sparsified soft shapes are transformed into sequences, with a shared expert learning inter-shape temporal patterns (see Section 4.3.2). Finally, the sparsified soft shape embeddings, along with the intra-shape and inter-shape embeddings learned by the soft shape learning block, are combined in a residual way. After stacking $L$ layers, the output is passed to a linear layer for classification training (see Section 4.4).

### 4.2. Soft Shape Sparsification

For each input time series $\mathcal{X}_n$, we use a 1D convolutional neural network (CNN) as the shape embedding layer to obtain $M = \frac{T-m}{q} + 1$ $(q < m)$ overlapping subsequence embeddings, denoted $\hat{\mathcal{S}}_n^m = \{\hat{\mathcal{S}}_{n,p}^m \mid 0 \leq p < T - m\}$. Each shape embedding $\hat{\mathcal{S}}_{n,p}^m$ is computed using the following convolution operation:

$$\hat{\mathcal{S}}_{n,p}^m = \left\{ \sum_{i=0}^{m-1} \mathbf{W}_i \mathcal{S}_{n,p}^m \mid p = 0, q, 2q, \ldots, T - m \right\}, \quad (2)$$

where $\mathbf{W}_i$ denotes the weights of the $i$-th filter within the kernel of the CNN, and $\hat{\mathcal{S}}_{n,p}^m \in \mathbb{R}^d$, with $d$ representing the dimension of the shape embedding. Like patch-based methods (Nie et al., 2023; Eldele et al., 2024), we incorporate

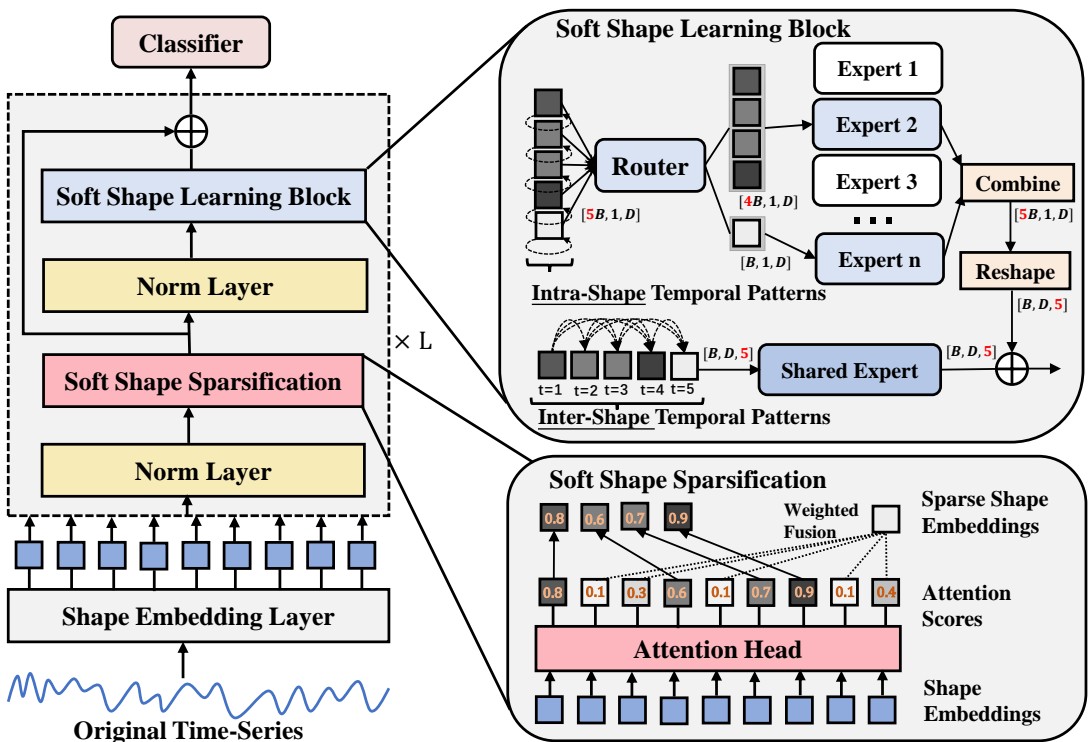

Figure 2: The general architecture of SoftShape model. SoftShape mainly involves: a) Soft Shape Sparsification converts shapes into soft forms and fuses those with lower scores into a single shape. b) Soft Shape Learning Block employs a MoE router to activate class-specific experts to learn intra-shape patterns and utilizes a shared expert to learn inter-shape patterns.

learnable positional embeddings into shape embeddings to capture temporal dependencies within the time series.

To identify the most discriminative shape embeddings in $\hat{\mathcal{S}}_n^m$, we employ a parameter-shared attention head across soft shape learning blocks at different depths. Unlike the self-attention mechanism in Transformers, we employ a gated attention mechanism (Ilse et al., 2018) that assumes independence among shape embeddings. This design facilitates label-guided identification of key shapes while maintaining linear computational complexity with respect to the number of shapes (Early et al., 2024). The attention head evaluates the contribution score of each $\hat{\mathcal{S}}_{n,p}^m$ for classification training, defined as follows:

$$\alpha(\hat{\mathcal{S}}_{n,p}^m) = \sigma\left(\mathbf{W}_2 \tanh\left(\mathbf{W}_1\hat{\mathcal{S}}_{n,p}^m + \mathbf{b}_1\right) + \mathbf{b}_2\right), \quad (3)$$

where $\sigma(\cdot)$ is the sigmoid function outputting $\alpha \in (0,1)$, $\tanh(\cdot)$ is the hyperbolic tangent function, $\mathbf{W}_1 \in \mathbb{R}^{d_{\text{attn}} \times d}$ and $\mathbf{W}_2 \in \mathbb{R}^{1 \times d_{\text{attn}}}$ are the weights of a linear layer, and $\mathbf{b}_1 \in \mathbb{R}^{d_{\text{attn}}}$ and $\mathbf{b}_2 \in \mathbb{R}$ are the bias terms.

The value of $\alpha(\hat{\mathcal{S}}_{n,p}^m)$ closer to 1 indicates a greater contribution to enhancing classification performance. Hence, we

can perform soft shape sparsification using scores from the attention head based on their ranking. For each attention score of $\hat{\mathcal{S}}_{n,p}^m$ ranked within the top $\eta$ proportion among the total $J = \frac{T-m}{q} + 1$ shape embeddings in $\hat{\mathcal{S}}_n^m$, we convert them into soft shape embeddings as follows:

$$\widetilde{\mathcal{S}}_{n,p}^m = \alpha(\hat{\mathcal{S}}_{n,p}^m)\hat{\mathcal{S}}_{n,p}^m. \quad (4)$$

For those with scores below the top $\eta$ proportion, we fuse them into a single shape embedding as follows:

$$\widetilde{\mathcal{S}}_{n,\text{fused}}^m = \sum_{p \in \mathcal{E}} \alpha(\hat{\mathcal{S}}_{n,p}^m)\hat{\mathcal{S}}_{n,p}^m, \quad (5)$$

where $\mathcal{E}$ denotes the index of scores that are below the top $\eta$ proportion. Finally, the soft shape embeddings are sorted according to their original order in $\hat{\mathcal{S}}_n^m$, with the fused embedding $\widetilde{\mathcal{S}}_{n,\text{fused}}^m$ appended at the end to form the sparsified soft shape embeddings $\widetilde{\mathcal{S}}_n^m$.

Through this sparsification process, we transform the time series $\mathcal{X}_n$ into sparsified soft shape embeddings $\widetilde{\mathcal{S}}_n^m$. Compared to directly using all shape embeddings in $\hat{\mathcal{S}}_n^m$ as input, the number of shape embeddings in $\widetilde{\mathcal{S}}_n^m$ is reduced, thereby decreasing the cost of the model's computation.

## 4.3. Soft Shape Learning Block

### 4.3.1. INTRA-SHAPE LEARNING

Intuitively, shapelets from different classes show distinct temporal patterns, whereas those from the same class are more similar. Recently, Chen et al. (2022); Chowdhury et al. (2023) theoretically indicated that the MoE router can direct input embeddings with similar class patterns to the same expert while filtering out class-irrelevant features. This allows the router to selectively activate a subset of experts (sub-networks) for each shape embedding, enabling class-specific feature learning while reducing computational overhead compared to activating all experts simultaneously. Therefore, we introduce a module that employs the MoE router to activate class-specific experts for learning intra-shape temporal patterns, thereby enhancing the discriminative capability of each soft shape embedding. The function of the MoE router is defined as:

$$G(\widetilde{\mathcal{S}}_{n,p}^m) = \text{TOP}_k(\text{softmax}(\mathbf{W}_t \widetilde{\mathcal{S}}_{n,p}^m)), \qquad (6)$$

$$\text{TOP}_k(v) = \begin{cases} v_i, & \text{if } v_i \text{ is in the top } k \text{ values of } v, \\ 0, & \text{otherwise.} \end{cases} \qquad (7)$$

where $\mathbf{W}_t \in \mathbb{R}^{\hat{C} \times d}$ are the learning weights of the router that convert the sparsified shape embedding $\widetilde{\mathcal{S}}_{n,p}^m$ into a vector of corresponding to the total number of experts $\hat{C}$. The $\text{TOP}_k(v)$ is applied after the softmax function to prevent the values of $G(\widetilde{\mathcal{S}}_{n,p}^m)$ being zero almost everywhere during training (Riquelme et al., 2021), particularly when $k = 1$.

In the paper, each MoE expert employs a lightweight MLP network to learn class-specific patterns using each soft shape embedding as input. The $e$-th expert function is defined as:

$$h_e(\theta, \widetilde{\mathcal{S}}_{n,p}^m) = \hat{G}_e(\widetilde{\mathcal{S}}_{n,p}^m)\text{GeLU}(\mathbf{W}_e \widetilde{\mathcal{S}}_{n,p}^m + \mathbf{b}_e), \quad (8)$$

$$\hat{G}_e(\widetilde{\mathcal{S}}_{n,p}^m) = \frac{e^{G_e(\widetilde{\mathcal{S}}_{n,p}^m)}}{\sum_{i \in \text{TOP}_k(v)} e^{G_i(\widetilde{\mathcal{S}}_{n,p}^m)}}, \qquad (9)$$

where $\mathbf{W}_e$ and $\mathbf{b}_e$ represent the weight and bias terms of the MLP network, and $\text{GeLU}(\cdot)$ denotes the gaussian error linear unit activation function. The MoE experts' parameters are shared across soft shape learning blocks at different depths, ensuring the efficiency of parameter optimization.

However, the MoE router often tends to converge to a state where it assigns higher weights to a few favored experts. This imbalance can cause the favored experts to train faster and dominate the expert selection process, resulting in underutilization and potential underfitting of less frequently chosen experts. To address this issue, following (Shazeer et al., 2017), we employ two loss functions as follows:

$$\mathcal{L}_{\text{imp}}(\widetilde{\mathcal{S}}_{n,p}^m) = \mathbf{W}_{\text{imp}}\text{CV}\left(\sum_{\widetilde{\mathcal{S}}_{n,p}^m \in \widetilde{\mathcal{S}}_n^m} G(\widetilde{\mathcal{S}}_{n,p}^m)\right)^2, \quad (10)$$

where $\mathbf{W}_{\text{imp}}$ is the learning weight parameters and $\text{CV}(\cdot)$ denotes the coefficient of variation, thus ensuring all experts contribute equally.

$$\mathcal{L}_{\text{load}}(\widetilde{\mathcal{S}}_{n,p}^m) = \mathbf{W}_{\text{load}}\text{CV}(\text{Load}(\widetilde{\mathcal{S}}_{n,p}^m))^2, \qquad (11)$$

where $\mathbf{W}_{\text{load}}$ is the learning weight parameters for load balancing and $\text{Load}(\widetilde{\mathcal{S}}_{n,p}^m)$ is the number of soft shape embeddings assigned to each expert, thereby reducing the risk of underutilization. By jointly optimizing $\mathcal{L}_{\text{imp}}$ and $\mathcal{L}_{\text{load}}$ in the overall loss function, we can alleviate the likelihood of underfitting in some experts. In addition, following Zoph et al. (2022), we apply a residual connection between the input shape embedding $\widetilde{\mathcal{S}}_{n,p}^m$ and the output $h_e(\theta, \widetilde{\mathcal{S}}_{n,p}^m)$ to enhance training stability.

### 4.3.2. INTER-SHAPE LEARNING

While the MoE router activates a few experts to learn class-specific temporal patterns within each soft shape, it treats the multiple shapes extracted from each time series sample independently, thereby overlooking the temporal dependencies among shapes. In general, learning intra-shape temporal patterns is beneficial for capturing local discriminative features, but it can be difficult to learn the global temporal patterns from the time series that are vital for classification. To this end, we introduce a shared expert to learn the temporal patterns among sparsified shapes within the time series.

The sparsified soft shape embeddings are first transformed into a sequence where each shape is treated as an individual unit. This transformation is defined as:

$$\mathcal{Q}_n^m = (\widetilde{\mathcal{S}}_n^m)^T, \quad \text{where} \quad \widetilde{\mathcal{S}}_n^m \in \mathbb{R}^{B \times Num \times d}, \quad (12)$$

where $\mathcal{Q}_n^m \in \mathbb{R}^{B \times d \times Num}$, and $B$ denotes the batch size, $Num = J \times \eta + 1$ is the number of sparsified soft shapes of each time series, and $d$ is the dimension of one soft shape embedding. Recent studies (Middlehurst et al., 2024) have shown that CNN-based models perform remarkably in TSC tasks. In this work, we employ a CNN-based Inception module (Ismail Fawaz et al., 2020) as the shared expert. The shared expert comprises three 1D convolutional layers with different kernel sizes, sliding along the third dimension of $\mathcal{Q}_n^m$ from 0 to $Num$. This allows it to learn multi-resolution temporal patterns across soft shapes effectively. For raw input time series samples, the input dimension containing $B$ univariate time series can be represented as $(B, 1, T)$, where $T$ is the length of each time series. Traditional CNN-based methods (Ismail Fawaz et al., 2020) typically transform each point in the time series into $d$-dimensional embeddings, resulting in an input dimension of $(B, d, T)$. Unlike the above, we treat each shape as a sequence point and convert it into $d$-dimensional embeddings for training.

As shown in Equations (1) and (5), with a higher sparsity rate $(1 - \eta)$, the actual sequence length $Num$ of $\mathcal{Q}_n^m$ fed

into the shared expert is significantly smaller than $T$, thus reducing the model's computational load. The shared expert ensures efficient learning of temporal patterns across sparsified soft shapes, better capturing global features of the time series for classification training.

### 4.4. The Training of SoftShape

The SoftShape model stacks $L$ layers of the soft shape learning block for classification training. Each layer's output is processed using a GeLU$(\cdot)$ activation function. The final layer outputs three types of embeddings: the sparsified soft shape embeddings, the intra-shape embeddings, which are combined through a residual way denoted as $O_n^m$. To accurately represent the learned attention scores for each shape embedding associated with the target class, we employ conjunctive pooling (Early et al., 2024) for training:

$$\hat{\mathcal{Y}}_n = \frac{1}{Num} \sum_{i=0}^{Num} \alpha(O_n^m)\phi(O_{n,i}^m), \qquad (13)$$

where $\phi$ denotes a linear layer as the classifier. For all time series samples, we use the cross-entropy loss for classification training, defined as:

$$\mathcal{L}_{\text{ce}}(\mathcal{X}, \mathcal{Y}) = -\frac{1}{B} \sum_{i=1}^{B} \sum_{j=1}^{C} 1\{y_i = j\} \log(\hat{\mathcal{Y}}_i^{\,j}), \quad (14)$$

where $\hat{\mathcal{Y}}_i^{\,j}$ is the predicted class probability at the $j$-th class for the sample $\mathcal{X}_i$. Therefore, the overall training objective for SoftShape is given by:

$$\mathcal{L}_{\text{total}} = \mathcal{L}_{\text{ce}} + \lambda(\mathcal{L}_{\text{imp}} + \mathcal{L}_{\text{load}}), \qquad (15)$$

where $\lambda$ is a hyperparameter to adjust the training loss ratio. Additionally, the pseudo-code for SoftShape can be found in Algorithm 1 located in the Appendix.

## 5. Experiments

To evaluate the performance of various methods for TSC, we conduct experiments using the UCR time series archive (Dau et al., 2019), a widely recognized benchmark in TSC (Ismail Fawaz et al., 2019). Many datasets in the UCR archive contain a significantly higher number of test samples compared to the training samples. Also, the original UCR time series datasets lack a specific validation set, increasing the risk of overfitting in deep learning methods. Following (Dau et al., 2019; Ma et al., 2024), we merge the original training and test sets of each UCR dataset. These merged datasets are then divided into train-validation-test sets at a ratio of 60%-20%-20%. This paper compares SoftShape against 19 baseline methods, categorized as follows:

* Deep Learning Methods (DL):

  – *CNN-based (DL-CNN):* FCN (Wang et al., 2017), T-Loss (Franceschi et al., 2019), SelfTime (Fan et al., 2020), TS-TCC (Eldele et al., 2021), TS2Vec (Yue et al., 2022), TimesNet (Wu et al., 2023b), InceptionTime (Ismail Fawaz et al., 2020), LightTS (Campos et al., 2023), ShapeConv (Qu et al., 2024), ModernTCN (Luo & Wang, 2024), and TSLANet (Eldele et al., 2024).
  – *Transformer-based (DL-Trans):* TST (Zerveas et al., 2021), PatchTST (Nie et al., 2023), Shapeformer (Le et al., 2024), and Medformer (Wang et al., 2024).
  – *Foundation models (DL-FM):* GPT4TS (Zhou et al., 2023) and UniTS (Gao et al., 2024).

* Non-Deep Learning Methods (Non-DL): RDST (Guillaume et al., 2022) and MultiRocket-Hydra (MR-H) (Dempster et al., 2023).

For information on datasets, baselines, and implementation details, please refer to Appendix A.

### 5.1. Main Results

Table 1: Test classification accuracy comparisons on 128 UCR time series datasets. The best results are in **bold**, and the second best results are underlined.

| | Methods | Avg. Acc | Avg. Rank | Win | P-value |
|---|---|---|---|---|---|
| DL-CNN | FCN | 0.8296 | 9.53 | 13 | 1.43E-12 |
| | T-Loss | 0.8325 | 11.12 | 9 | 2.95E-14 |
| | SelfTime | 0.8017 | 13.80 | 0 | 4.53E-25 |
| | TS-TCC | 0.7807 | 13.96 | 0 | 1.60E-15 |
| | TS2Vec | 0.8691 | 8.43 | 9 | 1.69E-15 |
| | TimesNet | 0.8367 | 10.13 | 7 | 4.22E-15 |
| | InceptionTime | 0.9181 | 4.05 | 29 | 7.39E-06 |
| | ShapeConv | 0.7688 | 13.91 | 5 | 3.58E-24 |
| | ModernTCN | 0.7938 | 11.37 | 9 | 1.78E-18 |
| | TSLANet | 0.9205 | 3.68 | 31 | 1.06E-03 |
| DL-Trans | TST | 0.7755 | 13.54 | 1 | 2.00E-19 |
| | PatchTST | 0.8265 | 9.56 | 12 | 1.27E-15 |
| | Medformer | 0.8541 | 9.26 | 7 | 7.15E-16 |
| DL-FM | GPT4TS | 0.8593 | 9.34 | 6 | 7.89E-16 |
| | UniTS | 0.8502 | 9.66 | 5 | 1.41E-13 |
| Non-DL | RDST | 0.8897 | 6.41 | 23 | 7.54E-10 |
| | MR-H | 0.8972 | 5.51 | 29 | 3.80E-07 |
| **SoftShape (Ours)** | | **0.9334** | **2.72** | **53** | - |

The classification accuracies on the test sets of the 128 UCR datasets are presented in Table 1, with detailed results provided in Appendix B.1. The average ranking (*Avg. Rank*) reflects the method's relative position based on test accuracy, where lower values signify higher accuracy. *Win* indicates the number of datasets where the baseline achieves the highest test accuracy. The *P-value* from the Wilcoxon signed-rank test (Demšar, 2006) assesses the significance of

differences in test accuracy between pairwise methods. A *P-value* $< 0.05$ suggests SoftShape outperforms the baseline.

Due to the high computational cost of LightTS (Campos et al., 2023) and Shapeformer (Le et al., 2024), their results are reported only on 18 selected UCR time series datasets in Table 10 of Appendix B.1. Detailed reasons for 18 UCR dataset selection are in Appendix B.3. The experimental results in Table 1 and Table 10 highlight SoftShape's superior classification performance, demonstrating its effectiveness in TSC. Also, the P-value results confirm the statistical significance of SoftShape's performance compared to baselines, highlighting its capacity to capture the discriminative patterns of time series data. The critical diagram (Figure 6) in the Appendix further supports the statistical significance of SoftShape's performance improvement.

### 5.2. Ablation Study

Table 2: Test accuracy of ablation study on 128 UCR datasets. The best results are in **bold**, and the second best results are underlined.

| Methods | Avg. Acc | Avg. Rank | Win | P-value |
|---|---|---|---|---|
| w/o Soft Sparse | 0.9123 | 3.04 | 29 | 4.69E-06 |
| w/o Intra | 0.9245 | 2.75 | 31 | 5.39E-04 |
| w/o Inter | 0.9022 | 3.74 | 19 | 1.81E-09 |
| w/o Intra & Inter | 0.8696 | 5.02 | 11 | 4.35E-16 |
| with Linear Shape | 0.9164 | 3.23 | 22 | 1.80E-09 |
| **SoftShape (Ours)** | **0.9334** | **2.04** | **60** | - |

The ablation studies include: 1) **w/o Soft Sparse** removes the soft sparse process, using only the selected top $\eta$ shape embeddings for training; 2) **w/o Intra** eliminates the intra-shape learning process via the class-specific experts; 3) **w/o Inter** removes the inter-shape learning process via the shared expert; 4) **w/o Intra & Inter** excludes both the intra-shape learning and inter-shape learning processes; 5) **with Linear Shape** employs a linear layer commonly used in (Nie et al., 2023; Wang et al., 2024) to replace the 1D CNN in Equation (2) as the shape embedding layer.

Table 2 presents the statistical ablation results, with detailed results provided in Appendix B.2. **w/o Soft Sparse** reduces accuracy and lowers the *Avg. Rank*, indicating the importance of this process in capturing the most helpful shape embeddings for classification. Similarly, **w/o Intra** highlights the significant role of the intra-shape module in SoftShape. Moreover, **w/o Inter** leads to a large drop in performance, underscoring the critical contribution of inter-shape relationships in learning global patterns of time series. When both the intra- and inter-shape learning processes are excluded (**w/o Intra & Inter**), performance further degrades, demonstrating that these two processes are foundational to SoftShape's effectiveness. Finally, **with Linear Shape** con-

firms the superiority of the CNN architecture for learning shape embeddings. These results confirm the importance of each component in the SoftShape model.

### 5.3. Shape Sparsification and Learning Analysis

Table 3: Test accuracy of different sparse ratios on 18 UCR datasets. The best results are in **bold**, and the second best results are underlined.

| Sparse Ratio | Avg. Acc | Avg. Rank | Win | P-value |
|---|---|---|---|---|
| 0% | 0.9461 | 2.44 | 4 | - |
| 10% | **0.9469** | **2.39** | **7** | 2.97E-01 |
| 30% | 0.9448 | 2.78 | 5 | 2.66E-01 |
| 50% | 0.9453 | 2.61 | 6 | 3.46E-01 |
| 70% | 0.9323 | 3.89 | 5 | 9.37E-03 |
| 90% | 0.9261 | 4.50 | 2 | 4.02E-04 |

Evaluating all 128 UCR time series datasets takes considerable time, so we selected 18 datasets from the UCR time series archive for analysis and subsequent experiments. Table 3 presents the test results when combining shape embeddings with weighted fusion at varying sparse ratio $(1 - \eta)$ proportions during sparsification, with detailed results provided in Appendix B.3. When $(1 - \eta) \leq 50\%$, test accuracy showed no significant decline compared to the non-sparsified case (0%), with all p-values $> 0.05$. Conversely, when $(1 - \eta) > 50\%$, particularly at 90%, performance declined significantly compared to 0%, indicating that excessive sparsification can remove some shape embeddings beneficial for capturing inter-shape global temporal patterns. Furthermore, the performance degradation under high sparsity ratios suggests that hard shapelet-based methods discard numerous informative subsequences, which may result in the loss of critical patterns useful for classification. As performance at 50% is statistically comparable to 0%, we set $\eta$ to 50% in the main experiments to optimize the computational efficiency of the soft shape learning block.

Table 4 illustrates the effect of the number of activated experts $(k)$ selected by the MoE router during intra-shape learning, with detailed results provided in Appendix B.3. The results show that when $k = 3$ or $k = 4$, the average rank is worse than when $k = 1$ or $k = 2$. This suggests that activating a greater number of class-specific experts for each soft shape increases computational cost while diminishing the class discriminability of the intra-shape embeddings.

Also, we select one non-deep learning method (MR-H) and three deep learning baselines for runtime analysis. MR-H uses randomly initialized convolutional kernels for feature extraction, with training times measured on a CPU. The deep learning methods are evaluated on a GPU. Inception-Time is an advanced deep learning method described in

Table 4: Test accuracy comparison of the number of activated experts on 18 UCR datasets. The best result is in **bold**.

| Activated Experts | $k$=1 | $k$=2 | $k$=3 | $k$=4 |
|---|---|---|---|---|
| Avg. Rank | **1.89** | 1.94 | 2.78 | 2.39 |

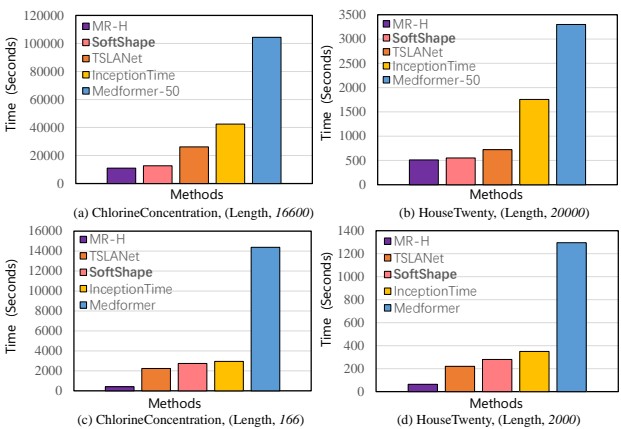

Figure 3: Running time analysis on the *ChlorineConcentration* and *HouseTwenty* datasets.

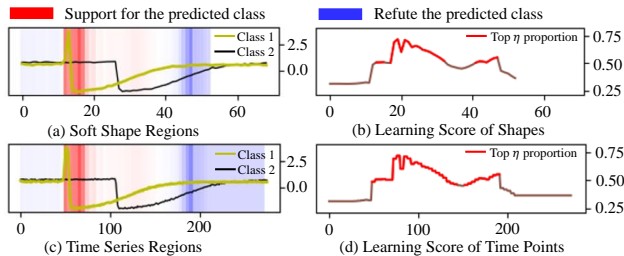

Figure 4: The MIL visualization on the *Trace* dataset.

(Middlehurst et al., 2024). TSLANet and Medformer are recent patch-based models utilizing CNNs and Transformers, respectively. To evaluate SoftShape's sparsification efficiency, we select two datasets: *ChlorineConcentration* (4,307 samples, length 166) and *HouseTwenty* (159 samples, length 2,000). In Figures 3(a) and 3(b), sequence lengths are extended to 16,600 and 20,000 by sample concatenation to simulate long-sequence scenarios. Given Medformer's high computational cost for long sequences, we report only its runtime at 50 epochs (out of a maximum of 500), denoted as Medformer-50. As shown in Figures 3(a) and 3(b), SoftShape outperforms all deep learning baselines and is slightly slower than MR-H. Specifically, Medformer's self-attention leads to higher training time with long sequences, while SoftShape's linear-complexity gated attention ensures greater efficiency. In short-sequence [Figure 3(c)] and small-sample [Figure 3(d)] scenarios, SoftShape exhibits slightly higher runtime than TSLANet and substantially higher than MR-H. In these settings, sparsification offers limited efficiency gains, as the number of sparsified shapes does not decrease significantly compared to a greater number of training samples with long sequences. Additionally, Table 14 in Appendix B.3 presents the inference times and parameter counts for the settings in Figures 3(c) and 3(d).

### 5.4. Visualization Analysis

To evaluate SoftShape's effectiveness and interpretability, we use Multiple Instance Learning (MIL) (Early et al., 2024) for visualization. MIL identifies key time points in time se-

ries that improve classification performance. This study applies MIL to highlight significant shapes. Figure 4 shows the discriminative regions and attention scores learned by SoftShape on *Class 1* of the *Trace* dataset. Deeper red indicates beneficial regions for classification, while deeper blue shows less relevant ones. Figures 4(a) and 4(b) treat each shape (length $m = 64$) as one sequence point, while Figures 4(c) and 4(d) map these results to the entire time series. SoftShape assigns higher attention scores to subsequences with significant differences between *Class 1* and *Class 2*, while sparsifying less discriminative regions. These findings indicate that SoftShape learns highly discriminative shapes as shapelets, enhancing interpretability through attention scores. Further MIL visualisations on the *Lightning2* dataset in the Appendix of Figure 7 support this finding.

The t-distributed Stochastic Neighbor Embedding (t-SNE) (Van der Maaten & Hinton, 2008) is used to visualize shape embeddings learned by SoftShape. Figure 5(a) shows input shape embeddings from 1D CNN (Eq. (2)) on the *CBF* dataset. Figures 5(b) and 5(c) present sparsified intra-shape and inter-shape embeddings from the soft shape learning block, while Figure 5(d) displays final output sparsified shape embeddings for classification. Compared to Figure 5(a), the intra-shape embeddings [Figure 5(b)] learned by activating class-specific experts form clusters for samples of the same class but still mix some samples from different classes. The inter-shape embeddings [Figure 5(c)] learned by a shared expert effectively separate different classes while dispersing same-class samples into multiple clusters. Combining intra- and inter-shape embeddings, Figure 5(d) balances class distinction and clusters samples of the same class closely together. These results indicate that SoftShape leverages intra- and inter-shape patterns to improve shape embedding discriminability. Further t-SNE visualization on the *TwoPatterns*, *Fiftywords*, and *ECG200* datasets with different classification difficulty can be seen in the Appendix of Figures 8, 9, and 10, respectively.

### 5.5. Hyperparameter Analysis

Table 5 presents the classification results of SoftShape for various sliding fixed step sizes $q$. We observed worse *Avg.*

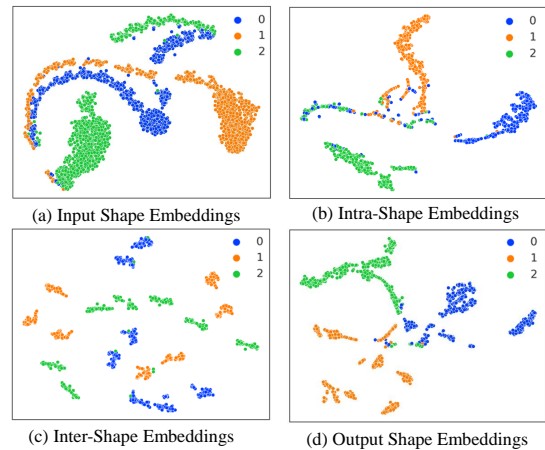

(a) Input Shape Embeddings  (b) Intra-Shape Embeddings

(c) Inter-Shape Embeddings  (d) Output Shape Embeddings

Figure 5: The t-SNE visualization on the *CBF* dataset.

Table 5: Test accuracy comparison of different sliding window sizes on 18 UCR datasets. The best result is in **bold**.

| Sliding Step Size | $q=1$ | $q=2$ | $q=3$ | $q=4$ | $q=m$ |
|---|---|---|---|---|---|
| Avg. Rank | 3.83 | 2.78 | 2.44 | **1.78** | 2.94 |

Table 6: Test accuracy comparison of model depth on 18 UCR datasets. The best result is in **bold**.

| Depth | $L=1$ | $L=2$ | $L=3$ | $L=4$ | $L=5$ | $L=6$ |
|---|---|---|---|---|---|---|
| Avg. Rank | 3.72 | **2.61** | 3.06 | 3.33 | 3.44 | 3.06 |

*Rank* values when $q=1$ or $q=m$ (non-overlapping). With $q=1$, the maximum subsequences result in redundancy and irrelevant information. Conversely, $q=m$ generates the fewest subsequences, possibly missing important discriminative subsequences. Table 6 examines the influence of model depth $L$, revealing that SoftShape achieves the lowest *Avg. Rank* at $L=2$, demonstrating shape embeddings capture discriminative features effectively, even at low depths. Detailed results of Tables 5 and 6 provided in Appendix B.4. Also, Tables 18, 19, 20, and 21 in Appendix B.4 present statistical test results on the impact of the maximum number of class-specific experts, class-specific expert networks, shared expert networks, and the hyperparameter $\lambda$.

To evaluate the impact of the hyperparameter shape length $m$ on classification performance, we conducted three experiments: (1) Val-Select: Choose a fixed $m$ for one SoftShape model using the validation set. (2) Fixed-8: Set $m=8$ for one SoftShape model. (3) Multi-Seq: Use fixed lengths (8, 16, 32) in parallel three SoftShape models, with cross-scale residual fusion (Liu et al., 2023) for classification. As shown in Table 7 (detailed results in Appendix B.4), there was no significant difference in performance between

Val-Select and Multi-Seq (p-value $< 0.05$). The Val-Select model also has fewer parameters and a lower runtime. In addition, Tables 22 and 23 in Appendix B.5 present analyses of SoftShape's time series forecasting performance.

Table 7: Test accuracy comparison of shape length $m$ on 18 UCR datasets. The best result is in **bold**.

| Methods | Val-Select | Fixed-8 | Multi-Seq |
|---|---|---|---|
| Avg. Rank | 1.72 | 2.28 | **1.44** |
| P-value | 2.88E-01 | 4.68E-02 | - |

## 6. Conclusion

In this paper, we propose a soft sparse shapes model for efficient time series classification. Specifically, we introduce a soft shape sparsification mechanism that merges low-discriminative subsequences into a single shape based on learned attention scores, reducing the number of shapes for shapelet learning. Moreover, we propose a soft shape learning block for learning intra-shape and inter-shape temporal patterns, enhancing shape embedding discriminability. Extensive experiments demonstrate that SoftShape achieves state-of-the-art classification performance while providing interpretable results. However, this study does not consider the modelling of relationships among multiple variables of the time series. In the future, we will investigate deep learning methods for multivariate TSC.

## Acknowledgements

We thank the anonymous reviewers for their helpful feedbacks. We thank Professor Eamonn Keogh and all the people who have contributed to the UCR time series classification archive. Zhen Liu and Yicheng Luo equally contributed to this work. The work described in this paper was partially funded by the National Natural Science Foundation of China (Grant No. 62272173), the Natural Science Foundation of Guangdong Province (Grant Nos. 2024A1515010089, 2022A1515010179), the Science and Technology Planning Project of Guangdong Province (Grant No. 2023A0505050106), the National Key R&D Program of China (Grant No. 2023YFA1011601), and the China Scholarship Council program (Grant No. 202406150081).

## Impact Statement

Our proposed work `SoftShape` aims to advance the field of Machine Learning by providing a more efficient and interpretable model for time series classification across various applications. There are many potential societal consequences of our work, none which we feel must be specifically highlighted here.

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

# A. Experimental Setup

## A.1. Datasets

In this study, we employ the UCR time series archive (Dau et al., 2019), a widely recognized benchmark for time series classification tasks (Ismail Fawaz et al., 2019; Middlehurst et al., 2024), to systematically evaluate the effectiveness of the proposed method. This archive encompasses datasets from diverse real-world domains, including human activity recognition, medical diagnostics, and intelligent transportation systems. The UCR archive consists of 128 distinct time series datasets.

As shown in Table 8, many datasets in the UCR archive have a notably greater number of test samples than training samples, such as the *CBF*, *DiatomSizeReduction*, *ECGFiveDays*, *MoteStrain*, and *TwoLeadECG* time series datasets. Moreover, the original UCR time series datasets lack a specific validation set. The limited amount of training samples and the lack of a validation set present challenges in fairly comparing various deep learning-based time series classification methods.

Following the experimental settings outlined in (Ma et al., 2024) and the recommendations of (Dau et al., 2019), we integrate the raw training and test sets and employ a five-fold cross-validation strategy to partition the dataset into training, validation, and test sets in a 3:1:1 ratio. Consistent with the approach in (Ma et al., 2024), each fold is sequentially designated as the test set, while the remaining four folds are randomly divided into training and validation sets in a 3:1 ratio. This cross-validation procedure ensures that each time series sample in the dataset is utilized as a test sample at once, thereby providing a comprehensive evaluation of classifier performance. Consequently, the specific indexing of samples during cross-validation does not substantially influence the final classification results, as each sample contributes to assessing the model's performance.

## A.2. Baselines

We conduct a comparative analysis of SoftShape against 19 baseline methods. Notably, nine of these baseline deep learning methods are derived from the experimental setting presented in (Ma et al., 2024), including FCN, T-Loss, SelfTime, TS-TCC, TST, TS2Vec, TimesNet, PatchTST, and GPT4TS. While Ma et al. (2024) review a broad range of techniques for time series pre-training, its experimental design for the UCR 128 time series datasets was specifically structured to evaluate the strengths and limitations of these nine deep learning models in the context of supervised time series classification. Therefore, we adopt these nine deep learning methods as baseline models for our experimental analysis, as described below:

- FCN[1] (Wang et al., 2017) employs a three-layer one-dimensional fully convolutional network along with a one-layer global average pooling layer for time series classification.

- T-Loss[2] (Franceschi et al., 2019) utilizes temporal convolutional networks as its foundation and presents an innovative triplet loss for learning representations of time series in the time series classification task.

- SelfTime[3] (Fan et al., 2020) is a framework for time series representation learning that explores both inter-sample and intra-temporal relationships for time series classification tasks.

- TS-TCC[4] (Eldele et al., 2021) is a framework for learning time series representations through temporal and contextual contrasting for classification tasks.

- TST[5] (Zerveas et al., 2021) is a framework for multivariate time series representation learning based on the Transformer architecture for classification tasks.

- TS2Vec[6] (Yue et al., 2022) is a general framework for time series representation learning across various semantic levels, aimed at tasks such as time series classification, forecasting, and anomaly detection. In the context of the time series classification task, the authors utilized temporal convolutional networks (TCN) as the foundational framework, leveraging the low-dimensional feature representation derived from TCN as input for a support vector machine classifier utilizing a radial basis function kernel for the classification process.

---

[1]https://github.com/cauchyturing/UCR_Time_Series_Classification_Deep_Learning_Baseline
[2]https://github.com/White-Link/UnsupervisedScalableRepresentationLearningTimeSeries
[3]https://github.com/haoyfan/SelfTime
[4]https://github.com/emadeldeen24/TS-TCC
[5]https://github.com/gzerveas/mvts_transformer
[6]https://github.com/yuezhihan/ts2vec

Table 8: Details of the UCR 128 time series datasets. "Train" represents the count of samples within the raw training set. "Test" represents the count of samples within the raw test set. "Total" represents the overall count of samples within the time series dataset. "Class" indicates the number of classes present within each time series dataset. "Length" refers to the sequence length of the univariate time series within the corresponding dataset. The presence of "Vary" signifies that the dataset includes instances with missing values.

| ID | Name | # Train | # Test | # Total | # Class | # Length | ID | Name | # Train | # Test | # Total | # Class | # Length |
|----|------|---------|--------|---------|---------|----------|----|------|---------|--------|---------|---------|----------|
| 1 | ACSF1 | 100 | 100 | 200 | 10 | 1460 | 65 | ItalyPowerDemand | 67 | 1029 | 1096 | 2 | 24 |
| 2 | Adiac | 390 | 391 | 781 | 37 | 176 | 66 | LargeKitchenAppliances | 375 | 375 | 750 | 3 | 720 |
| 3 | AllGestureWiimoteX | 300 | 700 | 1000 | 10 | Vary | 67 | Lightning2 | 60 | 61 | 121 | 2 | 637 |
| 4 | AllGestureWiimoteY | 300 | 700 | 1000 | 10 | Vary | 68 | Lightning7 | 70 | 73 | 143 | 7 | 319 |
| 5 | AllGestureWiimoteZ | 300 | 700 | 1000 | 10 | Vary | 69 | Mallat | 55 | 2345 | 2400 | 8 | 1024 |
| 6 | ArrowHead | 36 | 175 | 211 | 3 | 251 | 70 | Meat | 60 | 60 | 120 | 3 | 448 |
| 7 | Beef | 30 | 30 | 60 | 5 | 470 | 71 | MedicalImages | 381 | 760 | 1141 | 10 | 99 |
| 8 | BeetleFly | 20 | 20 | 40 | 2 | 512 | 72 | MelbournePedestrian | 1194 | 2439 | 3633 | 10 | 24 |
| 9 | BirdChicken | 20 | 20 | 40 | 2 | 512 | 73 | MiddlePhalanxOutlineAgeGroup | 400 | 154 | 554 | 3 | 80 |
| 10 | BME | 30 | 150 | 180 | 3 | 128 | 74 | MiddlePhalanxOutlineCorrect | 600 | 291 | 891 | 2 | 80 |
| 11 | Car | 60 | 60 | 120 | 4 | 577 | 75 | MiddlePhalanxTW | 399 | 154 | 553 | 6 | 80 |
| 12 | CBF | 30 | 900 | 930 | 3 | 128 | 76 | MixedShapesRegularTrain | 500 | 2425 | 2925 | 5 | 1024 |
| 13 | Chinatown | 20 | 343 | 363 | 2 | 24 | 77 | MixedShapesSmallTrain | 100 | 2425 | 2525 | 5 | 1024 |
| 14 | ChlorineConcentration | 467 | 3840 | 4307 | 3 | 166 | 78 | MoteStrain | 20 | 1252 | 1272 | 2 | 84 |
| 15 | CinCECGTorso | 40 | 1380 | 1420 | 4 | 1639 | 79 | NonInvasiveFetalECGThorax1 | 1800 | 1965 | 3765 | 42 | 750 |
| 16 | Coffee | 28 | 28 | 56 | 2 | 286 | 80 | NonInvasiveFetalECGThorax2 | 1800 | 1965 | 3765 | 42 | 750 |
| 17 | Computers | 250 | 250 | 500 | 2 | 720 | 81 | OliveOil | 30 | 30 | 60 | 4 | 570 |
| 18 | CricketX | 390 | 390 | 780 | 12 | 300 | 82 | OSULeaf | 200 | 242 | 442 | 6 | 427 |
| 19 | CricketY | 390 | 390 | 780 | 12 | 300 | 83 | PhalangesOutlinesCorrect | 1800 | 858 | 2658 | 2 | 80 |
| 20 | CricketZ | 390 | 390 | 780 | 12 | 300 | 84 | Phoneme | 214 | 1896 | 2110 | 39 | 1024 |
| 21 | Crop | 7200 | 16800 | 24000 | 24 | 46 | 85 | PickupGestureWiimoteZ | 50 | 50 | 100 | 10 | Vary |
| 22 | DiatomSizeReduction | 16 | 306 | 322 | 4 | 345 | 86 | PigAirwayPressure | 104 | 208 | 312 | 52 | 2000 |
| 23 | DistalPhalanxOutlineAgeGroup | 400 | 139 | 539 | 3 | 80 | 87 | PigArtPressure | 104 | 208 | 312 | 52 | 2000 |
| 24 | DistalPhalanxOutlineCorrect | 600 | 276 | 876 | 2 | 80 | 88 | PigCVP | 104 | 208 | 312 | 52 | 2000 |
| 25 | DistalPhalanxTW | 400 | 139 | 539 | 6 | 80 | 89 | PLAID | 537 | 537 | 1074 | 11 | Vary |
| 26 | DodgerLoopDay | 78 | 80 | 158 | 7 | 288 | 90 | Plane | 105 | 105 | 210 | 7 | 144 |
| 27 | DodgerLoopGame | 20 | 138 | 158 | 2 | 288 | 91 | PowerCons | 180 | 180 | 360 | 2 | 144 |
| 28 | DodgerLoopWeekend | 20 | 138 | 158 | 2 | 288 | 92 | ProximalPhalanxOutlineAgeGroup | 400 | 205 | 605 | 3 | 80 |
| 29 | Earthquakes | 322 | 139 | 461 | 2 | 512 | 93 | ProximalPhalanxOutlineCorrect | 600 | 291 | 891 | 2 | 80 |
| 30 | ECG200 | 100 | 100 | 200 | 2 | 96 | 94 | ProximalPhalanxTW | 400 | 205 | 605 | 6 | 80 |
| 31 | ECG5000 | 500 | 4500 | 5000 | 5 | 140 | 95 | RefrigerationDevices | 375 | 375 | 750 | 3 | 720 |
| 32 | ECGFiveDays | 23 | 861 | 884 | 2 | 136 | 96 | Rock | 20 | 50 | 70 | 4 | 2844 |
| 33 | ElectricDevices | 8926 | 7711 | 16637 | 7 | 96 | 97 | ScreenType | 375 | 375 | 750 | 3 | 720 |
| 34 | EOGHorizontalSignal | 362 | 362 | 724 | 12 | 1250 | 98 | SemgHandGenderCh2 | 300 | 600 | 900 | 2 | 1500 |
| 35 | EOGVerticalSignal | 362 | 362 | 724 | 12 | 1250 | 99 | SemgHandMovementCh2 | 450 | 450 | 900 | 6 | 1500 |
| 36 | EthanolLevel | 504 | 500 | 1004 | 4 | 1751 | 100 | SemgHandSubjectCh2 | 450 | 450 | 900 | 5 | 1500 |
| 37 | FaceAll | 560 | 1690 | 2250 | 14 | 131 | 101 | ShakeGestureWiimoteZ | 50 | 50 | 100 | 10 | Vary |
| 38 | FaceFour | 24 | 88 | 112 | 4 | 350 | 102 | ShapeletSim | 20 | 180 | 200 | 2 | 500 |
| 39 | FacesUCR | 200 | 2050 | 2250 | 14 | 131 | 103 | ShapesAll | 600 | 600 | 1200 | 60 | 512 |
| 40 | FiftyWords | 450 | 455 | 905 | 50 | 270 | 104 | SmallKitchenAppliances | 375 | 375 | 750 | 3 | 720 |
| 41 | Fish | 175 | 175 | 350 | 7 | 463 | 105 | SmoothSubspace | 150 | 150 | 300 | 3 | 15 |
| 42 | FordA | 3601 | 1320 | 4921 | 2 | 500 | 106 | SonyAIBORobotSurface1 | 20 | 601 | 621 | 2 | 70 |
| 43 | FordB | 3636 | 810 | 4446 | 2 | 500 | 107 | SonyAIBORobotSurface2 | 27 | 953 | 980 | 2 | 65 |
| 44 | FreezerRegularTrain | 150 | 2850 | 3000 | 2 | 301 | 108 | StarLightCurves | 1000 | 8236 | 9236 | 3 | 1024 |
| 45 | FreezerSmallTrain | 28 | 2850 | 2878 | 2 | 301 | 109 | Strawberry | 613 | 370 | 983 | 2 | 235 |
| 46 | Fungi | 18 | 186 | 204 | 18 | 201 | 110 | SwedishLeaf | 500 | 625 | 1125 | 15 | 128 |
| 47 | GestureMidAirD1 | 208 | 130 | 338 | 26 | Vary | 111 | Symbols | 25 | 995 | 1020 | 6 | 398 |
| 48 | GestureMidAirD2 | 208 | 130 | 338 | 26 | Vary | 112 | SyntheticControl | 300 | 300 | 600 | 6 | 60 |
| 49 | GestureMidAirD3 | 208 | 130 | 338 | 26 | Vary | 113 | ToeSegmentation1 | 40 | 228 | 268 | 2 | 277 |
| 50 | GesturePebbleZ1 | 132 | 172 | 304 | 6 | Vary | 114 | ToeSegmentation2 | 36 | 130 | 166 | 2 | 343 |
| 51 | GesturePebbleZ2 | 146 | 158 | 304 | 6 | Vary | 115 | Trace | 100 | 100 | 200 | 4 | 275 |
| 52 | GunPoint | 50 | 150 | 200 | 2 | 150 | 116 | TwoLeadECG | 23 | 1139 | 1162 | 2 | 82 |
| 53 | GunPointAgeSpan | 135 | 316 | 451 | 2 | 150 | 117 | TwoPatterns | 1000 | 4000 | 5000 | 4 | 128 |
| 54 | GunPointMaleVersusFemale | 135 | 316 | 451 | 2 | 150 | 118 | UMD | 36 | 144 | 180 | 3 | 150 |
| 55 | GunPointOldVersusYoung | 136 | 315 | 451 | 2 | 150 | 119 | UWaveGestureLibraryAll | 896 | 3582 | 4478 | 8 | 945 |
| 56 | Ham | 109 | 105 | 214 | 2 | 431 | 120 | UWaveGestureLibraryX | 896 | 3582 | 4478 | 8 | 315 |
| 57 | HandOutlines | 1000 | 370 | 1370 | 2 | 2709 | 121 | UWaveGestureLibraryY | 896 | 3582 | 4478 | 8 | 315 |
| 58 | Haptics | 155 | 308 | 463 | 5 | 1092 | 122 | UWaveGestureLibraryZ | 896 | 3582 | 4478 | 8 | 315 |
| 59 | Herring | 64 | 64 | 128 | 2 | 512 | 123 | Wafer | 1000 | 6164 | 7164 | 2 | 152 |
| 60 | HouseTwenty | 40 | 119 | 159 | 2 | 2000 | 124 | Wine | 57 | 54 | 111 | 2 | 234 |
| 61 | InlineSkate | 100 | 550 | 650 | 7 | 1882 | 125 | WordSynonyms | 267 | 638 | 905 | 25 | 270 |
| 62 | InsectEPGRegularTrain | 62 | 249 | 311 | 3 | 601 | 126 | Worms | 181 | 77 | 258 | 5 | 900 |
| 63 | InsectEPGSmallTrain | 17 | 249 | 266 | 3 | 601 | 127 | WormsTwoClass | 181 | 77 | 258 | 2 | 900 |
| 64 | InsectWingbeatSound | 220 | 1980 | 2200 | 11 | 256 | 128 | Yoga | 300 | 3000 | 3300 | 2 | 426 |

- TimesNet[7] (Wu et al., 2023b) employs TimesBlock to identify multiple periods and capture temporal 2D variations from transformed 2D tensors using a parameter-efficient inception block for time series modeling.

---

[7] https://github.com/thuml/TimesNet

- PatchTST[8] (Nie et al., 2023) segment time series into patches while assuming that channels are independent for multivariate time series forecasting. Considering that GPT4TS (Zhou et al., 2023) has utilised similar patch and channel-independent strategies for time series modeling, we employ only the patch strategy and the channel-independent approach by integrating a transformer for time series classification.

- GPT4TS[9] (Zhou et al., 2023) is a unified framework for time-series modeling that utilises pre-trained large language models. Specifically, GPT4TS adopts the same patch and channel-independent strategies used in PatchTST (Nie et al., 2023) for time series classification tasks.

Furthermore, Middlehurst et al. (2024) conduct a comprehensive analysis of the advantages and limitations of eight categories of time series classification methods: distance-based, feature-based, interval-based, shapelet-based, dictionary-based, convolution-based, deep learning-based, and hybrid-based approaches. However, in their experimental evaluation, Middlehurst et al. (2024) utilize only a training set and a test set, without incorporating a validation set for deep learning-based methods performance assessment.

Specifically, for the selected deep learning-based time series classification methods, Middlehurst et al. (2024) employ the training epoch with the lowest training loss to generate test results. This approach increases the risk of overfitting, as the deep learning model is optimized solely on training data, potentially compromising its generalization capability. Nonetheless, the findings of (Middlehurst et al., 2024) indicate that the deep learning-based InceptionTime (Ismail Fawaz et al., 2020) model achieves superior classification performance. Additionally, experimental results (Middlehurst et al., 2024) demonstrate that HIVE-COTE version 2 (HC2) (Middlehurst et al., 2021), which is a hybrid-based method, MultiROCKET-Hydra (MR-H) (Dempster et al., 2023), a convolution-based method, and RDST, a shapelet-based approach, exhibit strong classification performance.

Given that HC2 is computationally intensive, we select the deep learning-based InceptionTime model along with two non-deep learning methods, MR-H and RDST, as baseline models for our study. The details of these baseline methods are outlined as follows:

- Random Dilated Shapelet Transform (RDST)[10] (Guillaume et al., 2022) is a shapelet-based algorithm that adopts many of the techniques of randomly parameterised convolutional kernels to obtain shapelet features for time series classification.

- MultiROCKET-Hydra (MR-H)[11] (Dempster et al., 2023) is a model that combines dictionary-based and convolution-based models, using competing convolutional kernels and combining key aspects of both Rocket and conventional dictionary methods for time series classification.

- InceptionTime[12] (Ismail Fawaz et al., 2020) is a deep convolutional neural network-based model for time series classification, drawing inspiration from the Inception-v4 architecture.

In recent years, scholars have introduced deep learning models based on convolutional neural networks, transformer architectures, and foundation models for time series classification tasks. For our experimental analysis, we have selected seven recently published methods as baseline models, which are detailed as follows:

- LightTS[13] (Campos et al., 2023) employs model distillation to reduce runtime by ensembling multiple classifiers. It consists of two stages: a teacher phase, where at least ten models (e.g., InceptionTime) are trained, and a student phase that distills their knowledge. Due to the time-intensive nature of the teacher phase, evaluation is limited to 18 selected UCR datasets. Moreover, since LightTS (student) underperforms compared to LightTS (teacher), only the latter's results are reported.

---

[8]https://github.com/yuqinie98/PatchTST

[9]https://github.com/DAMO-DI-ML/NeurIPS2023-One-Fits-All

[10]https://github.com/aeon-toolkit/aeon/blob/main/aeon/classification/shapelet_based/_rdst.py

[11]https://github.com/aeon-toolkit/aeon/blob/main/aeon/classification/convolution_based/_mr_hydra.py

[12]https://github.com/timeseriesAI/tsai/blob/main/tsai/models/InceptionTime.py

[13]https://github.com/d-gcc/LightTS

- ShapeConv[14] (Qu et al., 2024) introduces an interpretable convolutional layer for time series analysis, where the convolutional kernels are designed to act as shapelets. This convolutional design is inspired by the Shapelet distance, a widely adopted approach in time series classification.

- Shapeformer[15] (Le et al., 2024) is a Shapelet Transformer that combines class-specific and generic transformer modules to effectively capture both types of features. Due to the time-consuming shapelet discovery process in Shapeformer, we evaluate its performance on 18 selected UCR datasets

- ModernTCN[16] (Luo & Wang, 2024) is a pure convolution architecture for time series modeling, which utilizes depthwise convolution and grouped pointwise convolution to learn cross-variable and cross-feature information in a decoupled way.

- TSLANet[17] (Eldele et al., 2024) is a time series lightweight adaptive metwork, as a universal convolutional model by combining an adaptive spectral block and interactive convolution block using patch-based technologies for time series modeling.

- UniTS[18] (Gao et al., 2024) is a unified pre-trained foundation model for time series, designed to handle various forecasting and classification tasks across diverse datasets. Following the experiments in Table 14 of UniTS, we use the UNITS-SUP×64 model as the baseline.

- Medformer[19] (Wang et al., 2024) is a transformer model designed for medical time series classification that utilizes cross-channel multi-granularity patching, and intra-inter granularity self-attention within and among granularities.

### A.3. Implementation Details

We employ the Adam optimizer with a learning rate of 0.001 and a maximum training epoch of 500. Also, for all baseline method training, we implement a consistent early stopping strategy based on validation set loss values (Ma et al., 2024). For all UCR datasets, we apply a uniform normalization strategy to standardize each time series within the dataset (Ismail Fawaz et al., 2019). For datasets containing missing values, we use the mean of observed values at each specific timestamp across all samples in the training set to impute missing values at corresponding timestamps within the time series samples (Ma et al., 2024). Following (Wang et al., 2017), we set the batch size for model training as $\min(x_{\text{train}}.\text{shape}[0]/10, 16)$, where $x_{\text{train}}.\text{shape}[0]$ represents the number of time series samples in the training set, and $\min(\cdot)$ denotes the function that selects the minimum value.

From Equation (1) in the main text, the length of time series shapelets, $m$, satisfies $2 \le m \le T - 1$. Directly using subsequences of varying lengths as model inputs significantly increases computational complexity and requires additional preprocessing to standardize input dimensions. To address this, Grabocka et al. (2014); Qu et al. (2024) employ a validation set to select a fixed-length $m$ of subsequence for shapelet learning, where $m$ is defined as $\gamma T$, with possible values $\{0.1T, 0.2T, ..., 0.8T\}$. This approach reduces the total number of candidate subsequences in Equation (1), thereby lowering computational costs. For patch-based time series classification (Nie et al., 2023; Wang et al., 2024), patch lengths are typically chosen from $\{4, 8, 16, 24, 32, 48, 64\}$. Our preliminary experiments indicate that setting shape length $m$ within this range yields better classification performance. However, due to variations in sequence lengths across UCR 128 datasets, a fixed-length setting may be problematic. For datasets with long sequences, setting $m = 4$ results in excessive candidate subsequences, while for short sequences, setting $m = 64$ may eliminate viable subsequences. To balance these issues, we select $m$ from $\{4, 8, 16, 24, 32, 48, 64, 0.1T, 0.8T\}$ using the validation set.

Additionally, we set the model depth $L$ to 2. The number of activated experts $k$ used for the Mixture of Experts (MoE) router in Equation (6) is set to 1, while the total number of experts $\hat{C}$ in MoE is defined as the total number of classes in each dataset. The parameter $\lambda$, which regulates the learning progression of the loss term in Equation (15), is set to 0.001. The sliding window size $q$ for extracting subsequences is set to 4. The $d_{attn}$ in Equation (3) is set to 8 (Early et al., 2024). The rate $\eta$ in Equation (5) is set to 50%. To mitigate the issue of inaccurate attention score evaluations for most shapes

---

[14]https://openreview.net/attachment?id=O8ouVV8PjF&name=supplementary_material

[15]https://github.com/xuanmay2701/shapeformer

[16]https://github.com/luodhhh/ModernTCN

[17]https://github.com/emadeldeen24/TSLANet

[18]https://github.com/mims-harvard/UniTS

[19]https://github.com/dl4mhealth/medformer

during the initial training phase, the model is warm-up trained for 150 epochs before initiating the soft shape sparsification process. For each UCR time series, we calculate the average test accuracy using five-fold test sets from a single seed. To ensure a fair comparison, we maintain a consistent random seed, as well as standardized dataset preprocessing and partitioning for all baseline methods. For SoftShape and deep learning-based baselines, the same optimizer and learning rate are applied during training, selecting the model with the lowest validation loss for test evaluation. Non-deep learning methods are trained on the training set and evaluated on the test set. Furthermore, for deep learning baselines, except LightTS (Campos et al., 2023), a single model instance is employed without ensembling multiple models initialized differently during both training and evaluation. Like (Ismail Fawaz et al., 2019; Ma et al., 2024), we evaluate performance based on the classification accuracy of the test set. Each experiment is conducted five times with five different random seeds, utilizing the PyTorch 1.12.1 platform and four NVIDIA GeForce RTX 3090 GPUs. The source code for SoftShape is available at https://github.com/qianlima-lab/SoftShape.

---

**Algorithm 1** The proposed SoftShape model.

---

**Input:** shape embedding layer $w_{shape}$, attention head $w_{attn}$, soft shape learning block $w_{block}$, a classifier $w_c$, sparse ratio $\eta$
**Output:** $w_{shape}, w_{attn}, w_{block}$, and $w_c$.

1: **Step one:** Obtain all shape embeddings $\hat{\mathcal{S}}_n^m$ through $w_{shape}$ via Eq. (2);
   Using a norm layer to process $\hat{\mathcal{S}}_n^m$, denoted as $norm(\hat{\mathcal{S}}_n^m)$;
2: **Step two:** Soft shape sparsification using $norm(\hat{\mathcal{S}}_n^m)$ as inputs;
   Calculate the attention score for all shapes via the $w_{attn}$;
   Convert scores in the top $\eta$ proportion of all shapes into soft shape embeddings via Eq. (4);
   Fuse those with scores below the top $\eta$ proportion into a single shape embedding via Eq. (5);
   Combine soft shape embeddings and the fused single shape embedding, denoted as $\widetilde{\mathcal{S}}_n^m$;
3: **Step three:** Intra-Shape and Inter-Shape Learning through $w_{block}$ using $\widetilde{\mathcal{S}}_n^m$ as inputs;
   Using a norm layer to process $\widetilde{\mathcal{S}}_n^m$, denoted as $norm(\widetilde{\mathcal{S}}_n^m)$;
   Obtain Intra-Shape Embedings via $w_{block}$ using Eq. (8), denoted as $\widetilde{\mathcal{S}}_{Intra}$;
   Obtain Inter-Shape Embedings via $w_{block}$ using a shared expert, denoted as $\widetilde{\mathcal{S}}_{Inter}$;
   Combine $\widetilde{\mathcal{S}}_n^m, \widetilde{\mathcal{S}}_{Intra}$, and $\widetilde{\mathcal{S}}_{Inter}$ in a residul way, denoted as $\widetilde{\mathcal{S}}_{out} = \widetilde{\mathcal{S}}_n^m + \widetilde{\mathcal{S}}_{Intra} + \widetilde{\mathcal{S}}_{Inter}$;
4: **Step four:** Set model depth to L for training by rerunning Step two and Step three $L$ times using $\widetilde{\mathcal{S}}_{out}$ as inputs, denoted the end outputs as $O_n^m$;
5: **Step five:** Classification training using $O_n^m$ as inputs;
   Update $w_{shape}, w_{attn}, w_{block}, w_c$ via training objective $\mathcal{L}_{total}$ using Eq. (15);

---

# B. Details of Experimental Results

## B.1. Main Results

The test classification accuracy results of SoftShape and 17 baseline methods on the UCR 128 time series dataset are presented in Table 9. Specifically, the test classification accuracy results of nine methods, including FCN, T-Loss, SelfTime, TS-TCC, TST, TS2Vec, TimesNet, PatchTST, and GPT4TS, are obtained from (Ma et al., 2024). Figure 6 illustrates the critical difference diagram and significance analysis results for SoftShape and the 17 baseline methods on the UCR 128 time series dataset. Additionally, Table 10 presents the detailed test classification accuracies of LightTS, Shapeformer, and SoftShape on the selected 18 UCR datasets.

Table 9: The detailed test classification accuracy comparisons on 128 UCR time series datasets. Among these, *Incep* refers to the InceptionTime method, and *MoTCN* refers to the ModernTCN method. The best results are in **bold**.

| ID | Dataset | FCN | T-Loss | Selftime | TS-TCC | TST | TS2Vec | TimesNet | PatchTST | GPT4TS | RDST | MR-H | Incep | ShapeConv | MoTCN | TSLANet | UniTS | Medformer | SoftShape |
|---|---|---|---|---|---|---|---|---|---|---|---|---|---|---|---|---|---|---|---|
| 1 | ACSF1 | 0.6250 | 0.8050 | 0.7250 | 0.5700 | 0.7050 | 0.8800 | 0.6000 | 0.7600 | 0.6200 | 0.8400 | 0.8450 | 0.9050 | 0.6000 | 0.4500 | 0.8700 | 0.8300 | 0.8350 | **0.9100** |
| 2 | Adiac | 0.6492 | 0.7811 | 0.7322 | 0.4003 | 0.7363 | 0.8066 | 0.8082 | 0.8358 | 0.8069 | 0.7848 | 0.8527 | 0.9354 | 0.5635 | 0.8580 | 0.8837 | 0.6596 | 0.8364 | **0.9374** |
| 3 | AllGestureWX. | 0.7110 | 0.7170 | 0.6461 | 0.7201 | 0.5280 | 0.7820 | 0.5120 | 0.5730 | 0.7220 | 0.7930 | 0.8050 | 0.8540 | 0.3170 | 0.7260 | 0.8970 | 0.8530 | 0.6130 | **0.8980** |
| 4 | AllGestureWY. | 0.6740 | 0.8110 | 0.7170 | 0.7615 | 0.3850 | 0.8400 | 0.5820 | 0.6730 | 0.7800 | 0.8360 | 0.8770 | 0.9140 | 0.2990 | 0.7470 | **0.9180** | 0.8690 | 0.8250 | 0.9110 |
| 5 | AllGestureWZ. | 0.7200 | 0.7470 | 0.6485 | 0.6567 | 0.4780 | 0.7730 | 0.6620 | 0.5890 | 0.7300 | 0.7850 | 0.8100 | 0.8920 | 0.3070 | 0.6540 | **0.9010** | 0.8460 | 0.7250 | 0.8850 |
| 6 | ArrowHead | 0.8958 | 0.8770 | 0.8155 | 0.8200 | 0.8722 | 0.8908 | 0.7821 | 0.7078 | 0.8348 | 0.9382 | 0.9430 | 0.9288 | 0.6638 | 0.8152 | **0.9672** | 0.9433 | 0.9434 | 0.9435 |
| 7 | Beef | 0.8250 | 0.8750 | 0.5167 | 0.6333 | 0.6667 | 0.6833 | 0.8000 | 0.4167 | **0.9333** | 0.8333 | 0.8500 | 0.7833 | 0.6833 | 0.8500 | 0.7500 | 0.7167 | 0.8333 | 0.8667 |
| 8 | BeetleFly | 0.9250 | **0.9750** | 0.8000 | 0.6250 | 0.6750 | 0.9250 | 0.7000 | 0.8500 | 0.7250 | 0.9000 | 0.9250 | 0.8500 | 0.9250 | 0.7500 | **0.9750** | 0.8750 | 0.9000 | **0.9750** |
| 9 | BirdChicken | 0.7833 | **0.9833** | 0.9000 | 0.6750 | 0.8500 | 0.9750 | 0.8250 | 0.8500 | 0.7500 | 0.9000 | 0.9000 | 0.9500 | 0.9500 | 0.8250 | 0.8750 | 0.8250 | 0.8250 | 0.9000 |
| 10 | BME | 0.6000 | 0.5333 | 0.9722 | 0.9889 | 0.9722 | 0.9944 | 0.9778 | 0.9500 | 0.9611 | **1.0000** | **1.0000** | **1.0000** | 0.6722 | 0.9833 | **1.0000** | 0.9889 | **1.0000** | 0.9944 |

**Learning Soft Sparse Shapes for Efficient Time-Series Classification**

| ID | Dataset | FCN | T-Loss | Selftime | TS-TCC | TST | TS2Vec | TimesNet | PatchTST | GPT4TS | RDST | MR-H | Incep | ShapeConv | MoTCN | TSLANet | UniTS | Medformer | SoftShape |
|----|---------|-----|--------|----------|--------|-----|--------|----------|----------|--------|------|------|-------|-----------|-------|---------|-------|-----------|-----------|
| 11 | Car | **1.0000** | **1.0000** | 0.6583 | 0.7167 | 0.7583 | 0.8833 | 0.8417 | 0.9083 | 0.9000 | 0.9500 | 0.9583 | 0.9583 | 0.7750 | 0.8250 | 0.9500 | 0.9083 | 0.8417 | 0.9583 |
| 12 | CBF | 0.9417 | 0.8250 | 0.9936 | 0.9967 | 0.9946 | **1.0000** | **1.0000** | **1.0000** | 0.9978 | **1.0000** | **1.0000** | **1.0000** | 0.9989 | **1.0000** | **1.0000** | 0.9978 | **1.0000** | **1.0000** |
| 13 | Chinatown | 0.9753 | 0.9807 | 0.9669 | 0.9507 | 0.9699 | 0.9726 | 0.9863 | 0.9836 | 0.9863 | 0.9699 | 0.9808 | 0.9836 | 0.9123 | 0.9233 | 0.9863 | 0.9781 | 0.9863 | **0.9890** |
| 14 | ChlorineCon. | 0.9984 | 0.9988 | 0.6116 | 0.8424 | 0.9974 | **0.9998** | 0.9993 | 0.9995 | **0.9998** | 0.9886 | 0.9954 | 0.9993 | 0.6973 | 0.9991 | 0.9988 | 0.9988 | 0.6427 | 0.9988 |
| 15 | CinCECGTorso | 0.9986 | 0.9944 | 0.9877 | 0.9995 | 0.9993 | 0.9972 | 0.9930 | **1.0000** | 0.9930 | **1.0000** | **1.0000** | 0.9923 | 0.9789 | 0.9993 | **1.0000** | 0.9958 | 0.9986 | **1.0000** |
| 16 | Coffee | **1.0000** | 0.9818 | 0.9273 | 0.9455 | 0.8758 | **1.0000** | **1.0000** | **1.0000** | **1.0000** | **1.0000** | **1.0000** | **1.0000** | 0.9667 | **1.0000** | **1.0000** | **1.0000** | 0.9818 | **1.0000** |
| 17 | Computers | 0.8800 | 0.7140 | 0.7920 | 0.6080 | 0.6880 | 0.6940 | 0.8120 | 0.8380 | 0.8080 | 0.8180 | 0.8540 | 0.8420 | 0.6660 | 0.6540 | **0.8920** | 0.8120 | 0.8280 | 0.8160 |
| 18 | CricketX | 0.9064 | 0.7795 | 0.7316 | 0.7327 | 0.7026 | 0.8321 | 0.7974 | 0.7821 | 0.7910 | 0.8628 | 0.8628 | 0.9013 | 0.8179 | 0.7231 | 0.9205 | 0.8923 | 0.8397 | **0.9321** |
| 19 | CricketY | 0.9000 | 0.7487 | 0.6904 | 0.7161 | 0.6962 | 0.8256 | 0.7808 | 0.7821 | 0.8026 | 0.8436 | 0.8487 | 0.8923 | 0.7692 | 0.7513 | 0.8987 | 0.8808 | 0.8500 | **0.9154** |
| 20 | CricketZ | 0.8833 | 0.7833 | 0.7411 | 0.7384 | 0.6936 | 0.8487 | 0.7962 | 0.7923 | 0.8026 | 0.8692 | 0.8526 | 0.8693 | 0.7744 | 0.7385 | 0.9064 | 0.8859 | 0.8564 | **0.9372** |
| 21 | Crop | 0.1312 | 0.7063 | 0.6634 | 0.7801 | 0.7709 | 0.7448 | 0.8503 | 0.1193 | 0.8764 | 0.7675 | 0.7805 | 0.8757 | 0.1833 | 0.0922 | 0.5781 | 0.0417 | 0.3043 | **0.8765** |
| 22 | DiatomSizeRe. | 0.9969 | 0.9969 | 0.9723 | 0.9783 | **1.0000** | 0.9969 | 0.9938 | **1.0000** | 0.9907 | **1.0000** | **1.0000** | 0.9969 | 0.9531 | 0.9877 | 0.9969 | 0.9719 | 0.9938 | **1.0000** |
| 23 | DistalPhalanxOut. | 0.9091 | 0.8274 | 0.7959 | 0.8200 | 0.8404 | 0.8125 | 0.8590 | 0.8779 | 0.8890 | 0.8311 | 0.8329 | 0.8850 | 0.7662 | 0.8089 | 0.9333 | 0.8070 | 0.8312 | **0.9518** |
| 24 | DistalPhalanxOut. | 0.8917 | 0.8471 | 0.7895 | 0.7646 | 0.8254 | 0.8185 | 0.8860 | 0.9009 | 0.8929 | 0.8323 | 0.8448 | 0.8802 | 0.7980 | 0.7752 | 0.9018 | 0.7363 | 0.6529 | **0.9054** |
| 25 | DistalPhalanxTW | 0.8423 | 0.7421 | 0.7514 | 0.7755 | 0.7551 | 0.7792 | 0.8499 | 0.8485 | 0.8263 | 0.7866 | 0.7700 | 0.8500 | 0.7421 | 0.7979 | 0.8796 | 0.7718 | 0.8349 | **0.9037** |
| 26 | DodgerLoopDay | 0.7486 | 0.4819 | 0.4742 | 0.5885 | 0.5377 | 0.6391 | 0.7107 | 0.6694 | 0.7403 | 0.6266 | 0.5315 | 0.7681 | 0.7871 | 0.5528 | 0.8244 | **0.8750** | 0.8621 | 0.8623 |
| 27 | DodgerLoopGame | 0.8167 | 0.8921 | 0.8282 | 0.8857 | 0.8988 | 0.9563 | 0.8484 | 0.8872 | 0.9001 | 0.9179 | 0.9175 | **0.9621** | 0.9373 | 0.7966 | 0.9563 | 0.8484 | 0.9242 | 0.9433 |
| 28 | DodgerLoopWeek. | 0.9812 | 0.9677 | 0.9681 | 0.9742 | 0.9679 | 0.9812 | 0.9688 | 0.9492 | 0.9165 | 0.9746 | 0.9873 | 0.9806 | 0.9937 | 0.9937 | 0.9760 | 0.9806 | **1.0000** | **1.0000** |
| 29 | Earthquakes | 0.7376 | 0.7417 | 0.7983 | 0.7505 | 0.7961 | 0.7983 | 0.9136 | 0.9115 | 0.9052 | 0.7809 | 0.7939 | 0.7939 | 0.8526 | 0.7333 | **0.9222** | 0.8613 | 0.8963 | 0.8917 |
| 30 | ECG200 | 0.8482 | 0.7983 | 0.7900 | 0.7950 | 0.8600 | 0.8800 | 0.9100 | 0.8763 | 0.8951 | 0.9200 | 0.9250 | 0.9250 | 0.8600 | 0.9050 | 0.9450 | 0.8850 | 0.9100 | **0.9475** |
| 31 | ECG5000 | 0.9350 | 0.8500 | 0.9317 | 0.9525 | 0.9502 | 0.9528 | 0.9708 | 0.9364 | 0.9746 | 0.9552 | 0.9566 | 0.9710 | 0.9432 | 0.5510 | 0.9760 | 0.9622 | 0.9460 | **0.9796** |
| 32 | ECGFiveDays | 0.9258 | 0.9478 | 0.9873 | 0.9972 | 0.9977 | **1.0000** | **1.0000** | **1.0000** | 0.9978 | **1.0000** | **1.0000** | **1.0000** | 0.9636 | **1.0000** | **1.0000** | **1.0000** | **1.0000** | **1.0000** |
| 33 | ElectricDevices | 0.6519 | 0.6547 | 0.7173 | 0.8639 | 0.8758 | 0.8848 | 0.8610 | 0.4883 | 0.8610 | 0.9024 | 0.9094 | **0.9459** | 0.7259 | 0.3500 | 0.8991 | 0.7921 | 0.7665 | 0.9208 |
| 34 | EOGHorizontalS. | **1.0000** | **1.0000** | 0.7579 | 0.6295 | 0.6781 | 0.7279 | 0.6092 | 0.8010 | 0.7581 | 0.7942 | 0.8453 | 0.8200 | 0.7056 | 0.7281 | 0.8993 | 0.8469 | 0.8344 | 0.9007 |
| 35 | EOGVerticalS. | 0.6731 | 0.8736 | 0.6774 | 0.6190 | 0.4806 | 0.6519 | 0.6369 | 0.8013 | 0.7804 | 0.7335 | 0.7984 | 0.8703 | 0.4764 | 0.6922 | **0.8965** | 0.8620 | 0.8220 | 0.8814 |
| 36 | EthanolLevel | 0.8307 | 0.5627 | 0.4864 | 0.2985 | 0.7919 | 0.5897 | 0.7690 | 0.7248 | 0.8337 | 0.7480 | 0.7261 | 0.9144 | 0.2929 | **0.9522** | 0.9184 | 0.4771 | 0.8019 | 0.9204 |
| 37 | FaceAll | 0.9462 | 0.9720 | 0.9552 | 0.9914 | 0.9720 | 0.9871 | 0.9733 | 0.8057 | 0.9877 | 0.9951 | 0.9951 | 0.9973 | 0.9324 | 0.9947 | 0.9880 | 0.9778 | 0.9778 | **0.9978** |
| 38 | FaceFour | 0.9557 | 0.9281 | 0.8830 | 0.9379 | 0.9375 | 0.9640 | 0.9557 | 0.9068 | 0.9826 | **1.0000** | 0.9909 | **1.0000** | **1.0000** | 0.9826 | 0.9913 | 0.9735 | 0.9826 | 0.9909 |
| 39 | FacesUCR | 0.9969 | 0.9729 | 0.9544 | 0.9899 | 0.9702 | 0.9902 | 0.9800 | 0.9824 | 0.9796 | 0.9938 | 0.9938 | 0.9973 | 0.9653 | 0.9840 | 0.9960 | 0.9889 | 0.9831 | 0.9956 |
| 40 | FiftyWords | 0.5072 | 0.8088 | 0.6705 | 0.7628 | 0.7381 | 0.8022 | 0.8276 | 0.8417 | 0.7709 | 0.8619 | 0.8718 | 0.9315 | 0.8674 | 0.8718 | 0.9249 | 0.8807 | 0.8685 | **0.9359** |
| 41 | Fish | 0.9686 | 0.8743 | 0.8514 | 0.8114 | 0.8628 | 0.9343 | 0.8000 | 0.9343 | 0.9143 | 0.9857 | 0.9829 | **0.9943** | 0.8371 | 0.8800 | 0.9629 | 0.9400 | 0.9086 | 0.9771 |
| 42 | FordA | 0.9734 | 0.9114 | 0.8787 | 0.9342 | 0.8990 | 0.9303 | 0.9348 | 0.9500 | 0.9403 | 0.9484 | 0.9537 | 0.9728 | 0.8935 | 0.9590 | 0.9740 | 0.9718 | 0.8450 | **0.9772** |
| 43 | FordB | 0.9312 | 0.9035 | 0.8771 | 0.9087 | 0.8650 | 0.9132 | 0.9046 | 0.9377 | 0.9287 | 0.9327 | 0.9345 | 0.9705 | 0.8934 | 0.9476 | 0.9658 | 0.9328 | 0.8498 | **0.9744** |
| 44 | FreezerRegular. | 0.9790 | 0.9980 | 0.9974 | 0.9969 | 0.9987 | 0.9977 | 0.9990 | 0.9393 | 0.9987 | 0.9993 | **0.9997** | 0.9993 | 0.7590 | 0.6967 | 0.9993 | 0.8100 | 0.6950 | 0.9990 |
| 45 | FreezerSmall. | 0.8359 | 0.9965 | 0.9975 | 0.9966 | 0.9979 | 0.9969 | 0.9969 | 0.8492 | 0.9983 | 0.9993 | **0.9997** | 0.9986 | 0.7575 | 0.5209 | **0.9997** | 0.9396 | 0.8351 | 0.9990 |
| 46 | Fungi | 0.9118 | **1.0000** | 0.9316 | 0.9557 | 0.9902 | 0.9649 | 0.8337 | 0.7276 | 0.9049 | **1.0000** | **1.0000** | **1.0000** | 0.9951 | 0.9415 | **1.0000** | **1.0000** | 0.9902 | **1.0000** |
| 47 | GestureMidAirD1 | 0.6954 | 0.5919 | 0.7067 | 0.6120 | 0.6181 | 0.6479 | 0.6099 | 0.6103 | 0.6449 | 0.7452 | 0.7632 | 0.7786 | 0.6810 | 0.7074 | 0.8493 | 0.8051 | 0.7813 | **0.8640** |
| 48 | GestureMidAirD2 | 0.5860 | 0.4973 | 0.5504 | 0.4853 | 0.5475 | 0.5270 | 0.5449 | 0.5692 | 0.5410 | 0.5859 | 0.5858 | 0.7606 | 0.5481 | 0.6308 | 0.7640 | 0.7287 | 0.7079 | **0.7962** |
| 49 | GestureMidAirD3 | 0.3819 | 0.3108 | 0.3664 | 0.3137 | 0.3462 | 0.3315 | 0.4559 | 0.4962 | 0.5282 | 0.4144 | 0.4291 | 0.6104 | 0.5364 | 0.5399 | 0.6577 | 0.6343 | 0.6094 | **0.7613** |
| 50 | GesturePebbleZ1 | 0.9541 | 0.6420 | 0.9043 | 0.8977 | 0.7761 | 0.9507 | 0.9541 | 0.8316 | 0.9158 | 0.9769 | 0.9703 | 0.9836 | 0.8289 | 0.9177 | **0.9967** | 0.9573 | 0.9572 | 0.9867 |
| 51 | GesturePebbleZ2 | 0.9343 | 0.6614 | 0.8817 | 0.8914 | 0.8159 | 0.9473 | 0.8850 | 0.7800 | 0.9382 | 0.9606 | 0.9540 | **0.9836** | 0.8127 | 0.9475 | 0.9738 | 0.9606 | 0.9409 | 0.9836 |
| 52 | GunPoint | **1.0000** | 0.9850 | 0.9550 | 0.9500 | 0.9700 | 0.9950 | 0.9850 | 0.9800 | 0.9800 | **1.0000** | **1.0000** | **1.0000** | 0.9350 | 0.9600 | 0.9900 | 0.9400 | 0.9800 | 0.9900 |
| 53 | GunPointAgeSpan | 0.9956 | 0.9756 | 0.9800 | 0.9578 | 0.9778 | 0.9889 | 0.9823 | 0.9889 | 0.9890 | 0.9956 | **1.0000** | 0.9889 | 0.9157 | 0.8024 | 0.9890 | 0.9179 | 0.9911 | 0.9868 |
| 54 | GunPointMale. | 0.9978 | 0.9911 | 0.9845 | 0.9823 | 0.9867 | 0.9956 | 0.9956 | **1.0000** | 0.9956 | **1.0000** | **1.0000** | 0.9933 | 0.9712 | 0.9845 | 0.9934 | 0.9845 | 0.9934 | 0.9956 |
| 55 | GunPointOld. | 0.9934 | 0.9468 | 0.9490 | 0.9512 | 0.9667 | 0.9956 | 0.9800 | 0.9912 | 0.9691 | 0.9956 | **0.9978** | 0.9712 | 0.7653 | 0.9180 | 0.9823 | 0.9623 | 0.9712 | 0.9779 |
| 56 | Ham | 0.6632 | 0.8080 | 0.7193 | 0.8275 | 0.8450 | 0.8741 | 0.9023 | 0.5420 | **0.9349** | 0.9064 | 0.8740 | 0.8461 | 0.6783 | 0.8458 | 0.8972 | 0.8788 | 0.9207 | 0.9161 |
| 57 | HandOutlines | 0.8956 | 0.9073 | 0.8425 | 0.8657 | 0.7170 | 0.9168 | 0.9029 | 0.9350 | 0.8956 | 0.9482 | 0.9482 | 0.9606 | 0.8934 | **0.9701** | 0.9460 | 0.9336 | 0.9131 | 0.9533 |
| 58 | Haptics | 0.6158 | 0.4879 | 0.4730 | 0.4990 | 0.4794 | 0.5594 | 0.6422 | 0.6672 | 0.6842 | 0.6241 | 0.5983 | 0.6969 | 0.5033 | 0.5862 | 0.6961 | 0.4899 | 0.6355 | **0.7522** |
| 59 | Herring | 0.6406 | 0.5929 | 0.6086 | 0.5452 | 0.6400 | 0.6806 | 0.6560 | 0.7604 | 0.8003 | 0.6868 | 0.6554 | 0.7185 | 0.6649 | 0.5791 | **0.8212** | 0.6554 | 0.7046 | 0.7671 |
| 60 | HouseTwenty | **0.9875** | 0.8935 | 0.9746 | 0.8244 | 0.7673 | 0.9496 | 0.8615 | 0.8617 | 0.8683 | 0.9750 | **0.9875** | 0.9500 | 0.9623 | 0.7429 | 0.9500 | 0.9121 | 0.9496 | 0.9688 |
| 61 | InlineSkate | 0.7046 | 0.6354 | 0.1922 | 0.4172 | 0.4754 | 0.6415 | 0.5369 | **0.8985** | 0.6499 | 0.6508 | 0.7354 | 0.7554 | 0.6446 | 0.6431 | 0.7769 | 0.5308 | 0.7662 | 0.8077 |
| 62 | InsectEPGR. | 0.9935 | 0.9839 | 0.9616 | 0.8554 | 0.8327 | 0.9807 | 0.8267 | 0.9617 | 0.8307 | **1.0000** | **1.0000** | **1.0000** | 0.7945 | 0.7818 | 0.9904 | 0.8779 | 0.9808 | 0.9968 |
| 63 | InsectEPGS. | 0.8761 | 0.9850 | 0.9774 | 0.8308 | 0.8235 | 0.9699 | 0.8011 | 0.9511 | 0.8198 | **1.0000** | **1.0000** | 0.9963 | 0.7973 | 0.8234 | 0.9813 | 0.9064 | 0.9514 | 0.9963 |
| 64 | InsectWing. | 0.7309 | 0.6673 | 0.6092 | 0.7119 | 0.6982 | 0.7077 | 0.7355 | 0.8609 | 0.8518 | 0.7286 | 0.7309 | 0.8305 | 0.6682 | 0.8327 | 0.8786 | 0.8750 | 0.8582 | **0.8882** |
| 65 | ItalyPowerDemand | **0.9891** | 0.9671 | 0.9674 | 0.9660 | 0.9726 | 0.9745 | 0.9863 | 0.9827 | 0.9818 | 0.9680 | 0.9735 | 0.9772 | 0.9644 | 0.9763 | **0.9891** | 0.9681 | 0.9781 | **0.9891** |
| 66 | LargeKitchenA. | 0.9507 | 0.8907 | 0.9403 | 0.7401 | 0.5693 | 0.9147 | 0.7520 | 0.8253 | 0.6747 | 0.9240 | 0.9480 | 0.9613 | 0.6187 | 0.6507 | 0.9533 | 0.8267 | 0.8240 | **0.9627** |
| 67 | Lightning2 | 0.7940 | 0.9090 | 0.7860 | 0.7187 | 0.6787 | 0.9010 | 0.8937 | 0.8219 | 0.8946 | 0.7280 | 0.7603 | **0.9430** | 0.8700 | 0.7193 | 0.9100 | 0.7857 | 0.9047 | 0.8857 |
| 68 | Lightning7 | 0.8406 | 0.8596 | 0.7419 | 0.7419 | 0.7059 | 0.8389 | 0.8264 | 0.7025 | 0.8378 | 0.7623 | 0.8328 | 0.9094 | **0.9101** | 0.6584 | 0.8892 | 0.6433 | 0.8754 | 0.8966 |
| 69 | Mallat | 0.9975 | 0.9875 | 0.9901 | 0.9857 | 0.9954 | 0.9962 | 0.9954 | 0.9996 | 0.9983 | **1.0000** | **1.0000** | 0.9983 | 0.9858 | **1.0000** | **1.0000** | 0.9821 | 0.9988 | 0.9988 |
| 70 | Meat | 0.9667 | **1.0000** | 0.8917 | 0.5417 | 0.9833 | **1.0000** | **1.0000** | 0.9750 | **1.0000** | 0.9833 | 0.9750 | 0.9917 | 0.4667 | **1.0000** | **1.0000** | 0.9667 | 0.9500 | 0.9917 |
| 71 | MedicalImages | 0.8397 | 0.8291 | 0.7530 | 0.8175 | 0.8010 | 0.8431 | 0.8677 | 0.8823 | 0.8772 | 0.8361 | 0.8536 | 0.9203 | 0.7872 | 0.8608 | 0.9212 | 0.9212 | 0.8932 | **0.9213** |
| 72 | MelbournePedestrian | 0.8145 | 0.8489 | 0.8485 | 0.9141 | 0.5888 | 0.8986 | 0.9567 | 0.6573 | 0.9321 | 0.9060 | 0.9203 | **0.9611** | 0.8271 | 0.2616 | 0.9532 | 0.8945 | 0.8641 | 0.9573 |
| 73 | MiddleAgeGroup. | 0.8177 | 0.7544 | 0.7527 | 0.7563 | 0.7607 | 0.7347 | 0.8140 | 0.8092 | 0.7692 | 0.7365 | 0.7310 | 0.8520 | 0.7454 | 0.7618 | **0.9170** | 0.7491 | 0.7382 | 0.8701 |
| 74 | MiddleCorrect. | 0.8834 | 0.8294 | 0.7936 | 0.7838 | 0.8507 | 0.8507 | **0.9339** | 0.9170 | 0.8924 | 0.8485 | 0.8608 | 0.9115 | 0.7105 | 0.7645 | 0.9059 | 0.7272 | 0.6262 | 0.9204 |
| 75 | MiddlePhalanxTW | 0.7074 | 0.6202 | 0.6184 | 0.6256 | 0.6456 | 0.6455 | 0.7309 | 0.7611 | 0.7027 | 0.6020 | 0.5931 | 0.7310 | 0.6310 | 0.6766 | **0.8377** | 0.7688 | 0.6383 | 0.8376 |
| 76 | MixedRegular. | 0.7463 | 0.9586 | 0.9580 | 0.9310 | 0.9296 | 0.9450 | 0.9053 | 0.7318 | 0.9645 | 0.9880 | **0.9897** | 0.9805 | 0.8660 | 0.9183 | 0.9764 | 0.9354 | 0.8468 | 0.9757 |
| 77 | MixedSmall. | 0.7339 | 0.9541 | 0.9530 | 0.9255 | 0.9275 | 0.9410 | 0.9038 | 0.6924 | 0.9545 | 0.9857 | **0.9885** | 0.9636 | 0.8606 | 0.9295 | 0.9537 | 0.9358 | 0.8075 | 0.9743 |
| 78 | MoteStrain | 0.9819 | 0.9670 | 0.9505 | 0.9435 | 0.9772 | 0.9717 | 0.9623 | 0.9757 | 0.9694 | **0.9906** | 0.9898 | 0.9866 | 0.8986 | 0.9474 | 0.9843 | 0.9623 | 0.9584 | 0.9843 |
| 79 | NonInvasive1. | 0.8542 | 0.9384 | 0.8799 | 0.9253 | 0.9129 | 0.9392 | 0.9349 | 0.9504 | 0.9639 | 0.9347 | 0.9634 | **0.9772** | 0.8580 | 0.9575 | 0.9692 | 0.9623 | 0.9365 | 0.9724 |
| 80 | NonInvasive2. | 0.8093 | 0.9461 | 0.9012 | 0.9400 | 0.9286 | 0.9490 | 0.9575 | 0.9451 | 0.9657 | 0.9469 | 0.9671 | 0.9740 | 0.8949 | 0.9681 | 0.9679 | 0.9673 | 0.9461 | **0.9782** |
| 81 | OliveOil | **0.9887** | 0.8304 | 0.5333 | 0.4000 | 0.5334 | 0.8500 | 0.8500 | 0.6967 | 0.9000 | 0.9000 | 0.9167 | 0.8333 | 0.4167 | 0.6500 | 0.7333 | 0.7833 | 0.4167 | 0.8333 |
| 82 | OSULeaf | 0.7000 | 0.8833 | 0.8370 | 0.6493 | 0.6111 | 0.8937 | 0.8247 | 0.8799 | 0.8085 | **0.9774** | 0.9751 | 0.9594 | 0.7607 | 0.7378 | 0.9526 | 0.7993 | 0.7903 | 0.9504 |
| 83 | PhalangesOutlines. | 0.3019 | 0.3664 | 0.7466 | 0.7803 | 0.8423 | 0.8439 | **0.9316** | 0.8619 | 0.8548 | 0.8363 | 0.8638 | 0.9097 | 0.7258 | 0.6204 | 0.9267 | 0.7517 | 0.6441 | 0.9300 |
| 84 | Phoneme | 0.7800 | 0.7200 | 0.4144 | 0.3587 | 0.1929 | 0.4204 | 0.1981 | 0.2781 | 0.5569 | 0.6000 | 0.5043 | 0.7616 | 0.3052 | 0.5469 | 0.5141 | 0.7161 | 0.6611 | **0.7844** |
| 85 | PickupGesture | 0.1091 | 0.1700 | 0.8000 | 0.7800 | 0.6200 | 0.8400 | 0.7200 | 0.6300 | 0.7000 | 0.8600 | 0.8700 | 0.8700 | 0.5400 | 0.6500 | 0.8700 | 0.7700 | 0.7500 | **0.9000** |
| 86 | PigAirwayPressure | 0.5164 | 0.6446 | 0.2182 | 0.0575 | 0.0353 | 0.4038 | 0.2446 | 0.3500 | 0.4280 | 0.8749 | **0.8878** | 0.6718 | 0.4437 | 0.2967 | 0.5593 | 0.4984 | 0.5848 | 0.6432 |
| 87 | PigArtPressure | 0.4822 | 0.6731 | 0.9456 | 0.0929 | 0.1312 | 0.9424 | 0.3826 | 0.5308 | 0.4893 | 0.9744 | 0.9648 | **1.0000** | 0.6969 | 0.3605 | 0.9458 | 0.6361 | 0.6974 | 0.9937 |
| 88 | PigCVP | 0.4348 | 0.5121 | 0.7083 | 0.0449 | 0.0321 | 0.8753 | 0.4181 | 0.5269 | 0.6079 | 0.9232 | 0.9166 | 0.9364 | 0.7420 | 0.3287 | 0.8212 | 0.4643 | 0.7042 | **0.9458** |
| 89 | PLAID | 0.8691 | 0.8424 | 0.3912 | 0.4832 | 0.7421 | 0.5503 | 0.5951 | 0.4107 | 0.5867 | 0.8305 | **0.9115** | 0.7478 | 0.3818 | 0.3045 | 0.6369 | 0.4887 | 0.4805 | 0.6192 |
| 90 | Plane | **1.0000** | **1.0000** | 0.9762 | 0.9714 | 0.9381 | 0.9905 | 0.9810 | 0.9952 | 0.9810 | **1.0000** | **1.0000** | **1.0000** | **1.0000** | 0.9857 | 0.9952 | 0.9714 | 0.9619 | **1.0000** |
| 91 | PowerCons | 0.9139 | 0.9194 | 0.8889 | 0.9444 | 0.9722 | 0.9861 | 0.9722 | 0.9333 | 0.9833 | 0.9722 | 0.9660 | 0.9694 | 0.7722 | 0.9556 | **0.9944** | 0.9528 | 0.9833 | **0.9944** |
| 92 | ProximalAgeGroup. | 0.8727 | 0.8496 | 0.8446 | 0.8231 | 0.8496 | 0.8545 | 0.8860 | 0.8727 | 0.8793 | 0.8364 | 0.8364 | 0.9058 | 0.8000 | 0.8562 | **0.9190** | 0.8413 | 0.8628 | 0.9091 |
| 93 | ProximalCorrect. | 0.9214 | 0.8889 | 0.8383 | 0.8303 | 0.8799 | 0.8833 | **0.9406** | 0.9170 | 0.8945 | 0.8687 | 0.9068 | 0.9192 | 0.7980 | 0.8373 | 0.9238 | 0.7531 | 0.7037 | 0.9294 |
| 94 | ProximalPhalanxTW | 0.8083 | 0.8215 | 0.8000 | 0.7653 | 0.8347 | 0.8281 | 0.8512 | 0.8798 | 0.8645 | 0.8116 | 0.8033 | 0.8777 | 0.8033 | 0.8198 | 0.9074 | 0.8248 | 0.8198 | **0.9273** |
| 95 | RefrigerationDevices | 0.5747 | 0.7773 | 0.7193 | 0.5170 | 0.5146 | 0.7627 | 0.5693 | 0.6400 | 0.7267 | 0.7800 | 0.7773 | 0.7667 | 0.5947 | 0.5347 | **0.8333** | 0.5253 | 0.6960 | 0.8027 |
| 96 | Rock | 0.6429 | 0.7143 | 0.6857 | 0.7857 | 0.7714 | 0.8286 | 0.8143 | 0.6940 | **0.9143** | **0.9143** | **0.9143** | 0.7429 | **0.9143** | 0.9000 | 0.8286 | 0.7286 | **0.9143** | 0.8286 |
| 97 | ScreenType | 0.6093 | 0.5480 | 0.6852 | 0.4805 | 0.4827 | 0.5520 | 0.6347 | 0.5093 | 0.6480 | 0.6720 | 0.6827 | 0.7613 | 0.4280 | 0.5573 | **0.8267** | 0.3907 | 0.6480 | 0.7027 |
| 98 | SemgHandGenderCh2 | 0.9378 | 0.7456 | 0.8745 | 0.9516 | 0.9111 | 0.9544 | 0.9367 | 0.9378 | 0.9389 | 0.9200 | 0.9511 | 0.9144 | 0.7044 | 0.8144 | 0.9744 | 0.9444 | 0.9211 | **0.9778** |
| 99 | SemgMovementCh2. | 0.7156 | 0.4056 | 0.5769 | 0.6980 | 0.4667 | 0.7767 | 0.7800 | 0.6311 | 0.7989 | 0.5933 | 0.7011 | 0.6744 | 0.3111 | 0.6611 | **0.8578** | 0.8422 | 0.7933 | 0.8256 |
| 100 | SemgHandSubjectCh2 | 0.6922 | 0.6356 | 0.8286 | 0.9310 | 0.8467 | 0.9344 | 0.9411 | 0.8278 | 0.9089 | 0.8944 | 0.9267 | 0.9433 | 0.7156 | 0.9044 | 0.9556 | 0.9400 | 0.9389 | **0.9633** |

| ID | Dataset | FCN | T-Loss | Selftime | TS-TCC | TST | TS2Vec | TimesNet | PatchTST | GPT4TS | RDST | MR-H | Incep | ShapeConv | MoTCN | TSLANet | UniTS | Medformer | SoftShape |
|----|---------|-----|--------|----------|--------|-----|--------|----------|----------|--------|------|------|-------|-----------|-------|---------|-------|-----------|-----------|
| 101 | ShakeWiimoteZ. | 0.9200 | 0.9100 | 0.8900 | 0.8500 | 0.6800 | 0.9300 | 0.5900 | 0.6900 | 0.7200 | 0.9200 | 0.9300 | 0.9600 | 0.8700 | 0.7400 | 0.9500 | 0.8400 | 0.8300 | **0.9800** |
| 102 | ShapeletSim | 0.9700 | 0.9200 | 0.9450 | 0.6050 | 0.5600 | 0.9850 | 0.5500 | 0.9150 | 0.6300 | **1.0000** | **1.0000** | **1.0000** | 0.8300 | 0.6450 | **1.0000** | 0.9800 | 0.8850 | 0.9950 |
| 103 | ShapesAll | 0.7267 | 0.8925 | 0.8835 | 0.7777 | 0.7292 | 0.9133 | 0.8325 | 0.8833 | 0.8500 | 0.9417 | 0.9533 | **0.9583** | 0.8225 | 0.8950 | 0.9083 | 0.9192 | 0.8617 | 0.9575 |
| 104 | SmallAppliances. | 0.7987 | 0.7227 | 0.8256 | 0.7095 | 0.5747 | 0.7347 | 0.7840 | 0.6387 | 0.7213 | 0.8133 | 0.8333 | 0.9008 | 0.5413 | 0.6573 | 0.7040 | 0.7840 | 0.8173 | **0.9053** |
| 105 | SmoothSubspace | 0.9467 | 0.9233 | 0.9367 | 0.9433 | 0.9800 | 0.9633 | 0.9767 | 0.9633 | 0.9133 | **0.9900** | 0.9833 | 0.9800 | 0.9433 | 0.8567 | 0.9567 | 0.9400 | 0.9733 | 0.9867 |
| 106 | SonyAISurface1. | 0.9984 | 0.9920 | 0.9726 | 0.9775 | 0.9823 | 0.9952 | 0.9952 | 0.9888 | 0.9888 | 0.9984 | **1.0000** | 0.9984 | 0.9646 | 0.9936 | 0.9904 | 0.9888 | 0.9823 | 0.9936 |
| 107 | SonyAISurface2. | 0.9990 | 0.9878 | 0.9376 | 0.9849 | 0.9847 | 0.9949 | 0.9939 | 0.9939 | 0.9949 | 0.9980 | 0.9990 | **1.0000** | 0.8990 | 0.9847 | 0.9969 | 0.9867 | 0.9837 | 0.9980 |
| 108 | StarLightCurves | 0.9803 | 0.9783 | 0.9793 | 0.9738 | 0.9734 | 0.9801 | 0.9616 | **0.9912** | 0.9818 | 0.9827 | 0.9838 | 0.9903 | 0.9686 | 0.9892 | 0.9910 | 0.9675 | 0.9694 | 0.9905 |
| 109 | Strawberry | 0.9756 | 0.9614 | 0.9426 | 0.9414 | 0.9735 | 0.9685 | 0.9766 | 0.9624 | 0.9736 | 0.9786 | 0.9797 | 0.9797 | 0.8790 | 0.9542 | 0.9817 | 0.9532 | 0.9054 | **0.9868** |
| 110 | SwedishLeaf | **0.9929** | 0.9404 | 0.9158 | 0.9423 | 0.9253 | 0.9502 | 0.9449 | 0.9778 | 0.9671 | 0.9707 | 0.9751 | 0.9849 | 0.9250 | 0.9618 | 0.9893 | 0.9760 | 0.9689 | 0.9867 |
| 111 | Symbols | **0.9961** | 0.9824 | 0.9798 | 0.9733 | 0.9765 | 0.9863 | 0.9618 | 0.9922 | 0.9667 | 0.9951 | 0.9951 | **0.9961** | 0.9686 | 0.9765 | 0.9833 | 0.9471 | 0.9745 | 0.9941 |
| 112 | SyntheticControl | 0.9700 | 0.9883 | 0.9683 | 0.9950 | 0.9767 | 0.9983 | 0.9767 | 0.9700 | 0.9717 | 0.9933 | 0.9933 | **1.0000** | 0.9783 | 0.9400 | 0.9967 | 0.9933 | 0.9900 | 0.9967 |
| 113 | ToeSegmentation1 | 0.9664 | 0.9551 | 0.9143 | 0.9181 | 0.6903 | 0.9588 | 0.8514 | 0.9219 | 0.8846 | 0.9776 | 0.9776 | 0.9812 | 0.9293 | 0.6761 | 0.9778 | 0.7500 | 0.8771 | **0.9852** |
| 114 | ToeSegmentation2 | 0.9282 | 0.9219 | 0.9283 | 0.8250 | 0.8435 | 0.9517 | 0.8316 | 0.8510 | 0.8143 | 0.9638 | 0.9517 | 0.9704 | 0.9706 | 0.8561 | 0.9522 | 0.9045 | 0.8554 | **0.9882** |
| 115 | Trace | **1.0000** | **1.0000** | 0.9950 | 0.9150 | 0.9700 | **1.0000** | 0.9400 | **1.0000** | 0.8900 | **1.0000** | **1.0000** | **1.0000** | **1.0000** | 0.9800 | **1.0000** | 0.8100 | **1.0000** | **1.0000** |
| 116 | TwoLeadECG | **1.0000** | 0.9983 | 0.9932 | 0.9965 | 0.9914 | 0.9991 | 0.9983 | 0.9991 | 0.9948 | 0.9991 | **1.0000** | **1.0000** | 0.8615 | 0.9983 | 0.9991 | 0.9991 | 0.9974 | **1.0000** |
| 117 | TwoPatterns | 0.9454 | **1.0000** | 0.9513 | 0.9990 | 0.9994 | **1.0000** | 0.9980 | 0.9990 | 0.9962 | **1.0000** | **1.0000** | **1.0000** | 0.9944 | **1.0000** | 0.9998 | 0.9994 | 0.9986 | **1.0000** |
| 118 | UMD | 0.8778 | 0.9944 | 0.9389 | 0.9611 | 0.9778 | 0.9944 | 0.9833 | **1.0000** | 0.9778 | **1.0000** | **1.0000** | **1.0000** | 0.9056 | **1.0000** | 0.9944 | 0.9833 | **1.0000** | 0.9944 |
| 119 | UWaveGestureAll. | 0.9245 | 0.9518 | 0.8595 | 0.9750 | 0.9602 | 0.9652 | 0.9504 | **0.9913** | 0.9708 | 0.9882 | 0.9897 | 0.9812 | 0.9252 | 0.9888 | **0.9913** | 0.9795 | 0.9757 | 0.9900 |
| 120 | UWaveGestureX. | 0.8665 | 0.8457 | 0.6680 | 0.8385 | 0.8091 | 0.8513 | 0.8924 | 0.9214 | 0.9002 | 0.8803 | 0.9015 | 0.9196 | 0.7977 | 0.9216 | 0.9413 | **0.9458** | 0.9219 | 0.9406 |
| 121 | UWaveGestureY. | 0.7906 | 0.7887 | 0.5342 | 0.7573 | 0.7300 | 0.7874 | 0.8455 | 0.8955 | 0.8399 | 0.8109 | 0.8481 | 0.8850 | 0.6936 | 0.8828 | **0.9194** | 0.9100 | 0.8777 | 0.9109 |
| 122 | UWaveGestureZ. | 0.8484 | 0.8021 | 0.6076 | 0.7762 | 0.4790 | 0.8024 | 0.8669 | 0.9006 | 0.8535 | 0.8314 | 0.8584 | 0.8812 | 0.7693 | 0.8781 | **0.9219** | 0.9143 | 0.8785 | 0.9170 |
| 123 | Wafer | **1.0000** | 0.9989 | 0.9587 | 0.9982 | 0.9992 | 0.9990 | 0.9986 | 0.9992 | 0.9992 | **1.0000** | **1.0000** | 0.9999 | 0.9785 | 0.9993 | 0.9999 | 0.9994 | 0.9971 | **1.0000** |
| 124 | Wine | 0.5411 | 0.9553 | 0.6743 | 0.5221 | 0.5854 | **0.9648** | 0.8289 | 0.5135 | 0.8561 | 0.9553 | 0.9644 | 0.7593 | 0.5490 | 0.5043 | 0.7494 | 0.4953 | 0.5130 | 0.8842 |
| 125 | WordSynonyms | 0.5967 | 0.8022 | 0.5718 | 0.7444 | 0.7083 | 0.7989 | 0.8442 | 0.8690 | 0.7823 | 0.8575 | 0.8740 | **0.9315** | 0.8254 | 0.8431 | 0.8928 | 0.8762 | 0.8486 | 0.9175 |
| 126 | Worms | 0.7906 | 0.6821 | 0.6860 | 0.5423 | 0.4148 | 0.7210 | 0.7446 | 0.8072 | 0.7097 | 0.7051 | 0.7357 | 0.8181 | 0.5856 | 0.5236 | 0.8429 | 0.7800 | 0.7876 | **0.8453** |
| 127 | WormsTwoClass | 0.8025 | 0.7634 | 0.7557 | 0.6898 | 0.6240 | 0.7716 | 0.7833 | **0.9037** | 0.8221 | 0.7715 | 0.7753 | 0.7983 | 0.6710 | 0.6906 | 0.8692 | 0.8414 | 0.8376 | 0.8802 |
| 128 | Yoga | 0.9718 | 0.9661 | 0.7699 | 0.9207 | 0.9497 | 0.9709 | 0.9539 | 0.9739 | 0.9476 | 0.9842 | 0.9906 | **0.9933** | 0.7661 | 0.9845 | 0.9694 | 0.8970 | 0.9718 | 0.9830 |
| | Avg. Acc | 0.8296 | 0.8325 | 0.8017 | 0.7807 | 0.7755 | 0.8691 | 0.8367 | 0.8265 | 0.8593 | 0.8897 | 0.8972 | 0.9181 | 0.7688 | 0.7938 | 0.9205 | 0.8502 | 0.8541 | **0.9334** |
| | Avg. Rank | 9.53 | 11.12 | 13.80 | 13.96 | 13.54 | 8.43 | 10.13 | 9.56 | 9.34 | 6.41 | 5.51 | 4.05 | 13.91 | 11.37 | 3.68 | 9.66 | 9.26 | **2.72** |
| | Win | 13 | 9 | 0 | 0 | 1 | 9 | 7 | 12 | 6 | 23 | 29 | 29 | 5 | 9 | 31 | 5 | 7 | **53** |

Table 10: The detailed test classification accuracy of LightTS, Shapeformer, and SoftShape on 18 UCR datasets.

| Dataset | LightTS | Shapeformer | SoftShape (Ours) |
|---------|---------|-------------|------------------|
| ArrowHead | **0.9480** | 0.8439 | 0.9435 |
| CBF | **1.0000** | 0.9978 | **1.0000** |
| CricketX | 0.9282 | 0.6564 | **0.9321** |
| DistalPhalanxOutlineAgeGroup | 0.8664 | 0.7921 | **0.9518** |
| DistalPhalanxOutlineCorrect | 0.8460 | 0.8151 | **0.9054** |
| ECG5000 | 0.9630 | 0.9166 | **0.9796** |
| EOGVerticalSignal | 0.7737 | 0.5635 | **0.9007** |
| EthanolLevel | 0.9044 | 0.7261 | **0.9204** |
| Fish | **0.9800** | 0.9114 | 0.9771 |
| GunPoint | 0.9850 | **0.9900** | **0.9900** |
| InsectWingbeatSound | 0.8300 | 0.6586 | **0.8882** |
| ItalyPowerDemand | 0.9708 | 0.9690 | **0.9891** |
| MelbournePedestrian | **0.9879** | 0.7134 | 0.9573 |
| MiddlePhalanxTW | 0.7673 | 0.6293 | **0.8376** |
| MixedShapesRegularTrain | **0.9795** | 0.8055 | 0.9757 |
| OSULeaf | **0.9523** | 0.7649 | 0.9504 |
| Trace | **1.0000** | **1.0000** | **1.0000** |
| WordSynonyms | 0.8892 | 0.6773 | **0.9161** |
| **Avg. Acc** | 0.9207 | 0.8017 | **0.9453** |
| **Avg. Rank** | 1.67 | 2.78 | **1.28** |
| **Win** | 7 | 2 | **13** |
| **P-value** | 8.91E-03 | 9.80E-06 | - |

## B.2. Ablation Study

Table 11 presents the detailed test classification accuracy results for various ablation components of SoftShape across 128 UCR time series datasets.

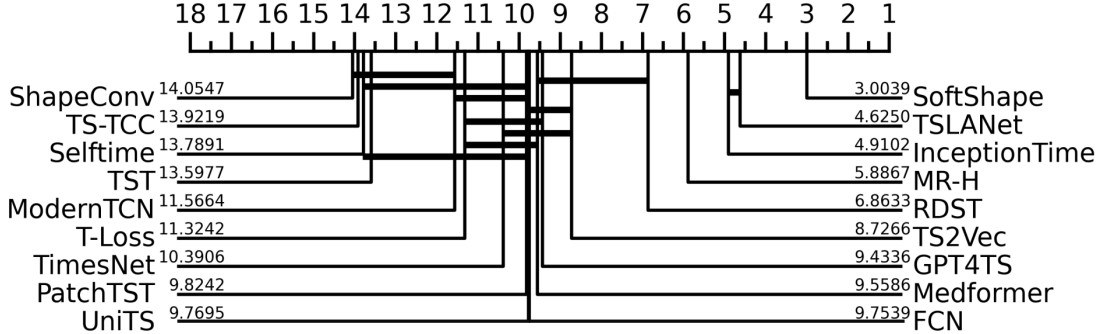

Figure 6: The critical diagram (CD) ([Demšar, 2006](#)) illustrates statistical testing comparisons between SoftShape and baseline methods on the 128 UCR time series datasets. A smaller CD value indicates better method performance. The absence of a connecting line between the two methods signifies a statistically significant performance difference.

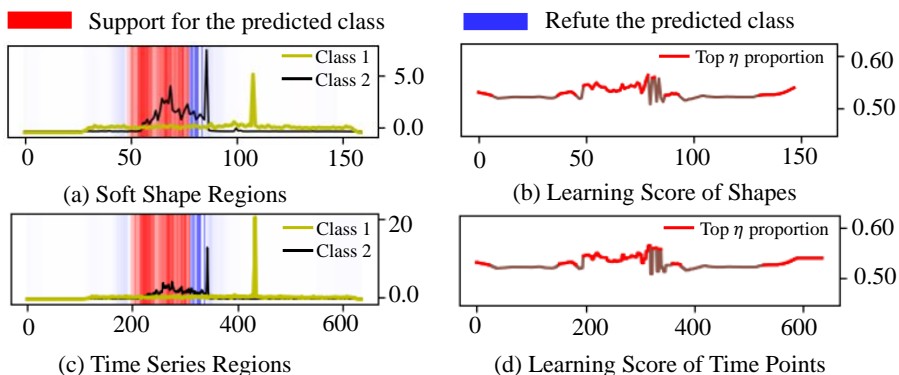

(a) Soft Shape Regions

(b) Learning Score of Shapes

(c) Time Series Regions

(d) Learning Score of Time Points

Figure 7: The MIL visualization on the *Lightning2* dataset.

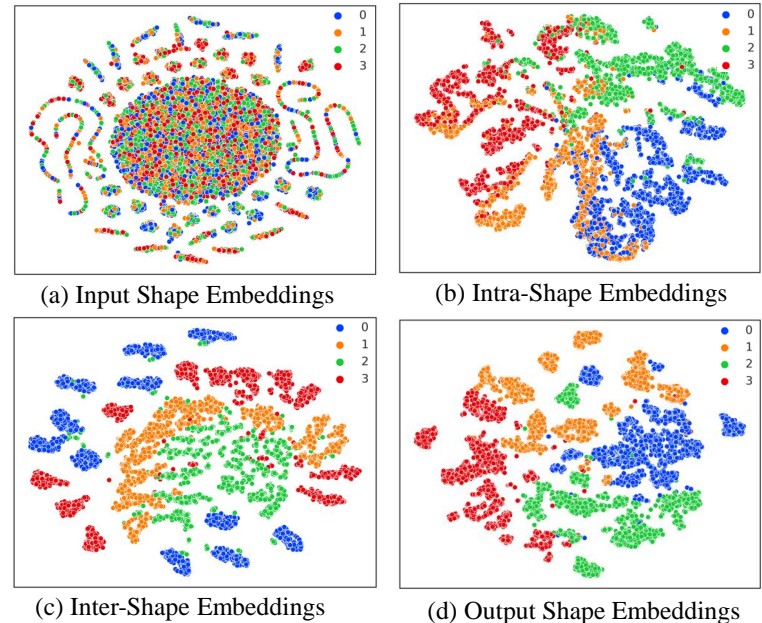

(a) Input Shape Embeddings

(b) Intra-Shape Embeddings

(c) Inter-Shape Embeddings

(d) Output Shape Embeddings

Figure 8: The t-SNE visualization on the *TwoPatterns* dataset.

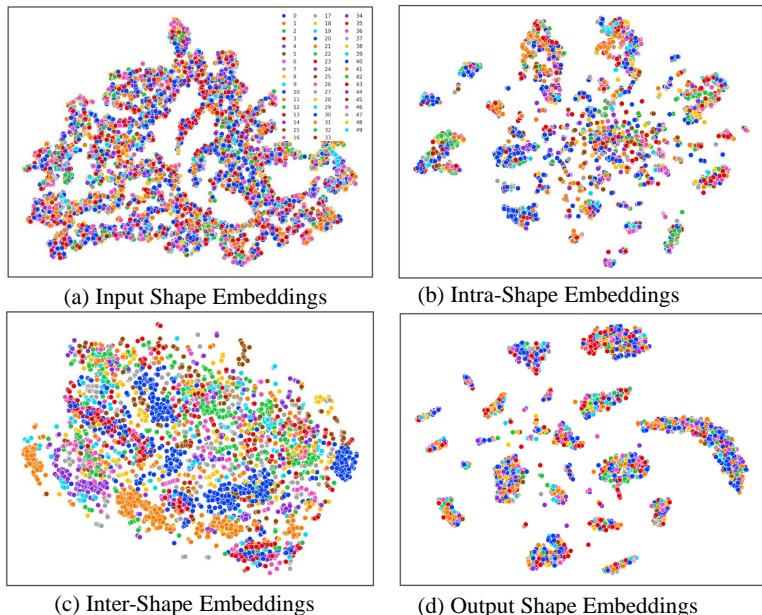

(a) Input Shape Embeddings

(b) Intra-Shape Embeddings

(c) Inter-Shape Embeddings

(d) Output Shape Embeddings

Figure 9: The t-SNE visualization on the *Fiftywords* dataset.

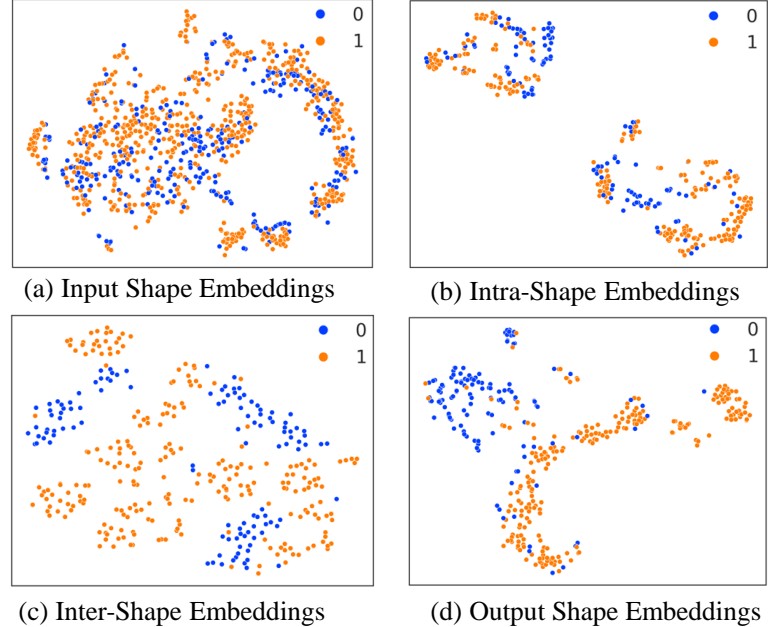

(a) Input Shape Embeddings

(b) Intra-Shape Embeddings

(c) Inter-Shape Embeddings

(d) Output Shape Embeddings

Figure 10: The t-SNE visualization on the *ECG200* dataset.

## B.3. Shape Sparsification and Learning Analysis

Due to the considerable time requirements for conducting experimental evaluations on all 128 UCR time series datasets for some baselines and hyperparameter analysis, 18 datasets were chosen from the 128 UCR datasets based on four key criteria: the number of samples, the length of the sample sequences, the number of classes, and the relevance to various application scenarios. Specifically, these 18 datasets vary in sample size from 200 to 5000, in sequence length from 24 to 1751, and in classes from 2 to 25. Besides, the selected datasets cover diverse application areas, including handwritten font sequence recognition (e.g., the *WordSynonyms* dataset), human activity recognition (e.g., the *CricketX* dataset), and medical diagnosis

Table 11: The detailed test classification accuracy of the ablation study on 128 UCR datasets. Among these, *w/o II* refers to the **w/o Intra & Inter** method. The best results are in **bold**.

| ID | Dataset | SoftShape | w/o Soft | w/o Intra | w/o Inter | w/o II | with Linear | ID | Dataset | SoftShape | w/o Soft | w/o Intra | w/o Inter | w/o II | with Linear |
|---|---|---|---|---|---|---|---|---|---|---|---|---|---|---|---|
| 1 | ACSF1 | **0.9100** | 0.8700 | 0.8150 | 0.8900 | 0.8700 | 0.8850 | 67 | Lightning2 | 0.8857 | **0.9110** | 0.8819 | 0.9033 | 0.8450 | 0.9023 |
| 2 | Adiac | **0.9374** | 0.8810 | 0.9297 | 0.8274 | 0.8094 | 0.9259 | 68 | Lightning7 | 0.8966 | 0.9308 | **0.9517** | 0.8997 | 0.9101 | 0.8894 |
| 3 | AllGestureWX. | **0.8980** | 0.8680 | 0.8970 | 0.8040 | 0.5850 | 0.8620 | 69 | Mallat | 0.9988 | **1.0000** | **1.0000** | 0.9996 | 0.9992 | 0.9996 |
| 4 | AllGestureWY. | **0.9110** | 0.9030 | 0.9080 | 0.8360 | 0.6260 | 0.8930 | 70 | Meat | 0.9917 | **1.0000** | 0.9917 | 0.9917 | 0.9917 | **1.0000** |
| 5 | AllGestureWZ. | **0.8850** | 0.8350 | 0.8786 | 0.8080 | 0.5960 | 0.8750 | 71 | MedicalImages | 0.9213 | 0.9221 | 0.9204 | 0.9134 | 0.7783 | **0.9256** |
| 6 | ArrowHead | 0.9435 | 0.9672 | **0.9765** | 0.9627 | 0.9248 | 0.9577 | 72 | MelbournePedestrian | 0.9573 | 0.9542 | 0.9570 | **0.9652** | 0.8466 | 0.9521 |
| 7 | Beef | **0.8667** | 0.8333 | 0.8500 | 0.7500 | 0.8000 | 0.8167 | 73 | MiddleAgeGroup. | 0.8701 | 0.8430 | **0.9134** | 0.8179 | 0.7743 | 0.8304 |
| 8 | BeetleFly | 0.9750 | **1.0000** | 0.9800 | 0.8750 | 0.8500 | 0.9250 | 74 | MiddleCorrect. | 0.9204 | **0.9260** | 0.9150 | 0.8598 | 0.8239 | 0.9070 |
| 9 | BirdChicken | 0.9000 | **0.9250** | 0.8750 | 0.9250 | 0.9250 | 0.8500 | 75 | MiddlePhalanxTW | 0.8376 | 0.8179 | **0.8629** | 0.7203 | 0.6746 | 0.8305 |
| 10 | BME | 0.9944 | **1.0000** | 0.9944 | 0.9944 | **1.0000** | 0.9944 | 76 | MixedRegular. | **0.9757** | 0.9750 | 0.9713 | 0.9627 | 0.9350 | 0.9682 |
| 11 | Car | **0.9583** | 0.9167 | 0.9483 | 0.9250 | 0.9333 | 0.9250 | 77 | MixedSmall. | **0.9743** | 0.9695 | 0.9719 | 0.9671 | 0.9446 | 0.9695 |
| 12 | CBF | **1.0000** | 1.0000 | 0.9989 | 1.0000 | 0.9978 | 0.9989 | 78 | MoteStrain | 0.9843 | 0.9843 | 0.9714 | 0.9835 | 0.9764 | **0.9851** |
| 13 | Chinatown | 0.9890 | 0.9890 | 0.9808 | **0.9896** | 0.9808 | 0.9863 | 79 | NonInvasive1. | 0.9724 | 0.9676 | **0.9787** | 0.9625 | 0.9554 | 0.9625 |
| 14 | ChlorineCon. | 0.9988 | 0.9988 | 0.9991 | 0.9991 | 0.9025 | **0.9998** | 80 | NonInvasive2. | **0.9782** | 0.9665 | 0.9684 | 0.9636 | 0.9575 | 0.9604 |
| 15 | CinCECGTorso | **1.0000** | 1.0000 | 0.9993 | 0.9993 | 1.0000 | 0.9993 | 81 | OliveOil | 0.8333 | 0.8167 | 0.8333 | 0.8167 | 0.7333 | **0.8667** |
| 16 | Coffee | **1.0000** | 1.0000 | 1.0000 | 1.0000 | 1.0000 | 1.0000 | 82 | OSULeaf | **0.9504** | 0.9257 | 0.9302 | 0.8626 | 0.8310 | 0.9077 |
| 17 | Computers | 0.8160 | 0.8120 | 0.8040 | 0.8540 | **0.8640** | 0.8240 | 83 | PhalangesOutlines. | 0.9300 | **0.9312** | 0.9135 | 0.9022 | 0.7957 | 0.9105 |
| 18 | CricketX | 0.9321 | 0.9064 | 0.9223 | 0.9038 | 0.8538 | **0.9346** | 84 | Phoneme | **0.7844** | 0.6545 | 0.6456 | 0.6234 | 0.6051 | 0.6567 |
| 19 | CricketY | 0.9154 | 0.8808 | 0.9069 | 0.8667 | 0.7974 | **0.9256** | 85 | PickupGesture. | 0.9000 | 0.8600 | 0.8900 | 0.9000 | 0.8100 | 0.8200 |
| 20 | CricketZ | 0.9372 | 0.9013 | 0.9133 | 0.8923 | 0.8179 | **0.9467** | 86 | PigAirwayPressure | **0.6432** | 0.2446 | 0.5664 | 0.5625 | 0.5662 | 0.4963 |
| 21 | Crop | 0.8765 | 0.7342 | 0.7435 | 0.7236 | 0.5973 | **0.8856** | 87 | PigArtPressure | **0.9937** | 0.8622 | 0.9904 | 0.8120 | 0.6972 | 0.9745 |
| 22 | DiatomSizeRe. | **1.0000** | 1.0000 | 0.9938 | 0.9938 | 1.0000 | 0.9938 | 88 | PigCVP | **0.9458** | 0.7926 | 0.9172 | 0.6783 | 0.6880 | 0.8344 |
| 23 | DistalPhalanxOut. | **0.9518** | 0.9462 | 0.9407 | 0.8975 | 0.8961 | 0.9462 | 89 | PLAID | **0.6192** | 0.5922 | 0.6025 | 0.5457 | 0.4879 | 0.5383 |
| 24 | DistalPhalanxOut. | 0.9054 | **0.9077** | 0.8848 | 0.8937 | 0.8631 | 0.8871 | 90 | Plane | **1.0000** | 0.9952 | 0.9902 | **1.0000** | 0.9952 | 0.9952 |
| 25 | DistalPhalanxTW | 0.9037 | 0.9092 | 0.9203 | 0.8851 | 0.8628 | **0.9292** | 91 | PowerCons | 0.9944 | 0.9917 | 0.9902 | **0.9972** | 0.9806 | 0.9750 |
| 26 | DodgerLoopDay | 0.8623 | 0.8625 | 0.8500 | **0.8698** | 0.8183 | 0.8433 | 92 | ProximalAgeGroup. | 0.9091 | **0.9107** | 0.9022 | 0.8810 | 0.8744 | 0.9074 |
| 27 | DodgerLoopGame | **0.9433** | 0.9308 | 0.9260 | 0.8813 | 0.8556 | 0.9268 | 93 | ProximalCorrect. | 0.9294 | **0.9541** | 0.9473 | 0.8485 | 0.8351 | 0.9294 |
| 28 | DodgerLoopWeek. | **1.0000** | 1.0000 | 0.9917 | 0.9937 | 0.9310 | 0.9935 | 94 | ProximalPhalanxTW | **0.9273** | 0.8876 | 0.9207 | 0.8529 | 0.8612 | 0.8942 |
| 29 | Earthquakes | 0.8917 | 0.9135 | 0.8940 | **0.9331** | 0.9135 | 0.9070 | 95 | RefrigerationDevices | **0.8027** | 0.7347 | 0.7940 | 0.7653 | 0.7480 | 0.7720 |
| 30 | ECG200 | 0.9475 | 0.9400 | 0.9400 | **0.9600** | 0.9450 | 0.9050 | 96 | Rock | 0.8286 | 0.8571 | **0.8857** | 0.8714 | 0.8571 | 0.8286 |
| 31 | ECG5000 | 0.9796 | 0.9810 | **0.9818** | 0.9720 | 0.9566 | 0.9796 | 97 | ScreenType | 0.7027 | 0.7693 | 0.7760 | **0.8253** | 0.7853 | 0.6933 |
| 32 | ECGFiveDays | **1.0000** | 1.0000 | 1.0000 | 1.0000 | 1.0000 | 1.0000 | 98 | SemgHandGenderCh2 | 0.9778 | 0.9211 | 0.9478 | **0.9811** | 0.9211 | 0.9456 |
| 33 | ElectricDevices | **0.9208** | 0.8675 | 0.8756 | 0.8651 | 0.7428 | 0.8654 | 99 | SemgMovementCh2. | 0.8256 | 0.7867 | **0.8489** | 0.8300 | 0.8022 | 0.8156 |
| 34 | EOGHorizontalS. | **0.9007** | 0.8386 | 0.8731 | 0.8109 | 0.7612 | 0.8275 | 100 | SemgHandSubjectCh2 | 0.9633 | **0.9667** | 0.9433 | 0.9611 | 0.9489 | 0.9544 |
| 35 | EOGVerticalS. | **0.8814** | 0.8413 | 0.8565 | 0.7212 | 0.7861 | 0.7723 | 101 | ShakeWiimoteZ. | **0.9800** | 0.9600 | 0.9700 | 0.9700 | 0.9100 | **0.9800** |
| 36 | EthanolLevel | 0.9204 | **0.9243** | 0.8895 | 0.6993 | 0.7730 | 0.8446 | 102 | ShapeletSim | 0.9950 | **1.0000** | **1.0000** | 0.9350 | 0.8900 | **1.0000** |
| 37 | FaceAll | 0.9978 | **0.9982** | 0.9908 | 0.9911 | 0.9538 | 0.9973 | 103 | ShapesAll | **0.9575** | 0.9158 | 0.9508 | 0.9317 | 0.8700 | 0.9442 |
| 38 | FaceFour | 0.9909 | **1.0000** | 1.0000 | 0.9913 | 0.9913 | 0.9913 | 104 | SmallAppliances. | 0.9053 | 0.8973 | **0.9067** | 0.8813 | 0.7747 | 0.8773 |
| 39 | FacesUCR | 0.9956 | 0.9960 | **0.9969** | 0.9880 | 0.9507 | 0.9964 | 105 | SmoothSubspace | **0.9867** | 0.9633 | 0.9800 | **0.9867** | 0.9300 | 0.9800 |
| 40 | FiftyWords | 0.9359 | 0.8972 | **0.9414** | 0.9395 | 0.8641 | 0.9359 | 106 | SonyAISurface1. | 0.9936 | 0.9920 | 0.9936 | **0.9968** | 0.9852 | 0.9936 |
| 41 | Fish | 0.9771 | **0.9800** | 0.9800 | 0.9314 | 0.9371 | 0.9571 | 107 | SonyAISurface2. | **0.9980** | 0.9839 | 0.9909 | 0.9939 | 0.9847 | 0.9969 |
| 42 | FordA | 0.9772 | 0.9742 | **0.9811** | 0.9259 | 0.8417 | 0.9793 | 108 | StarLightCurves | **0.9905** | 0.9675 | 0.9679 | 0.9563 | **0.9905** | 0.9656 |
| 43 | FordB | **0.9744** | 0.9683 | 0.9721 | 0.9240 | 0.8655 | 0.9696 | 109 | Strawberry | **0.9868** | 0.9718 | 0.9756 | 0.9746 | 0.9705 | 0.9827 |
| 44 | FreezerRegular. | 0.9990 | 0.9990 | **0.9997** | 0.9973 | 0.9877 | 0.9990 | 110 | SwedishLeaf | **0.9867** | 0.9813 | 0.9804 | 0.9778 | 0.9484 | 0.9858 |
| 45 | FreezerSmall. | 0.9990 | 0.9993 | **0.9997** | 0.9972 | 0.9920 | 0.9986 | 111 | Symbols | **0.9941** | 0.9912 | 0.9922 | 0.9863 | 0.9902 | 0.9931 |
| 46 | Fungi | **1.0000** | 1.0000 | 1.0000 | 1.0000 | 0.9951 | 1.0000 | 112 | SyntheticControl | 0.9967 | 0.9917 | 0.9905 | **0.9983** | 0.9867 | 0.9917 |
| 47 | GestureMidAirD1 | **0.8640** | 0.7550 | 0.8553 | 0.8287 | 0.8169 | 0.8201 | 113 | ToeSegmentation1 | 0.9852 | 0.9739 | **0.9855** | 0.9000 | 0.9037 | 0.9741 |
| 48 | GestureMidAirD2 | **0.7962** | 0.6693 | 0.7847 | 0.7520 | 0.6871 | 0.7584 | 114 | ToeSegmentation2 | 0.9882 | 0.9824 | **0.9884** | 0.9865 | 0.9283 | 0.9820 |
| 49 | GestureMidAirD3 | **0.7613** | 0.5629 | 0.7226 | 0.6935 | 0.6135 | 0.6815 | 115 | Trace | **1.0000** | 0.9950 | 0.9950 | 0.9950 | 0.9800 | **1.0000** |
| 50 | GesturePebbleZ1 | 0.9867 | 0.9934 | **0.9967** | 0.9672 | 0.9606 | **0.9967** | 116 | TwoLeadECG | **1.0000** | 0.9991 | 0.9981 | 0.9991 | 0.9957 | 0.9991 |
| 51 | GesturePebbleZ2 | **0.9836** | 0.9770 | 0.9836 | 0.9738 | 0.9705 | 0.9738 | 117 | TwoPatterns | **1.0000** | 1.0000 | 1.0000 | 0.9964 | 0.9864 | **1.0000** |
| 52 | GunPoint | 0.9900 | 0.9950 | **1.0000** | 0.9900 | 0.9950 | 0.9850 | 118 | UMD | 0.9944 | **1.0000** | 0.9833 | 0.9889 | 0.9444 | 0.9889 |
| 53 | GunPointAgeSpan | **0.9868** | 0.9867 | 0.9667 | 0.9823 | 0.9801 | 0.9779 | 119 | UWaveGestureAll. | 0.9900 | 0.9891 | 0.9931 | 0.9864 | 0.9779 | **0.9933** |
| 54 | GunPointMale. | 0.9956 | 0.9934 | **0.9978** | 0.9956 | **0.9978** | 0.9956 | 120 | UWaveGestureX. | **0.9406** | 0.9270 | 0.9404 | 0.9087 | 0.8336 | 0.9241 |
| 55 | GunPointOld. | 0.9779 | 0.9778 | 0.9845 | 0.9712 | 0.9601 | **0.9889** | 121 | UWaveGestureY. | **0.9109** | 0.9044 | 0.9013 | 0.8881 | 0.7224 | 0.9087 |
| 56 | Ham | 0.9161 | 0.9116 | **0.9395** | 0.9349 | 0.9301 | 0.8882 | 122 | UWaveGestureZ. | **0.9170** | 0.8706 | 0.8704 | 0.8591 | 0.7722 | 0.8841 |
| 57 | HandOutlines | **0.9533** | 0.9365 | 0.9000 | 0.9343 | 0.9197 | 0.9248 | 123 | Wafer | **1.0000** | 0.9999 | 0.9996 | 0.9996 | 0.9990 | **1.0000** |
| 58 | Haptics | 0.7522 | 0.6770 | **0.7652** | 0.7349 | 0.7327 | 0.6744 | 124 | Wine | **0.8842** | 0.7763 | 0.7225 | 0.6036 | 0.5047 | 0.8115 |
| 59 | Herring | 0.7671 | 0.7351 | 0.7985 | 0.7828 | **0.8375** | 0.7991 | 125 | WordSynonyms | **0.9175** | 0.8961 | 0.9093 | 0.8873 | 0.8331 | 0.9094 |
| 60 | HouseTwenty | 0.9688 | **0.9750** | 0.9550 | 0.9698 | 0.9375 | 0.9688 | 126 | Worms | 0.8453 | 0.8304 | 0.8305 | 0.8458 | **0.8583** | 0.7834 |
| 61 | InlineSkate | **0.8077** | 0.6908 | 0.7169 | 0.7246 | 0.7062 | 0.7431 | 127 | WormsTwoClass | **0.8802** | 0.8786 | 0.8648 | 0.8307 | 0.8380 | 0.8726 |
| 62 | InsectEPGR. | 0.9968 | **1.0000** | 0.9968 | 0.9329 | 0.9744 | 0.9936 | 128 | Yoga | 0.9830 | 0.9788 | 0.9773 | 0.9703 | 0.9230 | 0.9609 |
| 63 | InsectEPGS. | **0.9963** | 0.9925 | 0.9925 | 0.9548 | 0.9329 | 0.9850 | | **Avg. Acc** | **0.9334** | 0.9123 | 0.9245 | 0.9022 | 0.8696 | 0.9164 |
| 64 | InsectWing. | 0.8882 | 0.8682 | 0.8786 | 0.8736 | 0.7955 | **0.8986** | | **Avg. Rank** | **2.04** | 3.04 | 2.75 | 3.74 | 5.02 | 3.23 |
| 65 | ItalyPowerDemand | 0.9891 | **0.9900** | 0.9836 | **0.9900** | 0.9726 | 0.9818 | | **Win** | **60** | 29 | 31 | 19 | 11 | 22 |
| 66 | LargeKitchenA. | 0.9627 | 0.9560 | **0.9640** | 0.9400 | 0.9107 | 0.9320 | | | | | | | | |

(e.g., the *ECG5000* dataset).

Table 12 presents the detailed test classification accuracy results across 18 UCR time series datasets for different values of the parameter $\eta$ during SoftShape sparsification. Table 13 provides the test classification accuracy results of SoftShape on 18 UCR time series datasets when varying the number of class-specific experts $k$ activated by the MoE router during intra-shape learning. Also, as baseline settings in Figure 3(c) and 3(d) of the main text, we report the parameter counts and inference times (in seconds) for SoftShape, MedFormer, TSLANet, InceptionTime, and MR-H on the *ChlorineConcentration* (4,307

samples, length 166) and *HouseTwenty* (159 samples, length 2,000) datasets. It is important to note that MR-H is executed on a CPU because its core modules do not use deep learning techniques, whereas other deep learning methods are run on an NVIDIA GeForce RTX 3090 GPU. As shown in Table 14, the differences in inference time between the deep learning methods are negligible. However, we observed that SoftShape demonstrates a slight advantage in inference time on the HouseTwenty dataset, which has a longer sequence length.

Table 12: The detailed test classification accuracy of sparse ratios on 18 UCR datasets. The best results are in **bold**.

| Dataset | $1-\eta=0$ | $1-\eta=0.1$ | $1-\eta=0.3$ | $1-\eta=0.5$ | $1-\eta=0.7$ | $1-\eta=0.9$ |
|---|---|---|---|---|---|---|
| ArrowHead | 0.9576 | 0.9576 | 0.9576 | 0.9435 | **0.9717** | 0.9435 |
| CBF | **1.0000** | **1.0000** | **1.0000** | **1.0000** | **1.0000** | **1.0000** |
| CricketX | 0.9365 | **0.9375** | 0.9331 | 0.9321 | 0.9211 | 0.9006 |
| DistalPhalanxOutlineAgeGroup | 0.9529 | 0.9510 | 0.9288 | 0.9518 | **0.9530** | 0.8861 |
| DistalPhalanxOutlineCorrect | 0.8951 | 0.8871 | **0.9055** | 0.9054 | 0.8780 | 0.8734 |
| ECG5000 | 0.9772 | 0.9802 | **0.9810** | 0.9796 | **0.9810** | 0.9696 |
| EOGVerticalSignal | 0.9145 | **0.9321** | 0.9310 | 0.9007 | 0.8730 | 0.8690 |
| EthanolLevel | **0.9214** | 0.9205 | 0.9205 | 0.9204 | 0.8985 | 0.8856 |
| Fish | 0.9742 | **0.9799** | 0.9656 | 0.9771 | 0.9628 | 0.9713 |
| GunPoint | 0.9850 | 0.9750 | 0.9850 | **0.9900** | 0.9850 | 0.9850 |
| InsectWingbeatSound | 0.8828 | 0.8760 | 0.8819 | **0.8882** | 0.8700 | 0.8860 |
| ItalyPowerDemand | 0.9891 | **0.9917** | 0.9846 | 0.9891 | 0.9837 | 0.9901 |
| MelbournePedestrian | 0.9511 | **0.9603** | 0.9545 | 0.9573 | 0.9391 | 0.9319 |
| MiddlePhalanxTW | 0.8340 | 0.8322 | 0.8268 | **0.8376** | 0.7474 | 0.7781 |
| MixedShapesRegularTrain | 0.9784 | 0.9781 | **0.9801** | 0.9757 | 0.9644 | 0.9620 |
| OSULeaf | 0.9414 | 0.9491 | 0.9413 | **0.9504** | 0.9436 | 0.9256 |
| Trace | **1.0000** | **1.0000** | **1.0000** | **1.0000** | **1.0000** | **1.0000** |
| WordSynonyms | **0.9382** | 0.9360 | 0.9294 | 0.9161 | 0.9095 | 0.9128 |
| Avg. Acc | 0.9461 | **0.9469** | 0.9448 | 0.9453 | 0.9323 | 0.9261 |
| Avg. Rank | 2.44 | **2.39** | 2.78 | 2.61 | 3.89 | 4.50 |
| Win | 4 | **7** | 5 | 6 | 5 | 2 |

## B.4. Hyperparameter Analysis

Table 15 provides the test classification accuracy results of SoftShape on 18 UCR time series datasets when varying the sliding fixed step size ($q$). Table 16 presents the detailed test classification accuracy results of SoftShape across 18 UCR time series datasets for different model depth ($L$) values. Table 17 presents the detailed test classification accuracies of SoftShape across 18 UCR datasets, evaluated with varying setting shape lengths $m$.

Tables 18, 19, 20, and 21 present the statistical test classification results of SoftShape on 18 selected UCR time series datasets. The evaluations are conducted under varying conditions, including different maximum numbers of experts for intra-shape learning, class-specific MoE expert networks, shared expert networks, and hyperparameter $\lambda$. For clarity and ease of analysis, only the statistical results of test accuracies are reported in Tables 18, 19, 20, and 21.

## B.5. Time Series Forecasting Results

For the forecasting task, we evaluate SoftShape on the *ETTh1*, *ETTh2*, *ETTm1*, and *ETTm2* datasets, using TS2Vec (Yue et al., 2022), TimesNet (Wu et al., 2023a), PatchTST (Nie et al., 2023), GPT4TS (Zhou et al., 2023), and iTransformer (Liu et al., 2024b) as baselines. Following the experimental setup of (Ma et al., 2024), we report the Mean Squared Error (MSE) and Mean Absolute Error (MAE) for all methods in Tables 22 and 23. Notably, we do not adjust hyperparameters for SoftShape across the four datasets, nor do we modify its shape embedding layer as done in (Nie et al., 2023). The results demonstrate that SoftShape outperforms TS2Vec and TimesNet, highlighting its potential for time series forecasting tasks.

Table 13: The detailed test accuracy of the number of activated experts on 18 UCR datasets. The best results are in **bold**.

| Dataset | $k$=1 | $k$=2 | $k$=3 | $k$=4 |
|---|---|---|---|---|
| ArrowHead | **0.9435** | 0.9324 | 0.9196 | 0.9147 |
| CBF | **1.0000** | **1.0000** | **1.0000** | **1.0000** |
| CricketX | 0.9321 | **0.9487** | 0.9334 | 0.9475 |
| DistalPhalanxOutlineAgeGroup | 0.9518 | **0.9665** | 0.9221 | 0.9518 |
| DistalPhalanxOutlineCorrect | **0.9054** | 0.8517 | 0.8574 | 0.8517 |
| ECG5000 | 0.9796 | **0.9840** | 0.9792 | 0.9640 |
| EOGVerticalSignal | 0.9007 | 0.9381 | **0.9490** | 0.9353 |
| EthanolLevel | 0.9204 | **0.9264** | 0.9054 | 0.9164 |
| Fish | **0.9771** | 0.9513 | 0.9571 | 0.9656 |
| GunPoint | **0.9900** | 0.9850 | 0.9850 | 0.9850 |
| InsectWingbeatSound | **0.8882** | 0.8782 | 0.8732 | 0.8659 |
| ItalyPowerDemand | **0.9891** | 0.9809 | 0.9809 | 0.9873 |
| MelbournePedestrian | 0.9573 | 0.9581 | 0.9568 | **0.9601** |
| MiddlePhalanxTW | 0.8376 | 0.8429 | 0.8231 | **0.8719** |
| MixedShapesRegularTrain | **0.9757** | 0.9675 | 0.8629 | 0.9535 |
| OSULeaf | **0.9504** | 0.9302 | 0.9390 | 0.8963 |
| Trace | **1.0000** | **1.0000** | **1.0000** | **1.0000** |
| WordSynonyms | 0.9161 | 0.9580 | 0.9536 | **0.9625** |
| **Avg. Acc** | **0.9453** | 0.9444 | 0.9332 | 0.9405 |
| **Avg. Rank** | **1.89** | 1.94 | 2.78 | 2.39 |
| **Win** | **10** | 6 | 3 | 5 |

Table 14: Comparison of model parameter counts and inference times.

| Methods | # Parameters | # Inference Time (Seconds) | |
|---|---|---|---|
| | | ChlorineConcentration | HouseTwenty |
| MR-H | - | 6.60 | 2.97 |
| InceptionTime | 387.7 K | 1.41 | 1.30 |
| TSLANet | 514.6 K | **1.37** | 1.29 |
| Medformer | 1360.6 K | 1.48 | 1.31 |
| **SoftShape (Ours)** | 472.5 K | 1.39 | **1.26** |

Table 15: The detailed test classification accuracy of sliding step size $q$ on 18 UCR datasets. The best results are in **bold**.

| Dataset | $q=1$ | $q=2$ | $q=3$ | $q=4$ | $q=m$ |
|---|---|---|---|---|---|
| ArrowHead | 0.8913 | 0.9289 | 0.9431 | 0.9435 | **0.9451** |
| CBF | **1.0000** | **1.0000** | **1.0000** | **1.0000** | **1.0000** |
| CricketX | 0.8936 | 0.9231 | 0.9103 | **0.9321** | 0.9295 |
| DistalPhalanxOutlineAgeGroup | 0.9147 | **0.9629** | 0.9610 | 0.9518 | 0.9184 |
| DistalPhalanxOutlineCorrect | 0.8688 | 0.8837 | 0.8540 | **0.9054** | 0.8632 |
| ECG5000 | 0.9672 | 0.9734 | 0.9822 | 0.9796 | **0.9824** |
| EOGVerticalSignal | 0.8551 | 0.9104 | 0.9408 | 0.9007 | **0.9533** |
| EthanolLevel | 0.8656 | 0.8736 | **0.9294** | 0.9204 | 0.8786 |
| Fish | 0.9485 | 0.9513 | 0.9713 | **0.9771** | 0.9399 |
| GunPoint | 0.9800 | 0.9850 | 0.9650 | **0.9900** | 0.9800 |
| InsectWingbeatSound | 0.7922 | 0.8563 | 0.8582 | **0.8882** | 0.8741 |
| ItalyPowerDemand | 0.9818 | 0.9873 | **0.9918** | 0.9891 | 0.9891 |
| MelbournePedestrian | 0.9538 | 0.9480 | 0.9518 | **0.9573** | 0.9431 |
| MiddlePhalanxTW | 0.7451 | 0.8267 | **0.8629** | 0.8376 | 0.7724 |
| MixedShapesRegularTrain | 0.9477 | 0.9272 | 0.8783 | **0.9757** | 0.9157 |
| OSULeaf | 0.9300 | **0.9774** | 0.9525 | 0.9504 | 0.9204 |
| Trace | **1.0000** | **1.0000** | **1.0000** | **1.0000** | **1.0000** |
| WordSynonyms | 0.8310 | 0.9017 | **0.9172** | 0.9161 | 0.8261 |
| **Avg. Acc** | 0.9092 | 0.9343 | 0.9372 | **0.9453** | 0.9240 |
| **Avg. Rank** | 3.83 | 2.78 | 2.44 | **1.78** | 2.94 |
| **Win** | 2 | 4 | 5 | **9** | 5 |

Table 16: The detailed test classification accuracy of model depth $L$ on 18 UCR datasets. The best results are in **bold**.

| Dataset | $L=1$ | $L=2$ | $L=3$ | $L=4$ | $L=5$ | $L=6$ |
|---|---|---|---|---|---|---|
| ArrowHead | **0.9437** | 0.9435 | 0.9390 | 0.9155 | 0.9155 | 0.9251 |
| CBF | **1.0000** | **1.0000** | 0.9989 | 0.9978 | 0.9989 | 0.9989 |
| CricketX | 0.8962 | 0.9321 | 0.9706 | **0.9783** | 0.9501 | 0.9539 |
| DistalPhalanxOutlineAgeGroup | 0.9629 | 0.9518 | 0.9740 | 0.9389 | **0.9741** | 0.9574 |
| DistalPhalanxOutlineCorrect | 0.9054 | 0.9054 | 0.8917 | 0.8962 | 0.8811 | **0.9077** |
| ECG5000 | **0.9830** | 0.9796 | 0.9770 | 0.9750 | 0.9810 | 0.9790 |
| EOGVerticalSignal | 0.8038 | 0.9007 | 0.9187 | **0.9422** | 0.9083 | 0.8979 |
| EthanolLevel | 0.8189 | **0.9204** | 0.8905 | 0.8975 | 0.9005 | 0.8945 |
| Fish | 0.9428 | 0.9771 | 0.9914 | 0.9914 | 0.9685 | **0.9942** |
| GunPoint | 0.9800 | **0.9900** | **0.9900** | **0.9900** | **0.9900** | **0.9900** |
| InsectWingbeatSound | **0.9237** | 0.8882 | 0.8905 | 0.8968 | 0.8764 | 0.8777 |
| ItalyPowerDemand | 0.9854 | 0.9891 | **0.9982** | 0.9864 | 0.9937 | 0.9882 |
| MelbournePedestrian | **0.9589** | 0.9573 | 0.9488 | 0.9567 | 0.9531 | 0.9578 |
| MiddlePhalanxTW | 0.8070 | **0.8376** | 0.7799 | 0.8052 | 0.7979 | 0.8124 |
| MixedShapesRegularTrain | 0.9282 | 0.9757 | 0.9778 | **0.9781** | 0.9689 | 0.9747 |
| OSULeaf | 0.9368 | 0.9504 | 0.9503 | 0.9414 | **0.9571** | 0.9233 |
| Trace | **1.0000** | **1.0000** | **1.0000** | **1.0000** | **1.0000** | **1.0000** |
| WordSynonyms | 0.8344 | 0.9161 | 0.9062 | 0.8974 | 0.8951 | **0.9228** |
| **Avg. Acc** | 0.9228 | **0.9453** | 0.9441 | 0.9436 | 0.9395 | 0.9420 |
| **Avg. Rank** | 3.72 | **2.61** | 3.06 | 3.33 | 3.44 | 3.06 |
| **Win** | **6** | 5 | 3 | 5 | 4 | 4 |

Table 17: The detailed test classification accuracy of shape length $m$ on 18 UCR datasets. The best results are in **bold**.

| Dataset | Val-Select | Fixed-8 | Multi-Seq |
|---|---|---|---|
| ArrowHead | 0.9435 | 0.9435 | **0.9436** |
| CBF | **1.0000** | **1.0000** | **1.0000** |
| CricketX | **0.9321** | 0.9282 | 0.9320 |
| DistalPhalanxOutlineAgeGroup | 0.9518 | **0.9611** | 0.9351 |
| DistalPhalanxOutlineCorrect | 0.9054 | 0.9065 | **0.9156** |
| ECG5000 | 0.9796 | 0.9610 | **0.9810** |
| EOGVerticalSignal | **0.9007** | 0.8759 | 0.8952 |
| EthanolLevel | 0.9204 | 0.8785 | **0.9303** |
| Fish | **0.9771** | 0.9686 | 0.9697 |
| GunPoint | **0.9900** | **0.9900** | **0.9900** |
| InsectWingbeatSound | 0.8882 | 0.9010 | **0.9055** |
| ItalyPowerDemand | **0.9891** | 0.9855 | 0.9867 |
| MelbournePedestrian | 0.9573 | 0.9641 | **0.9647** |
| MiddlePhalanxTW | 0.8376 | 0.8352 | **0.8414** |
| MixedShapesRegularTrain | 0.9757 | 0.9808 | **0.9849** |
| OSULeaf | **0.9504** | 0.9392 | 0.9444 |
| Trace | **1.0000** | **1.0000** | **1.0000** |
| WordSynonyms | **0.9161** | 0.9028 | 0.9139 |
| **Avg. Acc** | 0.9453 | 0.9401 | **0.9463** |
| **Avg. Rank** | 1.72 | 2.28 | **1.44** |
| **Win** | 9 | 4 | **11** |

Table 18: The statistical test classification accuracy of the total number of experts $\hat{C}$ on 18 UCR datasets. And $C$ denotes the total number of classes within each time series dataset. The best results are in **bold**.

| Metric | $\hat{C}=C/2$ | $\hat{C}=C$ | $\hat{C}=2C$ |
|---|---|---|---|
| Avg. Acc | 0.9215 | **0.9453** | 0.9408 |
| Avg. Rank | 2.33 | **1.39** | 1.83 |
| Win | 4 | **12** | 6 |

Table 19: The statistical test classification accuracy across different networks used as MoE experts on 18 UCR datasets. TSLANet (ICB) indicates that the ICB block within TSLANet is used as the MoE expert. The best results are in **bold**.

| Metric | FCN | TSLANet (ICB) | Original MoE expert |
|---|---|---|---|
| Avg. Acc | 0.8652 | 0.9049 | **0.9453** |
| Avg. Rank | 2.56 | 1.83 | **1.44** |
| Win | 4 | 7 | **10** |

Table 20: The statistical test classification accuracy and parameter counts across different networks used as the shared expert on 18 UCR datasets. The best results are in **bold**.

| Metric | Transformer | MLP | Inception |
|---|---|---|---|
| # Parameters | 422.5 K | 157.8 K | 179.5 K |
| Avg. Acc | 0.8239 | 0.8103 | **0.9453** |
| Avg. Rank | 2.22 | 2.44 | **1.11** |
| Win | 4 | 1 | **16** |

Table 21: The statistical test classification accuracy across different $\lambda$ on 18 UCR datasets. The best results are in **bold**.

| Metric | $\lambda = 0.0001$ | $\lambda = 0.001$ | $\lambda = 0.01$ | $\lambda = 0.1$ | $\lambda = 1$ | $\lambda = 10$ | $\lambda = 100$ |
|---|---|---|---|---|---|---|---|
| Avg. Acc | 0.9417 | 0.9453 | **0.9471** | 0.9374 | 0.9394 | 0.9438 | 0.9431 |
| Avg. Rank | 3.00 | **2.50** | 2.61 | 4.11 | 3.72 | 3.72 | 3.56 |
| Win | 6 | 6 | **8** | 4 | 4 | 5 | 7 |

Table 22: The detailed test forecasting performance (Mean Squared Error, MSE) of different methods. The best results are in **bold**.

| Methods | | TS2Vec | TimesNet | PatchTST | GPT4TS | iTransformer | **SoftShape (Ours)** |
|---|---|---|---|---|---|---|---|
| Metric | | MSE | MSE | MSE | MSE | MSE | MSE |
| ETTh1 | 24 | 0.5952 | 0.3485 | 0.3890 | 0.3102 | **0.3065** | 0.5239 |
| | 48 | 0.6316 | 0.3991 | 0.4362 | 0.3529 | **0.3451** | 0.5160 |
| | 168 | 0.7669 | 0.4846 | 0.5304 | 0.4600 | **0.4307** | 0.4564 |
| | 336 | 0.9419 | 0.5583 | 0.5928 | 0.5167 | **0.4889** | 0.4992 |
| | 720 | 1.0948 | 0.5886 | 0.6123 | 0.6878 | 0.5184 | 0.5099 |
| | Avg. Value | 0.8061 | 0.4758 | 0.5121 | 0.4655 | **0.4179** | 0.5011 |
| | Avg. Rank | 6 | 3.4 | 4.4 | 3 | **1.2** | 3 |
| ETTh2 | 24 | 0.4478 | 0.2129 | 0.2176 | 0.2009 | **0.1831** | 0.2542 |
| | 48 | 0.6460 | 0.2824 | 0.2722 | 0.2738 | **0.2413** | 0.2808 |
| | 168 | 1.7771 | 0.4461 | 0.4092 | 0.4436 | **0.3725** | 0.3824 |
| | 336 | 2.1157 | 0.4875 | 0.4670 | 0.5011 | **0.4368** | 0.4429 |
| | 720 | 2.5823 | 0.5193 | 0.4721 | 0.5389 | **0.4479** | 0.4722 |
| | Avg. Value | 1.5138 | 0.3896 | 0.3676 | 0.3917 | **0.3363** | 0.3665 |
| | Avg. Rank | 6 | 4.2 | 2.8 | 3.8 | **1** | 3.2 |
| ETTm1 | 24 | 0.1970 | 0.2416 | 0.2522 | 0.2004 | 0.2251 | 0.2980 |
| | 48 | 0.2682 | 0.3194 | 0.3202 | 0.3006 | 0.3019 | 0.3197 |
| | 96 | 0.3735 | 0.3558 | 0.3553 | 0.3008 | 0.3393 | 0.3389 |
| | 288 | 0.7566 | 0.4411 | 0.4207 | 0.3712 | 0.4147 | 0.3953 |
| | 672 | 1.8217 | 0.6567 | 0.4878 | 0.4570 | 0.4880 | 0.4706 |
| | Avg. Value | 0.6834 | 0.4029 | 0.3672 | **0.3260** | 0.3538 | 0.3645 |
| | Avg. Rank | 4 | 4.6 | 4.4 | **1.4** | 3.2 | 3.4 |
| ETTm2 | 96 | 0.3502 | 0.1877 | 0.1876 | 0.1861 | **0.1830** | 0.1873 |
| | 192 | 0.5684 | 0.2748 | 0.2544 | 0.2624 | **0.2507** | 0.2545 |
| | 336 | 0.9589 | 0.3922 | 0.3173 | **0.3164** | 0.3179 | 0.3190 |
| | 720 | 2.5705 | 0.4477 | 0.4179 | 0.4246 | **0.4161** | 0.4248 |
| | Avg. Value | 1.1120 | 0.3256 | 0.2943 | 0.2974 | **0.2919** | 0.2964 |
| | Avg. Rank | 5.6 | 4.92 | 2.88 | 2.28 | **1.84** | 3.48 |

Table 23: The detailed test forecasting performance (Mean Absolute Error, MAE) of different methods. The best results are in **bold**.

| Methods | | TS2Vec | TimesNet | PatchTST | GPT4TS | iTransformer | **SoftShape (Ours)** |
|---|---|---|---|---|---|---|---|
| Metric | | MAE | MAE | MAE | MAE | MAE | MAE |
| ETTh1 | 24 | 0.5313 | 0.3873 | 0.4128 | 0.3626 | **0.3589** | 0.4817 |
| | 48 | 0.5566 | 0.4169 | 0.4379 | 0.3904 | **0.3818** | 0.4794 |
| | 168 | 0.6405 | 0.4670 | 0.4879 | 0.4661 | **0.4304** | 0.4481 |
| | 336 | 0.7334 | 0.5175 | 0.5193 | 0.4960 | **0.4602** | 0.4719 |
| | 720 | 0.8098 | 0.5287 | 0.5442 | 0.5858 | **0.4982** | 0.4951 |
| | Avg. Value | 0.6543 | 0.4635 | 0.4804 | 0.4602 | **0.4259** | 0.4752 |
| | Avg. Rank | 6 | 3.4 | 4.4 | 3 | **1.2** | 3 |
| ETTh2 | 24 | 0.5032 | 0.2929 | 0.3043 | 0.2917 | **0.2719** | 0.3344 |
| | 48 | 0.6184 | 0.3429 | 0.3388 | 0.3443 | **0.3120** | 0.3475 |
| | 168 | 1.0569 | 0.4361 | 0.4172 | 0.4476 | **0.3940** | 0.4042 |
| | 336 | 1.1759 | 0.4687 | 0.4610 | 0.4876 | **0.4433** | 0.4484 |
| | 720 | 1.3521 | 0.4893 | 0.4734 | 0.5146 | **0.4585** | 0.4721 |
| | Avg. Value | 0.9413 | 0.4060 | 0.3989 | 0.4172 | **0.3759** | 0.4013 |
| | Avg. Rank | 6 | 3.6 | 3 | 4.2 | **1** | 3.2 |
| ETTm1 | 24 | 0.3179 | 0.3115 | 0.3186 | 0.2769 | 0.2967 | 0.3564 |
| | 48 | 0.3784 | 0.3637 | 0.3593 | 0.3344 | 0.3474 | 0.3644 |
| | 96 | 0.4496 | 0.3864 | 0.3793 | 0.3519 | 0.3740 | 0.3751 |
| | 288 | 0.6672 | 0.4304 | 0.4162 | 0.3989 | 0.4163 | 0.4072 |
| | 672 | 1.0452 | 0.5324 | 0.4527 | 0.4483 | 0.4577 | 0.4512 |
| | Avg. Value | 0.5717 | 0.4049 | 0.3852 | **0.3621** | 0.3784 | 0.3908 |
| | Avg. Rank | 5.6 | 4.4 | 3.6 | **1** | 2.8 | 3.6 |
| ETTm2 | 96 | 0.4381 | 0.2678 | 0.2728 | 0.2781 | **0.2652** | 0.2744 |
| | 192 | 0.5732 | 0.3220 | 0.3140 | 0.3271 | **0.3101** | 0.3167 |
| | 336 | 0.7532 | 0.3830 | 0.3527 | 0.3650 | **0.3524** | 0.3572 |
| | 720 | 1.2483 | 0.4244 | **0.4075** | 0.4331 | 0.4090 | 0.4186 |
| | Avg. Value | 0.7532 | 0.3493 | 0.3367 | 0.3508 | **0.3342** | 0.3418 |
| | Avg. Rank | 5.92 | 3.88 | 2.32 | 4 | **1.56** | 3.32 |

