# OpenReview forum: "Learning Soft Sparse Shapes for Efficient Time-Series Classification"
_ICML.cc/2025/Conference — ICML 2025 spotlightposter_

### Official Review · Reviewer_XKP7 · 2025-02-24

**Overall Recommendation:** 3

**Summary:**

The paper focuses on the univariate time series classification problem. It proposes a method called Soft Sparse Shapes (SoftShape) model. The model has two major components: 1) a soft shape sparsification module, and 2) a soft shape learning block. The soft shape sparsification module categorizes the input tokens into two groups based on their attention scores. The tokens with higher attention scores are passed to the next layer, while those with lower attention scores are added together. In the second module, the tokens are processed in two different paths. The first path utilizes a mixture of experts to process the location information, while the second path uses a time series model to capture the global information. The authors have conducted experiments to evaluate the performance of the proposed method.

**Claims And Evidence:**

1. The proposed method provides excellent performance for time series classification. This claim is mostly supported by the experimental results.
2. The design of both modules is important for the resulting performance. This claim is mostly supported by the ablation study. It would be better if the authors provided examples of how each design resolves problems that other models failed to address through theoretical analysis or demonstrated it with simple toy data.

**Essential References Not Discussed:**

To the best of my knowledge, there are no essential references missing from the paper's discussion.

**Experimental Designs Or Analyses:**

One minor drawback is that the ablation study is only conducted on a subset of datasets. Another drawback is that the subsequence length $m$ is not included in the hyper-parameter analysis.

**Methods And Evaluation Criteria:**

The proposed methods and evaluation criteria make sense for the problem at hand.

**Other Comments Or Suggestions:**

The definition of sparsity rate $\eta$ is not clear. It would be better to properly define it, so readers understand the role of this hyper-parameter. How does the setting of $\eta$ affect another variable $Num$?

**Other Strengths And Weaknesses:**

- Strengths: The proposed method is novel and outperforms prior methods.
- Weaknesses: The paper heavily relies on performance to motivate the proposed method. It would be better if some kind of theoretical analysis or demonstration with toy data could help the reader understand the merit of the proposed modules.

**Questions For Authors:**

1. In Equation 1, the set of all possible shapelets consists of subsequences of different lengths. However, I think the proposed method only uses one subsequence length. Can you explain the trade-off associated with this design discrepancy?
2. For the soft shape sparsification module, can you provide more details about how the proposed method differs from the standard self-attention mechanism, and the pros and cons of these designs (proposed versus standard)?
3. What is the definition of $\eta$, and what is its relationship with $Num$?
4. In the "with linear shape" setting in the ablation study, can you explain more about how the linear shape mechanism is implemented?
5. What is the effect of the hyper-parameter $m$ on the classification performance?

**Relation To Broader Scientific Literature:**

The paper proposes a novel time series classification model for univariate time series data.

**Theoretical Claims:**

This paper does not present any proofs or theoretical analyses.

---

> ### Author Rebuttal · Authors · 2025-03-31
>
> **W1: The claim is mostly supported by the ablation study. It would be better if the authors provided examples through theoretical analysis or demonstrated it with simple toy data.**
>
> **A**: SoftShape include:
>
> 1)	Soft Shape Sparsification
>
> - To justify its design, we provide toy real-world data (Figure 1 in the main text) with two classes: a robotic dog walking on carpet vs. cement. Existing methods discard shapes of data in a hard way, losing valuable information, whereas SoftShape uses attention-based soft weighting to preserve key shapes.
>
> 2)	Soft Shape Learning Block
>
> - Intra-shape learning: We use the MoE router to learn local class-discriminative shape features. Theoretical support: Theorem 4.2 in [1] states that the MoE router learns cluster-center features, simplifying complex issues into classification problems. Lemma 4.1 in [2] shows that MoE routers group class-discriminative patterns while filtering irrelevant patches.
>
> - Inter-shape learning: We use a shared expert to capture global temporal patterns via multi-scale kernels.  t-SNE visualizations of toy data (Figures 5 & 8 in the main text) show improved separation of mixed-class clusters over intra-shape learning.
>
> [1] Towards understanding the mixture-of-experts layer in deep learning. NIPS, 2022.
>
> [2] Patch-level routing in mixture-of-experts is provably sample-efficient for convolutional neural networks. ICML, 2023.
>
> ---
>
> **W2: The ablation study is only conducted on a subset of datasets.**
>
> **A**: We performed ablation on 128 UCR datasets, with findings consistent with the subset of datasets.  Statistical results:
>
> | Method  | Avg. Rank | P-value|
> |----------------------|-----------|------------|
> | w/o Soft Sparse| 3.04| 4.06E-03   |
> | w/o Intra  | 2.75| 4.69E-03|
> | w/o Inter | 3.74| 3.90E-07|
> | w/o Intra & Inter   | 5.02  | 8.95E-16   |
> | with Linear Shape   | 3.23| 4.69E-03   |
> | SoftShape| 2.04| -  |
>
> ---
>
> **W3 & Q1: The subsequence length $m$ is not included in the hyper-parameter analysis. While all shapelets include varying lengths, SoftShape uses only one. Can you explain the trade-off of this design?**
>
> **A**: Using variable-length subsequences as input data raises computational complexity.  Also, due to varying sequence lengths in UCR sub-datasets, larger $m$ may suit longer sequences, while smaller $m$ may be better in other cases. Following [3,4], we determine a fixed $m$ via a validation set with fewer training epochs. Experiments with a chosen $m$ reduce computation time and ensure good results, as seen in the answer to Q5.
>
> [3] Learning time-series shapelets. KDD, 2014.
>
> [4] CNN kernels can be the best shapelets. ICLR, 2024.
>
> ---
>
> **Q2: Can you detail how the attention mechanism used in the paper differs from self-attention and compare their pros and cons?**
>
> **A**: As noted in [5], the key difference is that gated attention (used in our method) assumes instance independence, enabling key instance identification via labels, whereas self-attention captures dependencies across all instances without using labels.
>
> | | Pros| Cons |
> |------------|--------------------------|------------------------------|
> | Gated Attnention | Linear complexity with instance counts. | Weak in capturing instance relationships|
> | Self-Attention |  Captures dependencies across instances.  |Higher complexity with instance counts|
>
> However, gated attention implicitly captures shape relationships via labels. Also, we use a shared expert to learn temporal patterns among shapes.
>
> [5] Attention-based deep multiple instance learning. ICML, 2018.
>
> ---
>
> **Q3: What is the definition of $\eta$, and what is its relationship with $Num$?**
>
> **A**: For a time series of length $T$, the number of subsequences of length $m$ is $J = \frac{T-m}{q}+1$, where $q$ is the sliding window step.  $\eta$ represents the ratio of top subsequences based on attention scores. The number of sparsified subsequences $\text{Num} = J \times \eta+1$.
>
> ---
>
> **Q4: Can you explain the implementation of the linear shape in the ablation study?**
>
> **A**: Replace the 1D CNN in Eq. (2) with a linear layer to convert each subsequence of length $m$ into an embedding of dimension $d$.
>
> ---
>
> **Q5: What is the effect of the hyper-parameter $m$ on the classification performance?**
>
> **A**: We conducted three experiments for $m$:
>
> 1. **Val-Select**: Choose a fixed m for one SoftShape model using the validation set.
> 2. **Fixed-8**: Set m = 8  for one SoftShape model.
> 3. **Multi-Seq**: Use fixed lengths (8, 16, 32) in parallel three SoftShape models, with residual fusion for classification.
>
> | Model | Avg. Rank | P-value   |
> |-----------------|-----------|-----------|
> | Val-Select      | 1.72      | 2.88E-01  |
> | Fixed-8         | 2.28      | 4.68E-02  |
> | Multi-Seq       | 1.44      | -         |
>
> We found no significant performance difference between Val-Select and Multi-Seq (p-value > 0.05). The Val-Select model also has fewer parameters and lower runtime.

---

> > ### Comment · Reviewer_XKP7 · 2025-04-02
> >
> > W1: Please see my follow-up questions below.
> > 1. Soft Shape Sparsification: From Figure 1, it is difficult to discern the benefit of soft versus hard shapelets, as the hard shapelet also appears capable of discriminating between the two classes.
> > 2. Soft Shape Learning Block:
> >   - Intra-shape learning: concern resolved.
> >   - Inter-shape learning: both CBF and TwoPatterns are relatively simple datasets, as DTW (learned_w) + 1NN achieves close to 100% accuracy according to [1]. Consequently, I'm not fully convinced by the visualizations presented.
> >
> > Overall, my remaining concerns regarding W1 primarily relate to the visualizations. The theoretical aspects have been clarified.
> >
> > W2: Concern resolved.
> >
> > W3 & Q1: My major concern regarding this weakness/question is that the search/learning space for shapelets should include subsequences of different lengths. However, as you mentioned, only one subsequence length is used in the learning process. There appears to be a misalignment between section 3.2 and the proposed method, which I believe should be addressed in the paper. Nevertheless, my concerns are partially answered in your response to Q5.
> >
> > Q2: Concern resolved.
> >
> > Q3: Concern resolved.
> >
> > Q4: Concern resolved.
> >
> > Q5: Concern resolved. I feel this result suggests a potential future direction: learning soft shapelets with multiple values of m within a single model.
> >
> > [1] https://www.cs.ucr.edu/~eamonn/time_series_data_2018/

---

> > > ### Author Response · Authors · 2025-04-04
> > >
> > > We sincerely appreciate your timely response and insightful questions.
> > >
> > > ---
> > >
> > > **W1-1: It is difficult to discern the benefit of soft versus hard shapelets, as the hard shapelet also appears capable of discriminating between the two classes.**
> > >
> > > **A**: In Figures 1(b) and (d) in the main text, while both hard and soft shapelets distinguish the two classes in a rough way, they differ in key details.
> > >
> > > - **Hard Shapelets**: Assign equal weight to selected subsequences, ignoring some overlapping segment information between classes within these subsequences. It captures local patterns of the selected subsequences but fails to retain the global patterns of the raw time series due to exclusion of unselected subsequences.
> > >
> > > - **Soft Shapelets**: Use attention scores to highlight key subsequences while integrating unselected ones without losing informantion. This enables Soft to capture the local patterns of selected subsequences while capturing global patterns of the raw time series through a shared expert.
> > >
> > > Figures 1(a) and (c) illustrate variations within the same class. In such cases, certain walking on carpet and cement samples may exhibit minimal class differences in selected hard shapelets. The updated Figure 1 compares hard and soft shapelets for both classes, please refer to [anonymous link](https://anonymous.4open.science/r/SoftFig-0a05).
> > >
> > > In the updated Figure 1 (b), Hard struggles to distinguish the third and fourth subsequences, omitting many informative ones (unmarked in red). In contrast, the updated Figure 1 (d) shows Soft assigns low attention scores to these regions, enhancing discrimination.
> > >
> > > ---
> > >
> > > **W1-2: CBF and TwoPatterns are relatively simple datasets, as DTW (learned_w) + 1NN achieves close to 100% accuracy.**
> > >
> > > **A**: For t-SNE visualization, datasets with low learning difficulty (high test accuracy) more clearly distinguish class clusters. Conversely, high-difficulty datasets produce indistinct clusters across methods, reducing the effectiveness of visual analysis.
> > >
> > > - Low learning difficulty (high accuracy)
> > >   - Better class cluster separation in t-SNE visualization.
> > >   - Example: **CBF** & **TwoPatterns** (Figures 5, 8) show clear distinctions across shape embeddings, as follows:
> > >     1. Output shape (best separation)
> > >     2. Inter-shape  (second best)
> > >     3. Intra-shape
> > >     4. Raw input shape (worst separation)
> > >
> > > - High learning difficulty (low accuracy or many classes)
> > >   - t-SNE shows indistinct clusters across methods.
> > >   - Example:
> > >     - **Haptics** (DTW+1NN accuracy: 0.38)
> > >     - **FiftyWords** (50 classes, DTW+1NN accuracy: 0.69)
> > >   - Clusters are harder to differentiate but show slight similarities to CBF & TwoPatterns (new Figures 2, 3).
> > >
> > > - Moderate learning difficulty (fewer classes, mid-range accuracy)
> > >   - Example: **ECG200** (DTW+1NN accuracy: 0.77).
> > >   - Visualization in the new Figure 4 aligns with Figures 5 & 8 in the main text but with less pronounced clusters.
> > >
> > > [See new Figures](https://anonymous.4open.science/r/SoftFig-0a05).
> > >
> > > ---
> > >
> > > **W3 & Q1: The search/learning space for shapelets should include subsequences of different lengths. Only one subsequence length is used in the learning process, which appears to be a misalignment between Section 3.2 and the proposed method.**
> > >
> > > **A**: The detailed explanations are as follows:
> > >
> > > **1. Shapelet Discovery**
> > >
> > > The shapelets proposed by [1] (Section 3.2) show that nearest neighbor algorithms are computationally expensive due to exhaustive subsequence searches of varying lengths. To enhance efficiency, [1] uses early abandon and entropy pruning to discard unpromising subsequences.
> > >
> > > **2. Gradient-Based Shapelet Learning**
> > >
> > > The gradient-based approach in [2] is efficient than nearest neighbor methods but struggles with variable-length data, requiring samples to be transformed into equal-length matrices for gradient optimization. To reduce learning time, studies [2,3,4] fix length m using a validation set. Similarly, we treat m as a hyperparameter, enabling flexible subsequence length selection before model training.
> > >
> > > **3. Learning Soft Shapelets**
> > >
> > > We define learning soft shapelets as using gradient-based algorithms to assign attention scores to fixed-length subsequences, capturing intra- and inter-shape discriminative patterns. Unlike prior shapelet methods, SoftShape uses a shared expert with multi-scale convolutional kernels (i.e., 10,20,40), enabling the model to capture dependencies across various subsequence lengths and mitigate fixed-length m limitations.
> > >
> > > As noted in Q5, an open challenge is learning soft shapelets with multiple m values within a single model. This remains unsolved in [2,3,4] and is a key focus of our future work.
> > >
> > > [1] Time series shapelets: a new primitive for data mining. KDD, 2009.
> > >
> > > [2] Learning time-series shapelets. KDD, 2014.
> > >
> > > [3] CNN kernels can be the best shapelets. ICLR, 2024.
> > >
> > > [4] Shapeformer: Shapelet transformer for multivariate time series classification. KDD, 2024.

---

### Official Review · Reviewer_6rFK · 2025-03-06

**Overall Recommendation:** 4

**Summary:**

This paper presents Soft Sparse Shapes (SoftShape) for efficient time-series classification. It introduces soft shape sparsification to improve training efficiency by converting subsequences into soft representations. The model further enhances performance by employing a mixture of experts for intra-shape and inter-shape temporal pattern learning. Through extensive experiments on 128 UCR time series datasets, SoftShape outperforms existing methods with significantly better accuracy and training efficiency.

**Claims And Evidence:**

Yes. The claims made in this paper are supported by extensive experiments on the UCR time series dataset, demonstrating SoftShape's superiority over 15 baseline methods. The P-values from the Wilcoxon signed-rank test confirm the statistical significance of SoftShape's performance improvements.

**Essential References Not Discussed:**

The paper thoroughly cites relevant works in time-series classification, particularly those on shapelet methods and MoE architectures.

**Experimental Designs Or Analyses:**

Yes. The paper employs a comprehensive experimental setup, comparing SoftShape with 15 baseline methods on 128 UCR datasets. The experimental design, including five-fold cross-validation, is sound and valid for evaluating the effectiveness of SoftShape. The use of test accuracy and P-values as evaluation metrics adds rigor to the analysis.

**Methods And Evaluation Criteria:**

Yes. The proposed method, SoftShape, makes sense for time series classification as it addresses the issue of sparse shape learning by using a soft sparsification process that preserves important subsequences. The evaluation criteria, mainly test accuracy, are suitable for demonstrating the method’s effectiveness on the benchmark UCR datasets.

**Other Comments Or Suggestions:**

The paper is well-written, and the proposed approach is highly innovative. It would be interesting to see how SoftShape scales to datasets of varying sizes and how its performance may change as the complexity of the dataset increases.

**Other Strengths And Weaknesses:**

**Strengths**:
SoftShape presents a novel approach to time-series classification, offering both interpretability and efficiency. A key strength of SoftShape is its ability to handle shape sparsification in a "soft" manner, which allows it to retain valuable discriminative features while improving computational efficiency.

**Weaknesses**:
One potential weakness of the SoftShape model is its reliance on fixed-length patch partitions for dividing the input time series. While this approach can be efficient, it may result in the loss of important discriminative information. Specifically, longer discriminative subsequences may be split across multiple fixed-length patches, potentially disrupting important temporal patterns that are crucial for classification. Additionally, shorter discriminative subsequences could be mixed with non-discriminative segments, which may reduce the model's ability to accurately capture distinguishing features. This could potentially degrade the overall classification performance, particularly when working with time series data that contains both long and short discriminative patterns.

**Questions For Authors:**

1.	It would be helpful to further explain the statement in Section 2.3 regarding how the shared expert enhances the discriminative power of shape embeddings. A more detailed explanation of this mechanism could strengthen the understanding of how it contributes to the overall performance of the model.
2.	Given that shapelets can vary in length, the use of fixed patch lengths in SoftShape may limit the ability to capture discriminative features that span different lengths. Exploring the possibility of variable patch lengths might enhance the model’s flexibility and potentially improve its performance. It would be valuable to discuss this further.
3.	The shared expert is currently not implemented with the same lightweight linear model as the class-specific expert. It might be worth considering whether using a lightweight linear model for the shared expert could simplify the architecture without negatively affecting performance.
4.	It would be useful to understand how λ affects the results and what range of values was tested during experimentation. This could provide valuable insight into the model’s sensitivity to this hyperparameter.
5.	While the paper compares SoftShape against various non-shapelet-based methods, it could be valuable to also include a comparison with other shapelet-based methods mentioned in Section 2.2.

**Relation To Broader Scientific Literature:**

The key contributions of SoftShape relate to previous work in time-series classification, particularly shapelet-based methods and mixture-of-experts (MoE) architectures. The paper builds on these techniques by introducing a soft shape sparsification mechanism and leveraging MoE to enhance performance, distinguishing it from prior work in time-series classification.

**Theoretical Claims:**

No theoretical claims or proofs in this paper.

---

> ### Author Rebuttal · Authors · 2025-03-31
>
> **W1 & Q2: The SoftShape model is reliance on fixed-length patch partitions for dividing the input time series, which may result in the loss of important discriminative information. Exploring the possibility of variable patch lengths might enhance the model’s flexibility and potentially improve its performance. It would be valuable to discuss this further.**
>
> **A**: Using shapes (or patches) of varying lengths as model inputs is intuitive but increases computational complexity and requires preprocessing for standardization. To address this, we follow [1,2] by selecting an optimal fixed-length shape via the validation set, balancing efficiency and performance. Also, SoftShape employs a shared expert with multi-scale convolutional kernels to capture dependencies across shapes, mitigating the limitations of fixed-length patches.
>
> Further, we conducted further experiments on the 18 selected UCR datasets. The settings and results are as follows.
>
> 1. **Val-Select**: Choose a fixed m for one SoftShape model using the validation set.
> 2. **Fixed-8**: Set m = 8  for one SoftShape model.
> 3. **Multi-Seq**: Use fixed lengths (8, 16, 32) in parallel three SoftShape models, with residual fusion for classification.
>
> | Model | Avg. Rank | P-value   |
> |-----------------|-----------|-----------|
> | Val-Select      | 1.72      | 2.88E-01  |
> | Fixed-8         | 2.28      | 4.68E-02  |
> | Multi-Seq       | 1.44      | -         |
>
> We found no significant performance difference between Val-Select and Multi-Seq (p-value > 0.05). The Val-Select model also has fewer parameters and lower runtime.
>
>
> [1] Learning time-series shapelets. KDD, 2014.
>
> [2] CNN kernels can be the best shapelets. ICLR, 2024.
>
> ---
>
> **Q1: A more detailed explanation of the shared expert mechanism could strengthen the understanding of how it contributes to the overall performance of the model.**
>
> **A**: For intra-shape learning, MoE treats each shape within a time series as an independent sample, making it difficult to capture dependencies between shapes. Additionally, while learning intra-shape temporal patterns helps extract local discriminative features, it struggles to capture global temporal patterns crucial for classification. To address this, we introduce a shared expert that converts soft shapes within a sample into a sequence and leverages Inception’s multi-scale convolutions to learn global temporal dependencies.
>
> ---
>
> **Q3: The shared expert is currently not implemented with the same lightweight linear model as the class-specific expert. It might be worth considering whether using a lightweight linear model for the shared expert could simplify the architecture without negatively affecting performance.**
>
> **A**: We replaced the shared expert from Inception (hidden size 128) with MLP (hidden size 256) and Transformer (hidden size 512) and conducted experiments on 18 UCR datasets.
>
> | Model       | Avg. Rank | Parameters |
> |------------|-----------|------------|
> | Inception  | 1.11      | 179.5 K    |
> | MLP        | 2.44      | 157.8 K    |
> | Transformer | 2.22     | 422.5 K    |
>
> The Parameters count represents the inter-shape module (shared expert). The results show that Inception and MLP have similar parameter sizes, both significantly smaller than Transformer. This suggests that using Inception as the shared expert provides a lightweight architecture while achieving better classification performance than MLP and Transformer.
>
> ---
>
> **Q4: It would be useful to understand how λ affects the results and what range of values was tested during experimentation.**
>
> **A**: We analyzed the impact of different λ values on SoftShape's classification performance using 18 selected UCR datasets.
>
> | λ         | 0.0001 | 0.001 | 0.01 | 0.1  | 1    | 10   | 100  |
> |-----------|--------|-------|------|------|------|------|------|
> | Avg. Rank | 3.00   | 2.50  | 2.61 | 4.11 | 3.72 | 3.72 | 3.56 |
>
> The results show that λ = 0.001 achieves the best performance, with minimal difference from λ = 0.01.
>
> ---
>
> **Q5: It could be valuable to also include a comparison with other shapelet-based methods mentioned in Section 2.2.**
>
> **A**: We selected ShapeConv [2] and ShapeFormer [3] as baselines. The comparison between ShapeConv and SoftShape on 128 UCR time series datasets is shown below:
>
> | Method     | Avg. Rank | P-value  |
> |------------|----------|----------|
> | ShapeConv  | 1.95     | 1.28E-23 |
> | SoftShape  | 1.03     | -        |
>
> Since Shapeformer requires a time-consuming shapelet discovery process before training, we provide a comparison on the 18 UCR datasets selected in the main text for Shapeformer.
>
> | Method     | Avg. Rank      | P-value  |
> |------------|---------------|----------|
> | Shapeformer| 1.94   | 3.13E-06 |
> | SoftShape  | 1             | -        |
>
> The results demonstrate that SoftShape outperforms both ShapeConv and ShapeFormer on the UCR time series datasets.
>
> [3] Shapeformer: Shapelet transformer for multivariate time series classification. KDD, 2024.

---

> > ### Comment · Reviewer_6rFK · 2025-04-04
> >
> > Thanks for the rebuttal, which addressed my concerns. I am happy to maintain the current rating.

---

> > > ### Author Response · Authors · 2025-04-04
> > >
> > > Thank you very much for your timely response and positive feedback.

---

### Official Review · Reviewer_byt4 · 2025-03-08

**Overall Recommendation:** 4

**Summary:**

This paper presents SoftShape, a learning-based soft sparse shapes model for time series classification, designed to enhance model interpretability. Specifically, SoftShape introduces the soft shape sparsification, replacing hard shapelets with soft shapelets to improve training efficiency. Moreover, SoftShape employs a dual-pattern learning approach, integrating Mixture-of-Experts (MoE)-driven and sequence-aware-driven mechanisms to capture both intra-shape and inter-shape temporal patterns, thereby improving the discriminability of learned soft shapes. Extensive experiments on 128 UCR time series datasets demonstrate that the proposed SoftShape model outperforms baseline methods, achieving state-of-the-art classification performance.

**Claims And Evidence:**

The submission is clear and convincing, as the authors have made the source code of the proposed model available and provided detailed experimental procedures along with comprehensive results in the appendix, thereby enhancing the study’s reproducibility.

**Essential References Not Discussed:**

[1]. LightTS: Lightweight Time Series Classification with Adaptive Ensemble Distillation, SIGMOD 2023.

[2]. UniTS: A Unified Multi-Task Time Series Model, NeurIPS 2024.

[3]. One Fits All: Power General Time Series Analysis by Pretrained LM, NeurIPS 2023.

**Experimental Designs Or Analyses:**

The overall experimental design and analysis presented in the paper are reasonable; however, there are several issues that need to be addressed:
1. The authors use MoE router-activated class-specific expert networks with MLPs to learn intra-shape temporal patterns. Meanwhile, they employ a CNN-based Inception module as a shared expert for learning inter-shape temporal patterns. However, in the experimental section, the authors do not provide an in-depth analysis of the networks used for class-specific experts and the shared expert.
2. The authors mention that the proposed SoftShape model requires 50 epochs for warm-up training in the experimental setup, but they do not discuss the rationale for using warm-up training in the experimental analysis. It is recommended that the authors include this aspect in the ablation study and provide further discussion.

**Methods And Evaluation Criteria:**

The benchmark datasets and evaluation criteria employed in this paper are appropriate. Additionally, the proposed method is well-suited for time series classification tasks, especially in real-world applications where interpretability is a key requirement.

**Other Comments Or Suggestions:**

The authors refer to the parameter q in Table 5 as the "sliding window size," whereas in Section 3.2, the term "fixed step size" is used. It is recommended that the authors review this terminology and make corresponding adjustments to ensure consistency.

**Other Strengths And Weaknesses:**

Strengths:
1. The writing and structure of the paper are clear, and the proposed soft shapelet sparsification approach effectively combines the interpretability advantages of shapelets for time series classification while significantly reducing the computational cost of model training.
2. The use of a mixture of experts to learn intra-shape temporal patterns is innovative. By employing selected experts to learn class-specific features, it enhances the discriminability of the learned shapes.
Weaknesses:
1. The authors do not thoroughly discuss the impact of the chosen expert networks for intra-shape and inter-shape learning on the final classification performance of SoftShape in the experimental section.
2. In Section 4.3.1 on intra-shape learning, the authors do not provide a detailed explanation of how the MoE combines the learned features from different experts to obtain the final intra-shape representations.
3. It would be better to compare the proposed method with LightTS [1], a lightweight time series classification framework.
4. It is encouraged to compare the proposed method with existing time series foundation models, such as UniTS [2] and OFA [3]. But this is not necessary.

**Questions For Authors:**

1. In Section 4.3.1, regarding the MoE approach for learning intra-shape temporal patterns, could the authors explain how the intra-shape features learned by each expert network (as described in Equation 8) are combined to form the final intra-shape representations?
2. For intra-shape learning, the authors use an MLP network as the base network for each expert. What impact would replacing this with networks such as FCN or TSLANet have on the classification performance of the proposed SoftShape model?
3. In the context of inter-shape learning, the authors use a CNN-based Inception module as the shared expert. How would substituting this with other architectures, such as a Transformer, affect the classification performance of SoftShape?
4. The authors indicate that the proposed SoftShape model requires 50 epochs for warm-up training. What adverse effects might occur if the warm-up training were removed?
5. In Figure 5, panel (a) appears to contain a significantly larger number (or density) of samples compared to panels (b), (c), and (d). Could the authors clarify why this discrepancy exists?
6. The proposed framework seems general and can also be applied to time series forecasting. In addition, the baselines are most designed for time series forecasting. Are there any specific modules for classification? Could you assess the performance of the proposed method on time series forecasting (this is not necessary)?

**Relation To Broader Scientific Literature:**

The use of soft shapelets for time series classification in this paper is novel and significantly improves interpretability, particularly in critical areas like medical time series classification. Building on prior shapelet-based methods, this approach introduces a more flexible and interpretable framework (i.e., MoE-driven and sequence-aware), offering substantial advancements in fields where understanding model decisions is crucial.

**Theoretical Claims:**

I have thoroughly examined the equations supporting the theoretical claims presented in this paper and have found no issues.

---

> ### Author Rebuttal · Authors · 2025-03-31
>
> **W1: The authors do not provide an in-depth analysis of the networks used for class-specific experts and the shared expert.**
>
> **A**: Please refer to the answers in Q2 and Q3.
>
> ---
>
> **W2 & Q4:  The authors do not discuss the rationale for using warm-up training.**
>
> **A**: During the early training phase, the model struggles to distinguish soft shapes, resulting in the fusion of many discriminative ones. Hence, we enable shape sparsification after warm-up training. Experiments on 18 selected UCR datasets showed average ranks of 1.27 and 1.56 with and without warm-up training, respectively, highlighting the strategy's effectiveness.
>
> ---
>
> **W3 & Q1: The authors do not provide a detailed explanation of how the MoE combines the learned features from different experts to obtain the intra-shape representations.**
>
> **A**: For each expert's input, we use the top k indices from the MoE router to determine the activated experts for all input shapes. First, we store the shape indices in an array. After all experts output their intra-shape representations, we combine them in the original order using the stored indices.
>
> ---
>
> **W4: It would be better to compare the proposed method with LightTS, a lightweight time series classification framework.**
>
> **A**: LightTS uses distillation to reduce runtime by ensembling multiple models. It consists of two stages: teacher and student. The teacher phase trains at least 10 classification models (e.g., InceptionTime), which are then used for distillation to create the student model. Since the teacher phase is time-consuming, we performed a comparison using the selected 18 UCR datasets.
>
> | Method             | Avg. Rank | P-value   |
> |--------------------|-----------|-----------|
> | LightTS (Student)  | 3.00      | 3.72E-14  |
> | LightTS (Teacher)  | 1.71      | 7.87E-03  |
> | SoftShape          | 1.18      | -         |
>
> The results show that SoftShape outperforms both LightTS (Teacher) and LightTS (Student).
>
> ---
>
> **W5: It is encouraged to compare the proposed method with existing time series foundation models, such as UniTS and OFA .**
>
> **A**: OFA corresponds to GPT4TS in the main text. We compared UniTS and OFA on the UCR 128 time series dataset:
>
> | Method     | Avg. Rank | P-value    |
> |------------|-----------|------------|
> | OFA        | 2.40      | 1.90E-12   |
> | UniTS      | 2.30      | 5.07E-12   |
> | SoftShape  | 1.21      | -          |
>
> SoftShape significantly outperforms both UniTS and OFA.
>
> ---
>
> **Q2: . What impact would replacing the MoE expert as FCN or TSLANet have on the classification performance of the proposed SoftShape model?**
>
> **A**: Our experiments on the 18 selected UCR datasets show that replacing the original MoE expert network with FCN and TSLANet reduces SoftShape's performance.
>
> | Method              | Avg. Rank |
> |---------------------|-----------|
> | FCN                 | 2.56      |
> | TSLANet             | 1.83      |
> | Original MoE expert | 1.44      |
>
>
> ---
>
> **Q3: How would substituting the shared expert other architectures, such as a Transformer, affect the performance of SoftShape?**
>
> **A**: We replaced the shared expert network with an MLP and Transformer, and the performance on the 18 selected UCR datasets is as follows:
>
> | Method      | Avg. Rank |
> |-------------|-----------|
> | Transformer | 2.22      |
> | MLP         | 2.44      |
> | Inception   | 1.11      |
>
> The results show that Inception performs better.
>
> ---
>
> **Q5: In Figure 5, panel (a) appears to contain a significantly larger number (or density) of samples compared to panels (b), (c), and (d). Could the authors clarify why this discrepancy exists?**
>
> **A**: Fig. 5 (a) shows the representations of all shapes, while Fig. 5 (b), (c), and (d) display the representations after soft shape sparsification. As a result, the shape density in (b), (c), and (d) is lower than in (a).
>
> ---
>
> **Q6: Are there any specific modules of SoftShape for classification? Could you assess the performance of the proposed method on time series forecasting?**
>
> **A**: Shapelets, first introduced by Ye & Keogh (2009) for time series classification, enhance interpretability. Most existing shapelet-based methods focus on classification tasks. SoftShape, a shapelet-based method, aims to improve performance and interpretability in classification tasks.
> We assessed SoftShape's forecasting performance on the following datasets using MSE-based average rank under TimesNet and TS2Vec's setting.
>
> | Method         | ETTh1 | ETTh2 | ETTm1 | ETTm2 |
> |----------------|-------|-------|-------|-------|
> | TS2Vec         | 6     | 6     | 4     | 5.6   |
> | TimesNet       | 3.4   | 4.2   | 4.6   | 4.92  |
> | PatchTST       | 4.4   | 2.8   | 4.4   | 2.88  |
> | GPT4TS         | 3     | 3.8   | 1.4   | 2.28  |
> | iTransformer   | 1.2   | 1     | 3.2   | 1.84  |
> | SoftShape | 3   | 3.2   | 3.4   | 3.48  |
>
> The above results show that SoftShape outperforms TS2Vec and TimesNet, indicating its potential for time series forecasting tasks.

---

> > ### Comment · Reviewer_byt4 · 2025-04-04
> >
> > Thanks for the response, which fixed my problems. I will increase my score slightly.

---

> > > ### Author Response · Authors · 2025-04-04
> > >
> > > Thank you very much for your timely response and positive comments.

---

### Official Review · Reviewer_cgnW · 2025-03-14

**Overall Recommendation:** 4

**Summary:**

This paper focus on time-series classification using shapelets. It introduce an  attention based sparsification mechanism that merges the less discriminative subsequences into a single shape based their learned attention scores. A Mixture of Experts (MoE) architecture is used to learn intra-shape patterns   and a shared expert to learn inter-shape patterns (temporal relationships between shapes). The method is evaluated  on 128 UCR time series datasets and achieves state-of-the-art results.

**Claims And Evidence:**

- Interpretability: The interpretability of the method is evaluted using  Multiple Instance Learning (MIL) on the Trace and Lightning2 datasets. This evaluation shows that the method assigns higher attention scores to subsequences with significant differences between classes.
- Performance: The paper claims superior performance   against state-of-the-art approaches. This is demonstrated through extensive experiments on 128 UCR time series datasets.
- Efficiency : A training time comparison is conducted wrt  three baselines (Medformer,TSLANet,InceptionTime). It shows that the proposed method is faster compared to these baselines. However I think that MultiRocket-Hydra (MR-H)  should be added to this comparison as a computationally efficient baseline.

**Essential References Not Discussed:**

Essential References are cited and discussed in the paper.

**Experimental Designs Or Analyses:**

The paper didn't propose any new experimental design. The method is evaluated on a classical time-series classification benchmark.

**Methods And Evaluation Criteria:**

- The proposed method is well motivated and the main design choices are correctly ablated.
- The method is extensively evaluated  on 128 UCR time series datasets a well recognized benchmark for TSC  (Ismail Fawaz et al., 2019).

**Other Comments Or Suggestions:**

The  paragraph listing the baselines used for comparisons (L#300 - L#316) need to be structured in a more informative way by indicated to which categorie each method belongs.

**Other Strengths And Weaknesses:**

**Strenghts**
- The paper is well written and easy to follow.
- The idea of using shapelets as token with a MoE model combined with soft shapelets selection seems novel and shows promising results.

**Weaknesses:**
-  While training times are compared, a comparison of inference times and number of parameters  is missing from the paper.

**Questions For Authors:**

See weaknesses.

**Relation To Broader Scientific Literature:**

The  contribution of the paper is two-fold:
- The computationally intensive shapelets sparsification is addressed with  soft selection instead of  the hard selection approach adopted by recent approaches (Li et al., 2021; Le et al., 2024) . Instead of discarding the less discriminative subsequences, the proposed method merges them using their attention scores. It also weights the discriminative shapelets with their corresponding scores to account for their varying importance.
- A mixture of experts  router is used to activate **classe-specific** experts for intra-shape temporal patterns. Similar to how patch tokens (shapelets here)  are processed by  mixture of experts models in  computer vision  (Chowdhury et al., 2023).

**Theoretical Claims:**

There are no theoretical claims in the paper.

---

> ### Author Rebuttal · Authors · 2025-03-31
>
> **W1: MultiRocket-Hydra (MR-H) should be added to this comparison as a computationally efficient baseline.**
>
> **A**: MR-H is a combination of the Hydra [1] and MultiRocket [2] algorithm.
>
> - Hydra uses randomly initialized convolutional kernels, grouped into $g$ groups per dilation with  $k$ kernels per group. These kernels transform input time series and count the closest matches at each time point. The counts for each group are concatenated and used to train a linear classifier.
>
> - MultiRocket applies random initialized convolutional kernels to the time series, performs standard scaling, and fits a classifier using the transformed data (default: RidgeClassifierCV).
>
> Hydra and MultiRocket utilize randomly initialized convolutional kernels to extract time series features, in contrast to deep learning methods that rely on a backpropagation algorithm. As a result, MR-H exhibits a fast runtime.
>
> The core modules of MR-H do not employ deep learning techniques and operate efficiently. Thus, the official implementation code of MR-H provided by the authors is based on a CPU version.  Hence, the reported training and inference times for MR-H are measured using a CPU, which is consistent with other deep learning-based baselines.
>
> Based on training time analysis from Figure 3 in the main text, MR-H took $411$ seconds on ChlorineConcentration and $64$ seconds on HouseTwenty, significantly faster than SoftShape (2743 and 280 seconds) and MedFormer (14377 and 1296 seconds). We will include a runtime analysis of MR-H in the revised version of Figure 3.
>
> [1] Hydra: Competing convolutional kernels for fast and accurate time series classification. DMKD, 2023.
>
> [2] MultiRocket: multiple pooling operators and transformations for fast and effective time series classification. DMKD, 2022.
>
> ---
>
> **W2: While training times are compared, a comparison of inference times and number of parameters is missing from the paper.**
>
> **A**: The sample length of a single time series may slightly influence the model parameter count for certain methods. The table below shows the average parameter count for the comparison methods across 128 UCR time series datasets, with $K$ denoting one thousand.
>
> | Method        | # Parameters  |
> |---------------|---------------|
> | FCN           | 266.9 K       |
> | T-Loss        | -             |
> | SelfTime      | -             |
> | TS-TCC        | 495.3 K       |
> | TST           | 25725.3 K     |
> | TS2Vec        | 1274.5 K      |
> | TimesNet      | 7428.3 K      |
> | PatchTST      | 1225.4 K      |
> | GPT4TS        | 11421.1 K     |
> | RDST          | -             |
> | MR-H          | -             |
> | InceptionTime | 387.7 K       |
> | ModernTCN     | 315.6 K       |
> | TSLANet       | 514.6 K       |
> | Medformer     | 1360.6 K      |
> | SoftShape     | 472.5 K       |
>
> T-loss and SelfTime are excluded from the parameter count due to difficulties in obtaining the parameter information from their official implementation code. Furthermore, RDST and MR-H are not deep learning algorithms, and thus, their parameter counts are not reported.
>
> Additionally, as baseline settings in Figure 3 of the main text, we report the inference times (in seconds) for SoftShape, MedFormer, TSLANet, InceptionTime, and MR-H on the ChlorineConcentration (4,307 samples, length 166) and HouseTwenty (159 samples, length 2,000) datasets.
>
> | Method         | ChlorineConcentration | HouseTwenty |
> |----------------|-----------------------|-------------|
> | MR-H           | 6.60                  | 2.97        |
> | InceptionTime  | 1.41                  | 1.30        |
> | TSLANet        | 1.37                  | 1.29        |
> | Medformer      | 1.48                  | 1.31        |
> | SoftShape      | 1.39                  | 1.26        |
>
> It is important to note that MR-H is executed on a CPU, whereas other deep learning methods are run on an NVIDIA GeForce RTX 3090 GPU. Overall, the differences in inference time between the deep learning methods are negligible. However, we observed that SoftShape demonstrates a slight advantage in inference time on the HouseTwenty dataset, which has a longer sequence length.
>
> ---
>
> **W3: The paragraph listing the baselines used for comparisons (L#300 - L#316) need to be structured in a more informative way by indicated to which category each method belongs.**
>
> **A**: Thank you for your suggestions. We have classified the 15 baseline methods into two primary groups: Deep Learning-based and Non-Deep Learning-based methods. Among the Deep Learning-based methods, we have further subdivided them into two categories based on their network architecture: CNN-based and Transformer-based. This categorization will be updated in the revised version. The specific categories are as follows:
>
> - Deep Learning-based methods:
>
>     a) CNN-based: FCN, T-Loss, SelfTime, TS-TCC, TS2Vec, TimesNet, InceptionTime, ModernTCN, TSLANet.
>
>     b) Transformer-based: TST, PatchTST, GPT4TS, Medformer.
>
> - Non-Deep Learning-based methods: RDST, MR-H.

---

### Decision · Program_Chairs · 2025-05-01

**Decision:**

Accept (spotlight poster)

**Comment:**

This paper proposed a Soft sparse Shapes (SoftShapes) model for efficient time series classification. In order to do so the authors introduce soft shape sparsification and soft learning blocks. All reviews are very positive, recognize the novelty of the findings, and appreciate the presentation of the paper. The discussion between the reviewers and the authors was productive, and the paper's scores improved after the rebuttal. Therefore, the AC suggests acceptance and encourages the authors to include the rebuttal discussion in the final version of the paper.